# Directed Semi-Simplicial Learning with Applications to Brain Activity Decoding

**Manuel Lecha**[1*]    **Andrea Cavallo**[2]    **Francesca Dominici**[3]    **Ran Levi**[4]
**Alessio Del Bue**[1]    **Elvin Isufi**[2]    **Pietro Morerio**[1]    **Claudio Battiloro**[3*]

[1]Istituto Italiano di Tecnologia [2]Delft University of Technology
[3]Harvard University [4]University of Aberdeen

## Abstract

Graph Neural Networks (GNNs) excel at learning from pairwise interactions but often overlook multi-way and hierarchical relationships. Topological Deep Learning (TDL) addresses this limitation by leveraging combinatorial topological spaces, such as simplicial or cell complexes. However, existing TDL models are restricted to undirected settings and fail to capture the higher-order directed patterns prevalent in many complex systems, e.g., brain networks, where such interactions are both abundant and functionally significant. To fill this gap, we introduce Semi-Simplicial Neural Networks (SSNs), a principled class of TDL models that operate on semi-simplicial sets, combinatorial structures that encode directed higher-order motifs and their directional relationships. To enhance scalability, we propose Routing-SSNs, which dynamically select the most informative relations in a learnable manner. We theoretically characterize SSNs by proving they are strictly more expressive than standard graph and TDL models, and they are able to recover several topological descriptors. Building on previous evidence that such descriptors are critical for characterizing brain activity, we then introduce a new rigorous framework for brain dynamics representation learning centered on SSNs. Empirically, we test SSNs on 4 distinct tasks across 13 datasets, spanning from brain dynamics to node classification, showing competitive performance. Notably, SSNs consistently achieve state-of-the-art performance on brain dynamics classification tasks, outperforming the second-best model by up to 27%, and message passing GNNs by up to 50% in accuracy. Our results highlight the potential of topological models for learning from structured brain data, establishing a unique real-world case study for TDL.

## 1 Introduction

Networks are commonly represented as graphs, i.e., a set of nodes and a set of unordered pairs of nodes, the edges, modeling pairwise interactions (Barabási, 2002). Graph Neural Networks (GNNs) (Scarselli et al., 2008), deep learning models operating on graph-structured data, have shown remarkable performance on several tasks from different domains, such as computational chemistry (Gilmer et al., 2017; Jumper et al., 2021), social network analysis (Xia et al., 2021; Kipf & Welling, 2017), and neuroscience (Bessadok et al., 2022). The success of GNNs is mainly due to their ability to synergize the flexibility of deep learning models with the inductive bias encoded in the graph structure (Bruna et al., 2014; Gori et al., 2005). Most GNNs are Message Passing Neural Networks (MPNNs) (Gilmer et al., 2017), which learn meaningful representations of node or edge data via local aggregation governed by the underlying graph connectivity.

However, many complex real-world systems exhibit higher-order interactions that go beyond simple pairwise relationships (Battiston et al., 2020; Millán et al., 2025). Such dependencies are naturally modeled by *Combinatorial Topological Spaces* (CTSs), mathematical objects such as simplicial or cell complexes that generalize graphs by encoding multi-way interactions as sets of nodes, i.e., the *simplices/cells*. CTSs induce set-containment relations among higher-order entities via hierarchical inclusion, extending conventional adjacency in graphs (Grady & Polimeni, 2010) and enabling powerful algebraic topological tools (Barbarossa & Sardellitti, 2020). *Topological Deep Learning* (TDL) (Battiloro, 2024; Bodnar, 2023; Hajij et al., 2022b; Papamarkou et al., 2024) builds on this structure,

---

*Corresponding authors. Emails: `manuel.lecha@iit.it`, `cbattiloro@hsph.harvard.edu`.

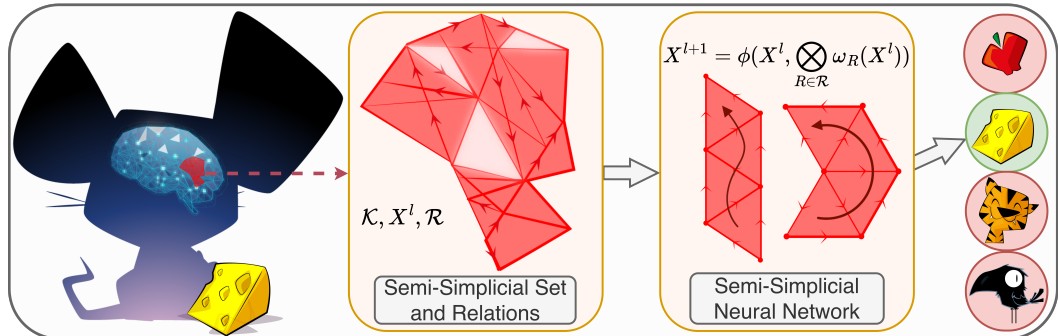

Figure 1: **Overview of the Semi-Simplicial Neural Networks framework for brain dynamics classification.** Given a connectome sample, represented as a directed graph (digraph), and its corresponding binary activity response to external stimulation, we jointly model them as an attributed semi-simplicial set $\mathcal{K}$ that captures higher-order co-activation patterns $X^l$. We then select a set of higher-order directed relations $\mathcal{R}$ induced by the topology of $\mathcal{K}$, and process $X^l$ with *Semi-Simplicial Neural Networks* (SSNs), where the relations in $\mathcal{R}$ define how information is propagated and updated. Our experiments show that this modeling choice is crucial for accurately predicting the eliciting stimulus.

extending GNNs into architectures that operate directly on CTSs. Compared to standard GNNs, TDL models have demonstrated stronger expressivity within the Weisfeiler–Leman (WL) hierarchy (Horn et al., 2022; Bodnar et al., 2021b), improved ability to capture long-range dependencies (Battiloro et al., 2025a; Giusti et al., 2023b;a), and robustness in heterophilic regimes (Bodnar et al., 2022).

Yet, simplicial complexes and other commonly used CTSs (Battiloro et al., 2024b; Bodnar et al., 2021a; Hajij et al., 2022b) remain limited when *directionality* governs system dynamics. Here, *directionality* refers to structural asymmetry; for example, a directed edge has a source and a target, and its reverse is a different edge, with one not implying the other—unlike orientation, which merely assigns a sign to the same undirected edge (e.g., positive vs. negative flux; see App. C.3–C.4). While several GNN variants incorporate directionality at the level of directed graphs (digraphs) (Tong et al., 2020a; Zhang et al., 2021; Rossi et al., 2024), the challenge of modeling *higher-order directional structure* in TDL remains largely unexplored. Building on the foundations of Riihimäki (2023), the short paper of Lecha et al. (2025) introduced *Directed Simplicial Neural Networks* (Dir-SNNs), based on directed cliques, totally ordered node sequences, connected in a direction-preserving manner. While this work formalized message passing on directed simplicial complexes, it is confined to a narrow class of spaces, lacks theoretical guarantees, and provides only limited synthetic validation.

*Brain networks* are among the most notable real-world domains requiring methods capable of processing higher-order, directional information (Giusti et al., 2016). Digraphs naturally capture the asymmetric flow of information in neuronal communication, from presynaptic to postsynaptic neurons, but fail to encode the higher-order co-activation patterns critical for understanding brain function. Higher-order directed motifs are abundant across scales (Sizemore et al., 2018; 2019), carry functional meaning (Nolte et al., 2019), and form spiking assemblies that enhance the representational fidelity of neural activity (Reimann et al., 2017; Ecker et al., 2024). To capture such structures, the emerging field of *Neurotopology* (Conceição et al., 2022; Reimann et al., 2017) employed semi-simplicial sets (Hatcher, 2005), general CTSs exemplified by directed flag complexes and tournaplexes (Govc et al., 2021). Conceição et al. (2022) proposed a topological featurization pipeline that aggregates invariants across meaningful neuron subpopulations, and together with Reimann et al. (2022), showed that such descriptors can recover stimulus identity in a biologically realistic neocortical model (Markram et al., 2015), even when traditional approaches fail. However, these handcrafted pipelines are fundamentally limited: they fix representational power in advance through predefined invariants, rely on carefully tuned sampling heuristics to select informative neuron subsets, and lack robustness under perturbations or shuffled activity (Conceição et al., 2022).

Therefore, the absence of a general, formal, and comprehensive TDL framework that leverages higher-order directionality motivates the methodological contributions of this work. Moreover, connecting these developments to neurotopology creates a unique opportunity to learn meaningful representations of dynamical brain activity, and motivates our applied contributions. For a detailed discussion of related work, see Appendix B.

**Contribution.** The methodological and applied advances of this work are listed below, and they address several open problems in TDL as identified by Papamarkou et al. (2024).

**C1.** We introduce *Semi-simplicial Neural Networks* (SSNs), the first TDL models explicitly designed for semi-simplicial sets. The rich variety of ways SSNs propagate information across simplices is formalized via face-map–induced relations collected in a relational algebra and generalizing common (directed) graph and topological adjacencies. To address scalability and efficiency (Open Problem 6 in (Papamarkou et al., 2024)), we further propose *Routing-SSNs* (R-SSNs), which employ a learnable routing mechanism (Shazeer et al., 2017; Wang et al., 2023) to dynamically select the top-$k$ most relevant relations, reducing parameter count and inference time. Theoretically, we prove that SSNs are strictly more expressive than message-passing GNNs (Gilmer et al., 2017), Directed GNNs (Dir-GNNs) (Rossi et al., 2024), and Message-Passing Simplicial Neural Networks (MPSNNs) (Bodnar et al., 2021b) in the WL hierarchy (Section 3).

**C2.** We propose a novel *topology-grounded framework* for higher-order representation learning of brain dynamics. Its core elements are *Dynamical Activity Complexes (DACs)*—directed simplicial complexes endowed with binary, time-evolving, features encoding neuronal co-activation: a neuron group is active at time $t$ if all constituent neurons fire simultaneously at $t$. Critically, we formally prove that SSNs operating on DACs are uniquely capable, among existing graph and TDL models, of recovering a broader class of topological invariants known to characterize brain network activity (Reimann et al., 2022), thanks to their ability to jointly encode directionality and higher-order structure (Open Problem 9 in (Papamarkou et al., 2024)). See Section 4 for details.

**C3.** We test SSNs on 4 distinct tasks across 13 datasets, spanning from brain dynamics to node classification, showing competitive performance. The brain dynamics classification tasks represent a *unique, meaningful, real-world case study for TDL*, where SSNs consistently achieve state-of-the-art results (see Figure 1, Open Problem 1 in (Papamarkou et al., 2024)). As such, the data and tasks also serve as a competitive public benchmark for graph-based and TDL models (Open Problem 2 of (Papamarkou et al., 2024)). Specifically, following Conceição et al. (2022); Reimann et al. (2022), we leverage stimulus–response activity from a biologically realistic neocortical microcircuit (Markram et al., 2015), evaluating: (i) classification from fixed topological brain samples-i.e., a feature classification task, and (ii) classification from randomly sampled neuron neighborhoods-i.e., a graph/complex classification task. SSNs achieve accuracy gains of over 50% compared to baseline MPNNs (Gilmer et al., 2017). Full details are in Section 5.

## 2 PRELIMINARIES

In this section, we revisit the foundational building blocks of our framework. We begin with *semi-simplicial sets*, which extend *simplicial complexes* by allowing multiple simplices over the same vertex set and support directionality. We then discuss *directed simplicial complexes* as a key subclass, with *directed flag complexes* arising canonically from digraphs. Next, we introduce a rich class of *face-map–induced relations*, which generalize standard topological adjacencies and extend them to higher-order directed structures. Finally, we formalize data integration via *attributed semi-simplicial sets*, focusing on time-varying binary features relevant to brain dynamics.

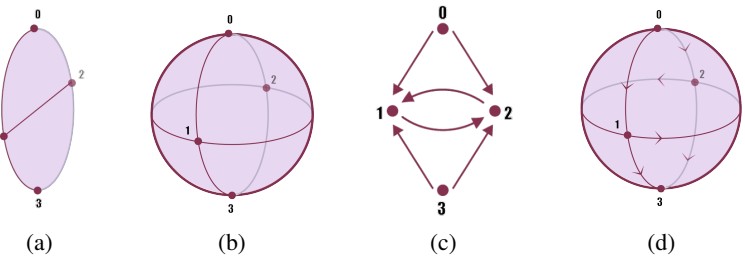

|      |      |      |      |
| :--: | :--: | :--: | :--: |
| (a)  | (b)  | (c)  | (d)  |

Figure 2: (a) A simplicial complex of dimension 2; (b) a semi-simplicial set of dimension 2; (c) a digraph; (d) its associated directed flag complex.

**Simplicial Complexes.** A simplicial complex is a pair $\widetilde{\mathcal{K}} = (V, \Sigma)$, where $V$ is a finite set of vertices and $\Sigma$ is a collection of non-empty finite subsets of $V$ satisfying two properties: **(P1)** for every $v \in V$, the singleton $\{v\}$ is in $\Sigma$; and **(P2)** if $\sigma \in \Sigma$ and $\tau$ is a non-empty subset of $\sigma$, then $\tau \in \Sigma$. The dimension of a simplex $\sigma \subseteq V$ is $\dim(\sigma) = |\sigma| - 1$; the dimension of the complex is $N = \max_{\sigma \in \Sigma} \dim(\sigma)$. We denote the set of $n$-simplices by $\Sigma_n = \{\sigma \in \Sigma \mid \dim(\sigma) = n\}$, thus $\Sigma = \bigcup_{n=0}^N \Sigma_n$. An example of a 2-dimensional simplicial complex is shown in Figure 2(a), with vertices $V = \{0, 1, 2, 3\}$, edges $\Sigma_1 = \{\{0,1\}, \{0,2\}, \{1,2\}, \{2,3\}, \{1,3\}\}$, and 2-simplices $\Sigma_2 = \{\{0,1,2\}, \{1,2,3\}\}$.

**Semi-Simplicial Sets.** Semi-simplicial sets generalize simplicial complexes by allowing multiple distinct simplices to share the same vertex set. Figure 2 illustrates this: in (b), the 2-simplices $(0, 1, 2)$ and $(0, 2, 1)$ represent two different triangles over the same vertex set $\{0, 1, 2\}$, whereas in (a) a simplicial complex admits only one simplex on $\{0, 1, 2\}$. This added flexibility is crucial for modeling directionality. For instance, in a digraph the edges $(0, 1)$ and $(1, 0)$ are distinct, even though both correspond to $\{0, 1\}$. Formally, a semi-simplicial set $\mathcal{S}$ consists of (i) sets $\{S_n\}_{n=0}^N$, where $S_n$ is the set of $n$-simplices, and (ii) face maps $d_i : S_n \to S_{n-1}$ for $n > 0$ and $0 \le i \le n$, specifying how simplices are glued. These maps satisfy the simplicial identity $d_i d_j = d_{j-1} d_i$ for $i < j$, ensuring consistency of the face structure. We denote the total set of simplices by $S = \bigcup_{n=0}^N S_n$.

**Directed Simplicial Complexes.** A directed simplicial complex $\mathcal{K}$ is a semi-simplicial set in which each simplex encodes a fully transitive directed structure. Formally, each set of $n$-simplices $\mathcal{K}_n$ consists of ordered $(n + 1)$-tuples $\sigma = (v_0, \dots, v_n)$ such that $(v_i, v_j)$ is a directed edge for every $i < j$. Directed simplicial complexes of dimensions 1 and 2 are illustrated in Figures 2(c–d). The face maps $d_i^n : \mathcal{K}_n \to \mathcal{K}_{n-1}$ act by deleting the $i$-th vertex, i.e., $d_i^n(\sigma) = (v_0, \dots, \hat{v}_i, \dots, v_n)$. For example, the 1-simplex $(0, 1)$ in Figure 2(c) has faces $d_0^1((0, 1)) = (1)$ and $d_1^1((0, 1)) = (0)$. We require $\mathcal{K}$ to contain all faces of its simplices, i.e., satisfy properties **(P1)** and **(P2)** described above.

**Directed Flag Complexes.** Given a digraph $\mathcal{G} = (V, E)$, the induced directed flag complex $\mathcal{K}_{\mathcal{G}}$ (Lütgehetmann et al., 2020) is the directed simplicial complex whose $n$-simplices are precisely the transitive $(n+1)$-cliques of $\mathcal{G}$—that is, ordered tuples $(v_0, \dots, v_n)$ of distinct vertices such that $(v_i, v_j) \in E$ for all $i < j$. In other words, each directed clique in $\mathcal{G}$ is promoted to a simplex. Figures 2(c–d) illustrate a digraph and its directed flag complex. Importantly, this construction is injective on isomorphism classes: non-isomorphic digraphs map to non-isomorphic complexes, ensuring that no structural information from the original digraph is lost (see Appendix C.1).

**Face Map Relations.** We make simplices interact in a direction-sensitive way using relations induced by face maps, generalizing binary adjacency relations in graphs, digraphs, and simplicial complexes. Each face map $d_i^n$ defines a binary relation $R_{d_i^n} = \{(\tau, \sigma) \mid \sigma \in S_n, \, d_i^n(\sigma) = \tau\}$, linking an $n$-simplex to its $i$-th face. Collectively, these relations generate a *face-map relation algebra* $\mathcal{R}_d$, closed under standard relational operations union $\cup$, intersection $\cap$, composition $\circ$ (chaining relations), and converse $^\top$ (reversing them; see App. C.5). For example, in directed simplicial complexes, for $n \ge 1$, $R_{n\downarrow,i,j} := R_{d_j^n}^\top \circ R_{d_i^n} = \{(\sigma, \tau) \mid d_i^n(\sigma) = d_j^n(\tau)\}$ relates two $n$-simplices whose $i$-th and $j$-th facets coincide (Fig. 3 (Top)). Compositions of such relations naturally define directed paths across simplices (Fig. 3 (Bottom)). Further illustrative examples are given in App. C.6.

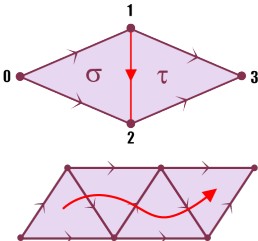

Figure 3: (Top) Directed 2-simplices $\sigma$ and $\tau$ related by $d_0^2(\sigma) = d_2^2(\tau) = e$, where $e$ denotes the shared edge highlighted in red. Hence $(\sigma, \tau) \in R_{2\downarrow,0,2}$. (Bottom) Composing $R_{2\downarrow,0,2}$ yields a directed path across 2-simplices.

**Attributed Semi-Simplicial Sets.** An attributed semi-simplicial set is a pair $\mathcal{S}_F = (S, F)$ where $S$ is a semi-simplicial set and $F$ is a map $F : S \to \mathbb{F}$ assigning each simplex $\sigma \in S$ a feature vector $x_\sigma = F(\sigma) \in \mathbb{R}^m$. For a chosen indexing[1] of the $n$-simplices, we define the *$n$-feature matrix* $X_n := [x_{\sigma_1}; \dots; x_{\sigma_{|S_n|}}] \in \mathbb{R}^{|S_n| \times m}$. Extending the indexing globally, we concatenate across all dimensions to obtain the global feature matrix $X := [X_0; \dots; X_N] \in \mathbb{R}^{|S| \times m}$. In this work, we often focus on the special case of *dynamic binary data*, where $\mathbb{F} = \mathbb{B}^T = \{0, 1\}^T \subset \mathbb{R}^T$. A binary dynamics map $B$ assigns to each simplex a binary activation vector $x_\sigma \in \{0, 1\}^T$ indicating whether it is active (1) or inactive (0) at each of $T$ discrete time steps.

---

[1] Given finite set $S$, a bijection $\{1, \dots, |S|\} \to S$.

## 3 SEMI-SIMPLICIAL NEURAL NETWORKS

TDL architectures typically represent higher-order relations as unordered vertex sets. In simplicial complexes, each vertex set admits at most one simplex, which prevents assigning distinct features to different orderings of the same vertices. While this is appropriate for symmetric interactions (Benson et al., 2016; Battiston et al., 2020), it breaks down when order is essential, as in brain dynamics (Markram et al., 2015), where the same clique of neurons may encode different information depending on firing order. Ignoring directionality creates two expressiveness gaps:

*(a) Information loss.* Symmetrizing a directed simplicial complex into an undirected one is non-injective: distinct motifs (e.g., transitive cliques versus directed cycles) can collapse to the same unordered simplex (see Appendix C.3).

*(b) Limited adjacencies.* TDL models typically define interactions via subset containment (simplices communicate only via shared vertices or hierarchical inclusion), which underutilizes directional higher-order propagation.

To address these limitations, we propose SSNs, a principled class of TDL architectures that operate directly on attributed semi-simplicial sets. This section supports Contribution **C1**.

**Semi-Simplicial Neural Networks (SSNs).** SSNs propagate information using face-map–induced relations $\mathcal{R} \subseteq \mathcal{R}_d$, leveraging the algebraic structure of semi-simplicial sets to capture directional motifs and their higher-order interactions. The $l$-th SSN layer updates features $X^l$ as

$$X^{l+1} = \phi\Big(X^l, \bigotimes_{R \in \mathcal{R}} \omega_R(X^l)\Big), \tag{1}$$

where $\omega_R$ is a relation-dependent learnable function, $\bigotimes$ aggregates multiple messages per simplex across relations, and $\phi$ is a learnable update function. This formulation subsumes a wide family of models: if $\omega_R = $ MPNN-D (Rossi et al., 2024), the SSN is a message passing architecture; if $\omega_R$ is masked self-attention, the SSN is a transformer-like architecture.

As a concrete example, for a digraph the relations

$$R_{\text{in}} = R_{d_0^1} \circ R_{d_1^1}^\top, \qquad R_{\text{out}} = R_{d_1^1} \circ R_{d_0^1}^\top$$

recover the standard in- and out-adjacency matrices $A_{\text{in}}$ and $A_{\text{out}}$, respectively. Setting $\mathcal{R} = \{R_{\text{in}}, R_{\text{out}}\}$, $\omega_R = $ MPNN-D, and $\bigotimes = \sum$, the resulting SSN coincides with Dir-GNN (Rossi et al., 2024), i.e.,

$$X^{l+1} = \sigma\big(A_{\text{in}} X^l W_{\text{in}} + A_{\text{out}} X^l W_{\text{out}}\big).$$

Further details are given in Appendix E.1.

**Routing SSNs.** While SSNs fully exploit the combinatorial structure of semi-simplicial sets, two key challenges arise: (1) *Scalability*: explicitly modeling many relations can lead to parameter growth, as each relation requires its own weights; and (2) *Relevance of relations*: not all relations contribute equally to learning, and some may be redundant or uninformative. To address these limitations, we propose Routing SSNs (R-SSNs), which incorporate a learnable gating mechanism (Fedus et al., 2022; Shazeer et al., 2017; Wang et al., 2023) to dynamically select the top-$k$ relations from predefined relation classes.

Given face-map–induced relations $\mathcal{R}$, let $\mathcal{P}_\mathcal{R} = \{\mathcal{R}_1, \ldots, \mathcal{R}_n\}$ be a partition of $\mathcal{R}$ that groups relations into semantic classes, enabling different interaction types to be modeled separately. For instance, one class may represent *interdimensional communication* (e.g., messages from 2-simplices to their 1-faces), while another captures *intradimensional communication* (e.g., direction-aware relations between 2-simplices). The $l$-th R-SSN layer updates features $X^l$ as

$$X^{l+1} = \phi\Big(X^l, \bigoplus_{\hat{\mathcal{R}} \in \mathcal{P}_\mathcal{R}} \bigotimes_{R \in \hat{\mathcal{R}}} G_R(X^l)\, \omega_R(X^l)\Big), \tag{2}$$

where $G_R(X^l) \in [0, 1]$ is a gating function that outputs normalized scores for relations within each class. We instantiate routing via a top-$k$ mechanism, setting $G_R(X^l) = 0$ for all but the $k$ highest-scoring relations in each class. Aggregation across relation classes is performed by $\bigoplus$. A detailed formulation of the routing mechanism, including gating strategies and the auxiliary losses used to prevent routing collapse, is given in Appendix E.2.

### 3.1 Theoretical properties of SSNs

We next provide a theoretical characterization of SSNs, focusing on their generality, WL-expressivity (Xu et al., 2019), and permutation equivariance under simplex reindexing.

**Generality.** SSNs unify and extend prior models across graphs and (directed) simplicial complexes.

**Proposition 1.** *Semi-simplicial neural networks (SSNs) subsume directed message-passing GNNs (Rossi et al., 2024), message-passing GNNs (Gilmer et al., 2017) on undirected graphs, message-passing simplicial neural networks (Bodnar et al., 2021b) on undirected simplicial complexes and Directed Simplicial Neural Networks (Lecha et al., 2025) on directed simplicial complexes.*

**WL Expressivity.** The Weisfeiler–Leman (WL) test (Weisfeiler & Leman, 1968) (see App.D.2) bounds the discriminative power of GNNs (Xu et al., 2019). Graph liftings (e.g., directed flag complexes) enrich representations with higher-order relations, mapping isomorphic graphs to isomorphic CTSs and separating non-isomorphic ones. Hence, TDL models can surpass standard graph-based models in expressivity (Horn et al., 2022). We now show that SSNs go further, strictly exceeding both Dir-GNNs and MPSNNs.

**Theorem 1.** *There exist SSNs strictly more expressive than directed graph neural networks (Dir-GNNs) (Rossi et al., 2024) at distinguishing non-isomorphic directed graphs.*

**Theorem 2.** *There exist SSNs strictly more expressive than message-passing simplicial neural networks (MPSNNs) (Bodnar et al., 2021b) at distinguishing non-isomorphic directed simplicial complexes.*

**Permutation Equivariance.** Matrix representations of semi-simplicial sets require arbitrary simplex indexing (see Section 2), which has no semantic meaning. SSNs must therefore be equivariant (or invariant) to simplex permutations, ensuring representations depend only on the underlying domain (Bronstein et al., 2021).

**Theorem 3** (Informal)**.** *Consider an attributed semi-simplicial set with a collection of face-map–induced relations $\mathcal{R} \subseteq \mathcal{R}_d$. An SSN layer, as defined in equation 1, is equivariant to simplex reindexing if, for each relation $R \in \mathcal{R}$, the message function $\omega_R$ and the aggregation operator $\bigotimes$ are permutation equivariant.*

The proofs of the statements in this section, and a detailed complexity analysis of SSNs are reported in App.D and App.F, respectively.

## 4 Topological Deep Representation Learning for Brain Dynamics

We introduce a principled framework for representation learning on dynamical brain activity, compatible with graph-based and TDL architectures. At its core are *Dynamical Activity Complexes (DACs)*: directed simplicial complexes that encode neuronal co-activation patterns over time. We prove that SSNs operating on DACs can recover a broad class of *invariants*, quantities preserved under equivalence (isomorphism; see App.C.1), that characterize brain activity (Reimann et al., 2017) and are defined in App.D.2. Crucially, this class strictly exceeds the invariants recoverable by graph-based or traditional TDL models. This section supports Contribution *C2*.

**Brain Activity Modeling.** Brain network activity modeling comprises two components: structural connectivity and dynamics. Structural connectivity is modeled by a directed graph $\mathcal{G} = (V, E)$, with nodes $V$ as neurons and edges $(u, v) \in E$ denoting synapses from presynaptic $u$ to postsynaptic $v$. Dynamics are represented by binary functions $\mathcal{B} = \{B : V \to \mathbb{B}^T\}$, encoding firing (1) and quiescent (0) states across $T$ discrete time bins under stimulus-driven inputs. Further details in App. G.1.

**Neurotopology at a Glance.** Given binary dynamics $B$ on a digraph $\mathcal{G} = (V, E)$, each time bin $t$ defines an activation state $B_t$, partitioning neurons into active $V^{1,t} = \{v \mid B_t(v) = 1\}$ and inactive $V^{0,t}$. The active set induces a subgraph $\mathcal{G}^{1,t} = \mathcal{G}[V^{1,t}]$ at time $t$, together with its directed flag complex $\mathcal{K}_{\mathcal{G}^{1,t}}$ capturing co-activation. Applying a topological invariant $\mathrm{T}$ across time yields a temporal signature of activity, $\mathrm{T}(\mathcal{F}_{\mathcal{K}_\mathcal{G}}) = [\mathrm{T}(\mathcal{K}_{\mathcal{G}^{1,1}}), \dots, \mathrm{T}(\mathcal{K}_{\mathcal{G}^{1,T}})] \in \mathbb{R}^T$. Reimann et al. (2017) showed that distinct stimuli induce distinct temporal signatures, measurable via invariants such as Euler characteristic or clique counts. Building on this, Conceição et al. (2022) introduced a topological featurization pipeline for stimuli classification within $\mathcal{G}$, aggregating signatures from neuron samples

with uncommon activity patterns. While effective in realistic neocortical models (Markram et al., 2015), this pipeline faces key limitations: (i) it relies on *predefined invariants*, fixing representational power in advance; (ii) it requires carefully designed *sampling strategies* to select informative neuron subsets, making results sensitive to the chosen subgraphs and to structural variability across samples; and (iii) it has been reported to lack robustness under shuffled or perturbed activity (Conceição et al., 2022). By contrast, our framework allows learning directly from arbitrary topologies, removing the dependence on predefined invariants and sampling heuristics.

**Dynamical Activity Complex.** We introduce *Dynamical Activity Complexes* (DACs), a representation that integrates higher-order structural connectivity with evolving brain activity in a form directly amenable to end-to-end learning. A DAC is a binary directed simplicial complex built from a sampled brain digraph and its associated binary dynamics, encoding activity over $T$ discrete time steps. Formally, let $\mathcal{G}_B$ be a dynamic binary digraph, where each vertex $v \in V$ is assigned a $T$-dimensional binary feature vector $B(v) \in \mathbb{B}^T$. The DAC is the attributed directed flag complex $\mathcal{K}_{\mathcal{G}, \tilde{B}} = (\mathcal{K}_\mathcal{G}, \tilde{B})$, with attribute map

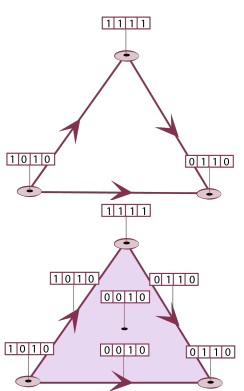

$$\tilde{B}(\sigma) = \left[ \min_{v \in \sigma} B_1(v), \ldots, \min_{v \in \sigma} B_T(v) \right] \in \mathbb{B}^T, \qquad (3)$$

so that a simplex $\sigma$ is active at time $t$ iff all its vertices are active at $t$. This construction captures higher-order co-activation motifs: for instance, in Fig. 4, the 2-simplex activates only at $t = 3$, the unique time all its vertices fire simultaneously. Crucially, DACs (i) *preserve isomorphisms*, and (ii) *encode the full temporal sequence of functional complexes*, retaining the ability to recover established invariants in neurotopology pipelines (Propositions 4–5 in Appendix G.2).

Figure 4: (Top) Dynamic binary graph; (Bottom) its DAC. Binary vectors indicating simplex activation over 4 time steps.

Table 1: Topological invariant computation across architectures.

| Model | size | dir | hodir | rc | td | ec |
|---|---|---|---|---|---|---|
| GNN | ✓ | ✗ | ✗ | ✗ | ✗ | ✗ |
| Dir-GNN | ✓ | ✓ | ✗ | ✓ | ✗ | ✗ |
| MPSNN | ✓ | ✗ | ✗ | ✗ | ✓ | ✓ |
| **SSN (Ours)** | ✓ | ✓ | ✓ | ✓ | ✓ | ✓ |

**Relevant Invariants.** Neurotopological studies (Reimann et al., 2017; Conceição et al., 2022; Reimann et al., 2022) have empirically validated the set of topological invariants $\mathcal{T} = \{\text{size}, \text{ec}, \text{td}, \text{dir}, \text{hodir}, \text{rc}\}$ as particularly effective for characterizing brain activity, as measured by predictive performance in stimulus-induced activity classification. Briefly: the *size* (size) invariant counts active neighbors per node at each time step, i.e., synaptically connected active neurons; the *directionality* (dir) invariant measures the imbalance between active outgoing and incoming edges per node, i.e., post- versus pre-synaptic neighbors; the *higher-order directionality* (hodir) extends this to $n$-simplices, comparing outgoing and incoming active cliques; the *reciprocity* (rc) invariant counts active nodes with active neighbors in both directions, i.e., neurons that are simultaneously pre- and post-synaptic; the *transitive degree* (td) invariant counts active directed 2-simplices incident to a node, i.e., transitive synaptic triads; and the *Euler characteristic* (ec) is the alternating sum of active simplices across dimensions. By construction of DACs, this Euler characteristic coincides with the one in Proposition 6. See App. G.3 for formal definitions and details.

**Theorem 4.** *Let $\mathcal{G}_B$ be a dynamic binary digraph with corresponding DAC $\mathcal{K}_{\mathcal{G}, \tilde{B}}$. For every invariant $\mathrm{T} \in \mathcal{T}$, there exists a set of face-map-induced relations $\mathcal{R}_\mathrm{T} \subset \mathcal{R}_d$ and a Semi-Simplicial Neural Network SSN as in equation 1 such that:*

$$\mathrm{SSN}(X, \mathcal{R}_\mathrm{T}) = \mathrm{T}(\mathcal{K}_{\mathcal{G}, \tilde{B}}).$$

*Moreover, the class of invariants recoverable by SSNs strictly exceeds that of message-passing neural networks (Gilmer et al., 2017), directed GNNs (Rossi et al., 2024), and message-passing simplicial networks (Bodnar et al., 2021b).*

A formal proof is given in Appendix G.4. Table 1 summarizes invariant computation across models, showing that SSNs uniquely recover the complete set of critical invariants necessary for comprehensive brain activity characterization. This establishes SSNs as a strict generalization of prior approaches, enabling fully data-driven, localized, and meaningful representations of brain dynamics.

## 5 NUMERICAL RESULTS

We empirically validate SSNs on four tasks across 13 datasets: two neural stimulus classification tasks (six datasets), edge regression (four datasets), and node classification (three datasets). Results for edge regression and node classification are reported in Appendix H.2 and Appendix H.3, respectively. In the main body, we show that the theoretical guarantees of SSNs (Sections 3–4) translate into measurable gains on brain-stimulus classification, a task for which we provide a principled justification for using SSNs and introduce a dedicated framework (Section 4).

Our brain network experiments test two hypotheses: (i) modeling *directed higher-order structure and interactions* increases expressivity, enabling SSNs to distinguish stimuli that collapse under graph-based or undirected TDL models; and (ii) SSNs extend neurotopology pipelines by reliably inferring stimulus identity from *arbitrary topological samples*, robust to structural variability and permutations. We evaluate these hypotheses in two settings of increasing difficulty: Task 5.1, feature classification with fixed topology and varying dynamics; and Task 5.2, graph/complex classification where both topology and dynamics vary. This section supports Contribution **C3**.

**Data Overview.** We build on the well-established NMC-model (Markram et al., 2015; Markram, 2006), a biologically detailed microcircuit of the somatosensory cortex in a two-week-old rat with simulated responses to external stimuli. The dataset consists of a large structural digraph $\mathcal{G} = (V, E)$ with $|V| = 31{,}346$ neurons and $|E| = 7{,}803{,}528$ synapses, together with 4495 binary dynamics functions. Following Conceição et al. (2022), neuronal activity is discretized into $T = 2$ time bins, yielding $\{B_i : V \to \mathbb{B}^2\}_{i=1}^{4495}$, where each $B_i$ assigns to every neuron $v \in V$ a two-dimensional binary vector encoding its activation response to one stimulus. Each stimulus corresponds to a uniformly random sample from eight distinct thalamocortical input patterns. Table 2 highlights the abundance of higher-order directed motifs, and further dataset details are provided in App. G.1.

| Type | Count |
|---|---|
| Neurons | 31,346 |
| Edges | 7,803,528 |
| Triangles | 76,936,601 |
| Tetrahedra | 65,939,554 |
| Pentachorons | 7,637,507 |

Table 2: Counts of directed simplices in the Microcircuit complex $\mathcal{K}_{\mathcal{G}}$.

**Experimental Setup.** We evaluate SSNs and their routing-based variant, R-SSNs (top-$k = 2$ relations, except $(4, 125\mu\text{m})$ where $k = 4$), against a comprehensive set of set-, graph-, and topology-based baselines, as recommended in the recent position paper (Bechler-Speicher et al., 2025). Specifically, we compare with: (i) Deep Sets (DS) (Zaheer et al., 2017); (ii) message-passing GNNs (Gilmer et al., 2017; Hamilton et al., 2017); (iii) Directed GNNs (Dir-GNNs) (Rossi et al., 2024); and (iv) message-passing Simplicial Neural Networks (MPSNNs) (Bodnar et al., 2021b). This selection isolates the contributions of handling relational, directional, and higher-order structure jointly. We also benchmark against the topological featurization pipeline of Conceição et al. (2022), which computes invariants followed by an SVM classifier (see App.G.3). To ensure fairness, we match parameter budgets across baselines by scaling hidden dimensions: DS-256, GNN-256, Dir-GNN-256, and the undirected higher-order baseline MPSNN-64. For SSNs, R-SSNs, and MPSNNs (both standard and 64-dim variants), we include simplices up to dimension two (triangles). SSNs use standard boundary/coboundary relations intra-dimensionally, combined with directed up/down adjacencies inter-dimensionally (see App. C.6). Hyperparameters for all models are tuned via grid search. SSNs and R-SSNs are instantiated as message-passing SSNs, as are all graph- and higher-order baselines, using SAGE-style (Hamilton et al., 2017) message functions. Additional results with attention-based baselines (Veličković et al., 2018) are reported in App. H.1.2.

### 5.1 TASK 1: CLASSIFYING DYNAMICAL BRAIN ACTIVITY IN FIXED VOLUMETRIC SAMPLES

Given a fixed brain structure and time-evolving neuronal activation patterns, the goal is to classify the stimulus that triggered the observed dynamics. Following Reimann et al. (2022), we extract volumetric samples $U \subset V$ defined by a neuronal population centroid $c \in \{4, 8\}$ and sampling radius $r \in \{125\mu\text{m}, 175\mu\text{m}, 325\mu\text{m}\}$. Each sample induces a subgraph $\mathcal{G}_U = \mathcal{G}[U]$, and we restrict the binary dynamics $B_i|_U$, lifting them as in equation 3 to obtain $4{,}495$ lifted dynamics $\tilde{B}_i|_U$ per sample. The task is then to classify each lifted dynamic by its associated stimulus identity.

| Model | $(4, 125\mu m)$ | $(4, 325\mu m)$ | $(8, 175\mu m)$ | M = 1 | M = 3 | M = 5 | # Par. | Par. (%) |
|---|---|---|---|---|---|---|---|---|
| TopoFeat+SVM | $42.14 \pm 1.19$ | $35.91 \pm 3.36$ | $45.32 \pm 1.68$ | $27.94 \pm 0.94$ | $27.87 \pm 0.89$ | $28.86 \pm 0.42$ | 312 | 0.3% |
| DS | $26.63 \pm 0.10$ | $19.47 \pm 1.16$ | $27.31 \pm 0.34$ | $23.28 \pm 0.48$ | $24.29 \pm 0.38$ | $25.09 \pm 0.14$ | 1,680 | 2% |
| DS-256 | $25.21 \pm 1.11$ | $19.52 \pm 1.44$ | $25.12 \pm 2.12$ | $25.24 \pm 0.32$ | $24.76 \pm 0.33$ | $25.87 \pm 0.19$ | 70,672 | 68% |
| GNN | $26.00 \pm 1.10$ | $21.56 \pm 1.14$ | $34.02 \pm 4.35$ | $25.43 \pm 0.43$ | $26.03 \pm 0.69$ | $26.49 \pm 1.02$ | 5,392 | 5% |
| GNN-256 | $24.70 \pm 1.31$ | $23.02 \pm 2.1$ | $33.47 \pm 4.15$ | $24.40 \pm 0.51$ | $27.60 \pm 0.91$ | $28.27 \pm 0.17$ | 70,672 | 68% |
| DirGNN | $36.71 \pm 2.00$ | $48.11 \pm 1.87$ | $53.72 \pm 2.89$ | $25.21 \pm 0.18$ | $31.08 \pm 1.04$ | $33.00 \pm 3.69$ | 9,744 | 9% |
| DirGNN-256 | $50.89 \pm 13.00$ | $60.02 \pm 0.75$ | $63.52 \pm 0.70$ | $25.43 \pm 0.54$ | $35.21 \pm 0.83$ | $39.41 \pm 0.27$ | 137,232 | 133% |
| MPSNN | $43.33 \pm 6.95$ | $42.65 \pm 3.71$ | $53.12 \pm 2.65$ | $27.45 \pm 0.50$ | $32.10 \pm 0.77$ | $33.68 \pm 3.04$ | 23,888 | 23% |
| MPSNN-64 | $46.85 \pm 6.71$ | $54.24 \pm 7.06$ | $64.02 \pm 4.91$ | $29.48 \pm 1.31$ | $34.91 \pm 0.73$ | $42.23 \pm 1.82$ | 90,768 | 88% |
| **R-SSN (Ours)** | $57.32 \pm 5.25$ | $79.64 \pm 1.84$ | $70.66 \pm 1.56$ | $28.68 \pm 0.79$ | $40.29 \pm 1.12$ | $48.20 \pm 0.70$ | 18,084 | 18% |
| **SSN (Ours)** | $75.13 \pm 1.28$ | $87.16 \pm 1.36$ | $78.32 \pm 7.03$ | $46.73 \pm 1.16$ | $61.35 \pm 1.07$ | $64.72 \pm 0.54$ | 103,184 | 100% |
| **Gain over 2-nd Best** | ↑ 24.24 % | ↑ 27.14 % | ↑ 14.30 % | ↑ 17.25 % | ↑ 26.14 % | ↑ 22.49 % | - | - |

Table 3: 8-class stimulus classification accuracy (%, higher is better) based on dynamic binary brain activation responses corresponding to one of eight distinct thalamic input patterns. Columns $(4, 125\mu m)$, $(4, 325\mu m)$, and $(8, 175\mu m)$ correspond to fixed volumetric brain samples (i.e., fixed topology, see Section 5.1). Columns $M = 1$, $M = 3$, and $M = 5$ correspond to the number of neuron neighborhoods $M$ sampled within the $(4, 325\mu m)$ volumetric sample (i.e., varying topology, see Section 5.2). The top $1^{st}$, $2^{nd}$, and $3^{rd}$ best results are highlighted. Active parameter counts (#Par.) and relative active parameter percentages (Par. %) at inference are reported with respect to SSNs.

We focus on three volumetric samples, $(c, r) \in \{(4, 125\mu m), (8, 175\mu m), (4, 325\mu m)\}$. These correspond to the most and least discriminative volumes under traditional dimensionality-reduction pipelines (Reimann et al., 2022), along with an additional case to test robustness to volumetric variability. All models employ a permutation-invariant mean readout for compatibility with Task 2, where structural variability is explicit. In Table 3 (columns $(4, 125\mu m)$, $(4, 325\mu m)$, and $(8, 175\mu m)$), we report mean accuracy and standard deviation over five splits (60% train, 20% validation, 20% test). Additional robustness analyses and attention-based experiments are provided in Appendix H.1.2.

## 5.2 Task 2: Classifying Neuron Neighbourhood Dynamical Activity Complexes

We also consider a more challenging setting in which heterogeneous neuron samples and their synaptic connections induce *topological variability*, moving from feature classification (Task 5.1) to graph/complex classification. Each neuron $n$ in $\mathcal{G}$ defines a neighborhood subgraph $\mathcal{G}_n = \mathcal{G}[\mathcal{N}(n)]$. For each binary dynamic, we sample $M$ neuron-neighborhood subgraphs uniformly at random without replacement from the 25 largest within a given volumetric sample $U$, following Reimann et al. (2022). Restricting each dynamic $B_i$ to its $M$ sampled subgraphs and lifting via equation 3 yields $M \times 4{,}495$ DACs.

We use the $(4, 325\mu m)$ component with $M \in \{1, 3, 5\}$ to test two hypotheses: (i) whether SSNs can robustly infer stimulus identity from *arbitrary neighborhoods*, and (ii) whether they exhibit strong *sample efficiency*, since smaller $M$ yields fewer training examples. Table 3 (columns $M = 1, 3, 5$) reports mean accuracy and standard deviation across five splits (60% train, 20% validation, 20% test). Robustness to volumetric and temporal resolution is discussed in Appendix H.1.4.

## 5.3 Discussion

Results in Table 3 show that SSNs accurately classify stimulus identity in fixed brain volumes (Task 5.1) and remain effective under arbitrary neighborhood topologies (Task 5.2), where structural variability is introduced. Their gains stem from jointly modeling higher-order directed motifs, prevalent at multiple scales in brain networks (Sizemore et al., 2018; Tadić et al., 2019; Sizemore et al., 2019; Andjelković et al., 2020), and direction-aware interactions, both of which are critical for capturing neural activity and function (Nolte et al., 2019; Reimann et al., 2017; Wang et al., 2010; Ecker et al., 2024; Reimann et al., 2022). These empirical findings mirror our theoretical results (Sections 3.1 and 4), which establish that SSNs surpass Dir-GNNs and MPSNNs in WL-expressivity (Theorems 1–2) and recover a broader class of (neuro)topological invariants (Theorem 4).

The baselines further highlight the limitations of existing approaches. Standard GNNs and DSs perform worst, reflecting their inability to capture higher-order and directional dependencies. SVMs on topological features perform better, confirming the importance of higher-order directed connectivity; however, they remain constrained by hand-crafted preprocessing and predefined invariants, suggesting that richer representations can be learned directly from data. Dir-GNNs and MPSNNs, which model directionality and hierarchy in isolation, also underperform relative to SSNs, indicating that either aspect alone is insufficient.

In the most challenging regime ($M = 1$), with structural variability and extreme data scarcity (one neuron neighborhood per dynamic), SSNs outperform all baselines by at least 17%, underscoring the value of strong inductive biases for sample efficiency (Bronstein et al., 2021). Importantly, SSNs achieve these gains with parameter counts comparable to or smaller than the baselines (Table 3, columns #Par. and Par.%). R-SSNs consistently rank second or third across all settings while using substantially fewer parameters, enabling faster training and inference. Detailed complexity and runtime analyses are provided in Appendix F and Appendix H.

## 6 Conclusion

We introduced Semi-Simplicial Neural Networks (SSNs), a new class of topological deep learning (TDL) architectures for data structured as semi-simplicial sets. SSNs generalize graph-based and existing TDL models by propagating information over simplices via face-map–induced relations that can be hierarchical, directed, and asymmetric. They subsume prior architectures and provide strictly stronger Weisfeiler–Leman (WL) expressivity.

By capturing a broader class of topological invariants of neural activity, SSNs establish a principled link between neurotopology and deep learning. Experiments on a biologically realistic neocortical model support these theoretical advantages in practice: SSNs consistently outperform state-of-the-art baselines on stimulus classification. Limitations and directions for future work are discussed in App. A. Overall, our theoretical and methodological contributions, together with an open-source codebase, lay the groundwork for scalable directed higher-order representation learning and suggest SSNs as a promising foundation for TDL models in complex scientific and real-world domains, including neuroscience.

## 7 Reproducibility Statement

We provide all information needed to reproduce our results. For the primary neural binary dynamics classification tasks, we describe the experimental setup in Section 5 and report full hyperparameter configurations in Appendix H.1. We also provide detailed protocols and hyperparameters for the additional baselines and ablations, including the topological-features baseline and attention-based variants.

For the Edge Traffic Regression and Node Classification tasks, we report the experimental protocols and hyperparameter settings in Appendix H.2 and Appendix H.3, respectively. Hardware specifications and computational resource requirements are documented in Appendix E.3.

Our code is available at https://github.com/ManuelLecha/ssn.

Data is available at https://zenodo.org/records/17700425.

## 8 Acknowledgements

The work of EI is supported by the TU Delft AI Labs programme, the NWO OTP GraSPA proposal #19497, and the NWO VENI proposal 222.032. The work of AC is supported by the NWO OTP GraSPA proposal #19497. The work of RL is partially supported by an EPSRC grant EP/Y028872/1. The work of FD and CB is supported by the National Institutes of Health Grant 1R01ES037156-01.

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

APPENDIX CONTENTS

## A  LIMITATIONS & FUTURE DIRECTIONS

In settings where data does not faithfully reflect inherent directionality—either because the inductive biases from topology and connectivity misalign with actual information flows, or because the system itself lacks directional structure—SSNs may introduce unnecessary memory overhead due to reduced parameter sharing across semantically distinct relations, without yielding significant accuracy gains. Nevertheless, as shown in our experiments, performance remains competitive and is not adversely affected. This limitation can be mitigated in two ways: (i) via our proposed routing mechanism, which adaptively selects the top-$k$ most informative relations during learning; and (ii) by investigating directed topological latent inference to recover meaningful directional structure. From a neurotopological perspective, these techniques cannot yet be applied *in vivo*, as they require access to detailed connectomic data—that is, structural information about the brain's wiring. However, this limitation is expected to diminish with the increasing availability of densely sampled electron microscopy connectomes co-registered with neural activity data. More significantly, the emergence of generative pipelines for constructing biologically realistic simulated digital brains offers a transformative alternative. These developments open the door to novel applications, including the design of intelligent systems inspired by biologically grounded architectures (Markram et al., 2024;

2025), where our model enables the learning of representations that capture high-level information abstractions within such systems. Beyond our current scope, several promising directions remain open, including the unsupervised discovery of neural activity motifs and the extension of our framework to broader neuroscience tasks. Alternative lifting strategies—such as tournaplexes and other directed combinatorial structures—are largely unexplored and may provide complementary inductive biases. Incorporating temporal dynamics and signed interactions (e.g., excitatory vs. inhibitory) (Millán et al., 2025) is another key avenue. We emphasize that our contribution is primarily theoretical and methodological: it establishes a connection between TDL and higher-order directed complex systems such as the brain. We do not foresee any potential for harm or misuse of this technology if applied responsibly.

## B  RELATED WORKS

**Topological Deep Learning.** TDL builds directly on pioneering efforts in Topological Signal Processing (TSP) (Barbarossa & Sardellitti, 2020; Schaub et al., 2021; Roddenberry et al., 2022; Sardellitti et al., 2021; Isufi et al., 2025; Battiloro, 2024), which underscore the importance of modeling multi-way relationships. Extensions of the WL test to simplicial and regular cell complexes (Bodnar et al., 2021b;a) have demonstrated that message passing on these higher-order structures outperforms its graph-based counterpart. Both convolutional (Ebli et al., 2020; Yang et al., 2022; Hajij et al., 2020; Yang & Isufi, 2023; Roddenberry et al., 2021; Hajij et al., 2022a; Yan & Kuruoglu, 2025) and attentional (Battiloro et al., 2024b; Giusti et al., 2022; Goh et al., 2022; Giusti et al., 2023a) architectures over simplicial and cell complexes have been introduced. Furthermore, message passing and diffusion on cellular sheaves over graphs (Hansen & Ghrist, 2019; Hansen & Gebhart, 2020; Bodnar et al., 2022; Battiloro et al., 2024c; Barbero et al., 2022) have proven particularly effective in heterophilic settings. Alternatives that eschew message passing altogether for simplicial complexes have been detailed in (Madhu et al.; Gurugubelli & Chepuri, 2024; Maggs et al., 2024), while an approach to infer a latent regular cell complex for downstream improvement has been introduced in (Battiloro et al., 2024a). Gaussian Processes on simplicial and cell complexes have been presented in (Yang et al., 2024; Alain et al., 2024). Comprehensive reviews of TDL can be found in (Papillon et al., 2023; 2025). A combinatorial framework for TDL—combinatorial complexes (CCs)—was proposed in (Hajij et al., 2022b). CCs relax the structural constraints of classical topological domains by treating cells as ranked, unordered subsets of vertices. This generality makes CCs broadly applicable but prevents them from distinguishing vertex orderings or encoding coherent face maps. As a consequence, CCs cannot represent directional or order-sensitive higher-order interactions, and structures such as directed simplicial complexes, ordered motifs, or tournaplexes fall outside their scope. Semi-simplicial sets (SSSs) provide exactly this missing structure: they support multiple ordered simplices on the same vertex set together with coherent face maps satisfying the simplicial identities, enabling principled modeling of directionality and order-aware higher-order interactions. These structural differences imply that CCs and SSSs capture different domains of objects, and neither framework subsumes the other. An extension of the CC framework was recently proposed in (Hajij et al., 2025). A fairly general topological expressivity analysis of TDL models is given in (Eitan et al., 2025). Quantum simplicial neural networks have been introduced in (Piperno et al., 2025), continuous variants in (Einizade et al., 2025), and geometrically equivariant architectures over simplicial and broader combinatorial spaces in (Eijkelboom et al., 2023; Battiloro et al., 2025b). Finally, software libraries and benchmarks for TDL have been released in (Hajij et al., 2024; Telyatnikov et al., 2025; Papillon et al., 2024; Ballester et al., 2024).

**Directed Graph and Topological Models.** A growing body of work has incorporated directionality into neural architectures to address the inherent asymmetry of relational data in complex systems. Early spatial methods distinguish incoming and outgoing messages via separate learnable weights, explicitly modeling node roles (Li et al., 2016; Abbass, 2018). Directed GNNs for node classification (Rossi et al., 2024; Tong et al., 2020b), for instance, have demonstrated superior performance on heterophilic benchmarks (Rossi et al., 2024; Koke & Cremers, 2023). Spectral approaches extend Laplacian-based learning to directed graphs: DiGCN uses a Personalized PageRank–based operator (Tong et al., 2020a), while MAGNet introduces the Magnetic Laplacian, encoding edge directionality in the complex phase (Zhang et al., 2021). This formulation was later extended to the Sign-Magnetic Laplacian, addressing key limitations and enabling principled learning on signed directed graphs (Fiorini et al., 2023). Extending this works, Fuchsgruber et al. (2025) propose the Magnetic Edge Laplacian for edge-level regression tasks in traffic forecasting. Notably, they

introduce both an orientation-equivariant variant—designed for direction-sensitive signals such as traffic flow—and an orientation-invariant variant—suited for direction-agnostic signals such as speed limits—thereby generalizing previous edge-based methods. Beyond pairwise structures, recent works extend to non-uniform, non-hierarchical settings using neural networks on directed hypergraphs (Fiorini et al., 2024; Ma et al., 2024), with applications in traffic forecasting (Luo et al., 2022) and human pose estimation (Cui et al., 2024). Bernárdez et al. (2025) further show that hierarchical, order-aware relations—built on the combinatorial framework of Hajij et al. (2022b)—better capture communication patterns in computer network systems. Finally, the preliminary work of Lecha et al. (2025) introduces Directed Simplicial Neural Networks, formalizing higher-order topological directionality for learning on directed simplicial complexes based on the theoretical foundations of Riihimäki (2023).

**Neurotopology.** The manifold hypothesis posits that neuronal activity is constrained to a lower-dimensional subspace, shaped by the structural limitations imposed by synaptic connectivity (Cunningham & Yu, 2014; Gallego et al., 2017; Curto & Morrison, 2019; Chambers & MacLean, 2016; Rubinov & Sporns, 2010; Bargmann & Marder, 2013; Bassett & Sporns, 2017). This concept builds on the foundational principle that structure shapes function, succinctly captured by Hebb's rule: "neurons that fire together, wire together." Yet, precisely how structural connectivity gives rise to neural manifolds remains a major open question. Biologically grounded computational models, such as the Blue Brain's (Markram, 2006) rat neocortical microcircuit (NMC) (Markram et al., 2015), offer a valuable platform for investigating this relationship, providing a detailed representation of brain structure modeled as a directed graph. Notably, recent work of Reimann et al. (2022) confirmed that the NMC model adheres to the manifold hypothesis. However, while directed graphs effectively capture the inherent directionality of synaptic transmission—from presynaptic to postsynaptic neurons (Reimann et al., 2017)—their dyadic nature limits their capacity to represent polyadic interactions and higher-order co-activation patterns that are critical to neural computation. Experimental studies have revealed that neuronal connectivity exhibits significant non-random higher-order structure. For instance, it becomes increasingly unlikely for random activity to produce coherent patterns on higher-dimensional cliques, suggesting that such patterns reflect greater functional complexity and information abstraction. Simplices—higher-dimensional analogs of cliques—have been identified as overexpressed motifs across multiple scales of brain networks (Sizemore et al., 2018; Tadić et al., 2019; Sizemore et al., 2019; Andjelković et al., 2020), and have been linked to functional relevance (Nolte et al., 2019). Alternatively, co-fluctuation frameworks such as (Santoro et al., 2024) derive meaningful weighted, undirected simplicial complexes by algebraically combining low-order fMRI signals to construct higher-order time series. Remarkably, human connectomes have been shown to contain undirected simplices of up to 16 and even 20 dimensions, despite comprising only 1, 115 nodes representing brain regions (Tadić et al., 2019). Crucially, neurons embedded in higher-order directed simplices with correlated activity exhibit reduced sensitivity to noise and stronger alignment with the underlying neural manifold (Cunningham & Yu, 2014; Gallego et al., 2017), ultimately improving representational fidelity (Nolte et al., 2019; Reimann et al., 2017; Wang et al., 2010). Recent findings also suggest that neural circuits balance robustness and efficiency by organizing into subpopulations with distinct topologies: low-complexity simplices promote efficient communication, while high-complexity ones support resilience—both coexisting within the same network (Ecker et al., 2024). These insights have fueled the rise of Neurotopology, a field modeling brain activity through topological spaces derived from spike co-activation patterns. Seminal works (Conceição et al., 2022; Reimann et al., 2017) propose modeling neural dynamics as time-indexed sequences of directed flag complexes (Lütgehetmann et al., 2020) and tournaplexes (Govc et al., 2021), derived from subdigraphs induced by active neurons. Brain dynamics are then characterized by computing topological invariants over these evolving structures. Building on this framework, Reimann et al. (2022) demonstrated that stimulus identity can be accurately decoded from the NMC model using this topological features—even when conventional methods fail. These results highlight topological featurization as a robust and complementary alternative to traditional manifold-based approaches.

# C  PRELIMINARIES

## C.1  ALGEBRAIC BACKGROUND

**Groups.** A *group* is a pair $(G, \cdot)$, consisting of a set $G$ and a binary operation $\cdot$ satisfying the following axioms: **(A1)** *Associativity:* for all $a, b, c \in G$ we have $a \cdot (b \cdot c) = (a \cdot b) \cdot c$; **(A2)** *Identity:* there exists an element $e \in G$ such that $ea = ae = a$ for all $a \in G$; **(A3)** *Inverse:* for each $a \in G$ there exists an element $a^{-1} \in G$ such that $a^{-1}a = aa^{-1} = e$. Given a set $S$, the set of all bijections $\rho : S \to S$ together with the composition of functions forms a group known as the *symmetric group of $S$*, denoted by $\mathrm{Sym}(S)$.

**Isomorphisms.** Let $X$ and $Y$ be two structured objects of the same type. An *isomorphism* $\psi : X \to Y$ is a bijective map with inverse $\psi^{-1} : Y \to X$ that preserves all relevant structure. If such a map exists, we say that $X$ and $Y$ are *isomorphic*, denoted $X \cong Y$. Two sets $S$ and $S'$ are isomorphic if there exists a bijection between them. Two directed graphs $\mathcal{G} = (V, E)$ and $\mathcal{G}' = (V', E')$ are isomorphic if there exists a bijection $\psi : V \to V'$ such that $(u, v) \in E$ if and only if $(\psi(u), \psi(v)) \in E'$. If the vertex sets $V$ and $V'$ are attributed—e.g., equipped with dynamic binary functions $B : V \to \{0, 1\}^T$ and $B' : V' \to \{0, 1\}^T$—we additionally require that the attributes are preserved under the inverse map: for all $v \in V$, $B(v) = B'(\psi^{-1}(v))$. In general for any two sets equipped with a collection of relations if set isomorphism preserves relation. Specifically, for directed simplicial complexes $\mathcal{K}$ and $\mathcal{K}'$, an isomorphism is a bijection $\psi$ on the vertex set that preserves simplices: a tuple $(v_0, \ldots, v_n) \in \Sigma_n$ is a simplex in $\mathcal{K}$ if and only if $(\psi(v_0), \ldots, \psi(v_n)) \in \Sigma'_n$ is a simplex in $\mathcal{K}'$.

**Automorphisms.** An *automorphism* of a structured object $\mathcal{S}$ is an isomorphism from $\mathcal{S}$ onto itself. The set of all automorphisms of $\mathcal{S}$, denoted $\mathrm{Aut}(\mathcal{S})$, forms a group under composition. Indeed, composition is associative, the identity map serves as the identity element, and each automorphism is invertible. If $S$ is a plain set, then any bijection (permutation) $\rho : S \to S$ is an automorphism, so $\mathrm{Aut}(S) = \mathrm{Sym}(S)$ is the full symmetric group on $S$.

## C.2  RELATIONAL PRELIMINARIES

**Binary Relations.** Let $S$ be a set. A *binary relation* $R$ on $S$ is any subset $R \subseteq S \times S$. A binary relation is *symmetric* if, for all $\sigma, \tau \in S$, $(\sigma, \tau) \in R$ implies $(\tau, \sigma) \in R$; *transitive* if, for all $\sigma, \tau, \kappa \in S$, $(\sigma, \kappa) \in R$ and $(\kappa, \tau) \in R$ together imply $(\sigma, \tau) \in R$; and *irreflexive* if $(\sigma, \sigma) \notin R$ for every $\sigma \in S$. We further say that $R$ is *$n$-uniform* if every element of $S$ is related to exactly $n$ other elements. Since relations are subsets of $S \times S$, canonical examples include the *universal relation* $1 = S \times S$, the *empty relation* $0 = \emptyset$, and the *identity relation* $\mathrm{id} = \{(\sigma, \sigma) \mid \sigma \in S\}$.

**Basic Operations.** Standard set operations extend naturally to relational operations:

$$R \cap R' = \{(\sigma, \tau) \in S \times S \mid (\sigma, \tau) \in R \text{ and } (\sigma, \tau) \in R'\},$$
$$R \cup R' = \{(\sigma, \tau) \in S \times S \mid (\sigma, \tau) \in R \text{ or } (\sigma, \tau) \in R'\}.$$

Furthermore, relational composition and converse operations are defined as:

$$R \circ R' := \{(\sigma, \tau) \in S \times S \mid \exists \kappa \in S \text{ such that } (\sigma, \kappa) \in R \text{ and } (\kappa, \tau) \in R'\},$$
$$R^\top := \{(\sigma, \tau) \in S \times S \mid (\tau, \sigma) \in R\}.$$

**Characteristic function.** Analogous to how subsets can be identified with their indicator functions, a relation $R \subseteq S \times S$ can be uniquely represented by its characteristic function $\chi_R : S \times S \to \{0, 1\}$. This function assigns a binary value to each ordered pair $(\sigma, \tau) \in S \times S$: it returns 1 if $(\sigma, \tau) \in R$ and 0 otherwise:

$$\chi_R((\sigma, \tau)) = \begin{cases} 1, & \text{if } (\sigma, \tau) \in R, \\ 0, & \text{if } (\sigma, \tau) \notin R. \end{cases}$$

**Matricial Representation.** Equipping the set $S$ with an ordering $S = \{\sigma_1, \ldots, \sigma_n\}$, the matricial representation of $R$ is the $n \times n$ matrix $(A_R)^i_j = \chi_R((\sigma_i, \sigma_j))$. We call it relation matrix or adjacency matrix. Such matrix representations underpin computational implementations, relational learning frameworks, and message-passing neural network architectures (Veličković, 2022).

**$n$-ary Relations.** An *$n$-ary relation* over a set $S$ is a subset $R \subseteq S^n$, that is, a collection of $n$-tuples drawn from $S$. For instance, ternary relations $R \subseteq S \times S \times S$ arise by extending binary relations via the natural join operation:

$$R \bowtie R' = \{ (\sigma, \kappa, \tau) \in S \times S \times S \mid (\sigma, \kappa) \in R \text{ and } (\kappa, \tau) \in R' \}.$$

This operation yields a ternary relation by joining the tuples from $R$ and $R'$ on their shared middle element $y$, thus composing relational information across three elements of the set $S$.

### C.3 Directionality and Symmetry in Relational and Combinatorial Topological Structures

**Directionality.** A relation of arity $n + 1$ over a vertex set $V$ is intrinsically *directional* when it is stored as an ordered tuple $(v_0, \ldots, v_n)$ and membership depends on the order of its entries. Such a relation is *symmetric* if membership is invariant under permutations: for every $(\sigma_0, \ldots, \sigma_n) \in R$ and every permutation $\rho \in \mathrm{Sym}(n+1)$, the tuple $(\sigma_{\rho(0)}, \ldots, \sigma_{\rho(n)})$ also belongs to $R$. Otherwise, the relation is *asymmetric*. Binary relations provide the most familiar case. An asymmetric binary relation $R \subseteq V \times V$—where $(\sigma, \tau) \in R$ does not necessarily entail $(\tau, \sigma) \in R$—is naturally represented by a directed graph $\mathcal{G} = (V, E)$ with $R = E$. In contrast, undirected graphs correspond to symmetric binary relations: the presence of $(\sigma, \tau)$ guarantees $(\tau, \sigma)$, so one may equivalently replace ordered pairs with unordered pairs $\{\sigma, \tau\}$ without loss of topological information. This viewpoint extends seamlessly to higher-order structures. A directed simplicial complex $\mathcal{K}$ can be regarded as an asymmetric relational structure over $V$, equipped with families of asymmetric relations $\Sigma_n \subseteq V^{n+1}$ of varying arities, which collect directed simplices of different dimensions. Thus, the asymmetric–symmetric distinction provides a unifying language for characterizing directed versus undirected combinatorial objects.

**Symmetrization.** Given an irreflexive relation $R$ on a set $S$ (equivalently, a simple directed graph $\mathcal{G} = (V, E)$), its *symmetrization* is defined as $R_{\mathrm{sym}} = R \cup R^T$, or in graph-theoretic terms, $E_{\mathrm{sym}} = E \cup E^T$. This construction associates an undirected graph $\mathcal{G}_{\mathrm{sym}}$ with the original directed graph via the canonical projection (or quotient map)

$$\pi((u, v)) = \{u, v\}, \quad (u, v) \in E_{\mathrm{sym}}.$$

There is a one-to-one correspondence between directed symmetric graphs and undirected graphs under this map. The same principle extends naturally to directed simplicial complexes. Given a directed simplicial complex $\mathcal{K}$—with directed graphs as its 1-skeleton—we define its *symmetrization map* $\pi : \mathcal{K} \to \mathcal{K}_{\mathrm{sym}}$ by sending each $n$-simplex $(v_0, \ldots, v_n)$ to the unordered set $\{v_0, \ldots, v_n\}$. While this map preserves the simplicial structure (i.e., maps directed simplicial complexes to valid abstract simplicial complexes), it discards ordering information by identifying every ordered simplex with all of its permutations. This loss of orientation leads to a reduction in expressive power, which we make precise in the following proposition.

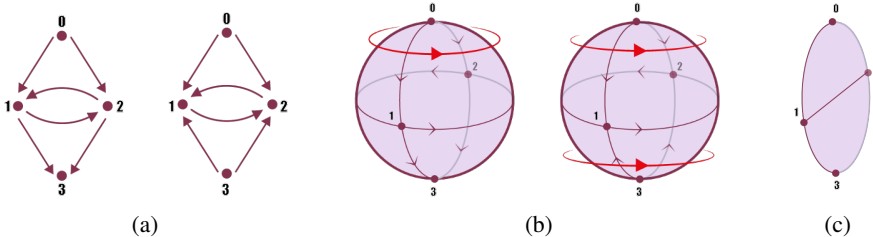

(a)               (b)               (c)

Figure 5: (a) A pair of non-isomorphic digraphs $\mathcal{G}$ and $\mathcal{G}'$, (b) their corresponding non-isomorphic directed flag complexes $\mathcal{K}_{\mathcal{G}}$ and $\mathcal{K}_{\mathcal{G}'}$, respectively, and (c) the resulting symmetrized simplicial complex $\mathcal{K}_{\mathrm{sym}}$, which is isomorphic in both cases, i.e., $\pi(\mathcal{K}_{\mathcal{G}}) \cong \pi(\mathcal{K}_{\mathcal{G}'})$.

**Proposition 2.** *The symmetrization map $\pi : \mathcal{K} \to \mathcal{K}_{\mathrm{sym}}$ is a simplicial structure* forgetful map*, that is, preserves isomorphisms but is not injective on isomorphism classes.*

*Proof.* Let $\psi$ be an isomorphism between $\mathcal{K}$ and $\mathcal{K}'$ and let $\psi_{\mathrm{sym}}(\sigma) = \{\psi(v) : v \in \sigma\}$. Note that, $\psi_{\mathrm{sym}}$ is well defined since different orderings of the same vertex set in $\sigma$ are mapped to the same set

$\{\psi(v) : v \in \sigma\}$. Moreover, by construction $\pi \circ \psi = \psi_{\text{sym}} \circ \pi$, implying preservation of simplicial structure (simplices are sent to simplices). Since $\psi$ is bijective and $\pi$ does not collapse vertices (only their ordering), it follows that $\psi_{\text{sym}}$ is a bijective simplicial map preserving the face structure. Hence, $\psi_{\text{sym}}$ is an isomorphism between $\mathcal{K}_{\text{sym}}$ and $\mathcal{K}'_{\text{sym}}$, and $\pi$ is *isomorphims-preserving*. Clearly, there exist non-isomorphic directed simplicial complexes $\mathcal{K}$ and $\mathcal{K}'$ such that $\pi(\mathcal{K}) \cong \pi(\mathcal{K}')$, implying the *non-injectivity*. Finally, note that the number of simplices in $\mathcal{K}_{\text{sym}}$ equals that in $\mathcal{K}$ if and only if no two distinct simplices in $\mathcal{K}$ share the same underlying vertex set—that is, each simplex has a unique ordering. Otherwise, $\pi$ collapses the redundant orderings, reducing the total face count (see Figure 5). □

Symmetrization naturally extends to attributed directed simplicial complexes via a fixed permutation-invariant aggregation of features.

**Transitivization.** Conversely, given a simplicial complex $\mathcal{K} = (V, \Sigma)$ endowed with an arbitrary total order $<$ on its vertex set $V$, one naturally obtains a unique directed simplicial complex $\mathcal{K}_{\text{dir}}$ by assigning to each simplex $\sigma \in \Sigma$ the directed simplex $\sigma_< = (v_0, v_1, \ldots, v_n)$, where the vertices of $\sigma$ are arranged in increasing order ($v_0 < v_1 < \cdots < v_n$). Consequently, for every pair of vertices $v_i, v_j \in \sigma$ with $i < j$, the directed edge $(v_i, v_j)$ is present in $\mathcal{K}_{\text{dir}}$. By construction, every simplex thus forms a transitive clique at the graph level.

## C.4 Direction vs. Orientation in Combinatorial Topological Structures

We clarify the conceptual distinction between *directionality* and *orientation*, two notions that are often conflated.

**Directionality.** As defined in Section C.3, a relation of arity $n + 1$ over a vertex set $V$ is *directional* when it is stored as an ordered tuple $(v_0, \ldots, v_n)$ and membership depends on the order. For instance, in a directed graph the edge $(0, 1)$ does not imply $(1, 0)$. Reversing an edge $(0, 1)$ vs. $(1, 0)$ or permuting a triangle $(0, 1, 2)$ vs. $(2, 1, 0)$ yields a different simplex that may encode different information. This property enables models to capture higher-order asymmetric motifs—such as brain co-activations (Sec. 4), traffic flows (App. H.2), or citation graphs (App. H.3). By contrast, symmetrization collapses distinct motifs (e.g., transitive cliques vs. cycles) into a single entity (Sec. C.3, Prop. 2, Fig. 5), erasing essential directional dependencies. Thus, directionality encodes irreducible asymmetry in the data.

**Orientation.** Given an unordered $k$-simplex $\sigma = \{v_0, \ldots, v_k\}$, an *orientation* is a choice of equivalence class of orderings of its vertices under even permutations. Formally, if $\rho \in \text{Alt}(k+1) \subseteq \text{Sym}(k+1)$ is an even permutation, then

$$[v_0, \ldots, v_k] = [v_{\rho(0)}, \ldots, v_{\rho(k)}].$$

The alternating group $\text{Alt}(k+1)$ consists of permutations that can be expressed as an even number of transpositions. Hence each $k$-simplex has exactly two orientations, corresponding to $\pm 1$. Orientation is thus a $\mathbb{Z}_2$-valued structure, while directionality involves the full symmetric group.

*Oriented vs. directed edge.* An oriented 1-simplex $[0, 1]$ assigns a sign: a positive orientation may encode flux from 0 to 1, and a negative orientation flux from 1 to 0, thus usually serving as a reference direction of current flow. Both are the same edge, differing only by sign. A directed edge instead distinguishes $(0, 1)$ and $(1, 0)$ as two distinct simplices, each possibly carrying different information.

*Oriented vs. directed triangle.* The unordered simplex $\{0, 1, 2\}$ admits two orientations: clockwise $(0, 1, 2) \sim (1, 2, 0) \sim (2, 0, 1)$ and counter-clockwise $(0, 2, 1) \sim (2, 1, 0) \sim (1, 0, 2)$. Orientation collapses the $3! = 6$ orderings into two parity classes. A directed 2-simplex, however, distinguishes all six ordered triples, allowing models to treat distinct transitive cliques as fundamentally different.

Orientation imposes a sign convention on an undirected simplex, while directionality generates genuinely distinct simplices on the same vertex set. As a consequence, models that ignore asymmetry cannot distinguish higher-order motifs, and their adjacency is restricted to set inclusion, preventing direction-aware propagation of information.

## C.5 FACE-MAP RELATION ALGEBRAS

Let $\mathcal{S}$ be a semi-simplicial set with simplices $S$ and face maps $d_i^n : S_n \to S_{n-1}$. Each face map naturally induces an irreflexive binary relation:

$$R_{d_i^n} = \{(\sigma, \tau) \mid \sigma \in S_n, \ d_i^n(\sigma) = \tau\} \in \mathcal{P}(S \times S),$$

which relates each simplex $\sigma \in S_n$ with its $i$-th facet in $S_{n-1}$.

**Positive Face Map Induced Relation Algebra.** The set of all binary relations on $S$, denoted as $\mathcal{P}(S \times S)$, forms a *positive relation algebra* closed under the operations of union ($\cup$), intersection ($\cap$), relational composition ($\circ$), and converse ($\top$). Specifically, the structure:

$$\mathcal{R} = (\mathcal{P}(S \times S), \cup, \cap, \circ, \top, \mathrm{id}, 0, 1)$$

satisfies standard axioms in (Tarski, 1941). Formally, we define the *face map relation algebra* $\mathcal{R}_d$ as the smallest positive relation subalgebra of $\mathcal{P}(S \times S)$ generated by the collection $\{R_{d_i^n} \mid n \geq 0, , 0 \leq i \leq n\}$, equipped with standard operations $\cup$, $\cap$, $\circ$, $\top$ together with relations $\mathrm{id}, 0, 1$.

For theoretical purposes, we consider ternary relations from natural joins $R \bowtie R'$ from $R, R' \in \mathcal{R}_d$. These higher-order relations play a role in our generalization of Weisfeiler–Leman-type procedures (see Appendix D.2).

## C.6 EXAMPLES OF COMMON GRAPH AND TOPOLOGICAL FACE-MAP RELATIONS

Overall, face-map–induced relations generalize classical binary relations found in graphs, digraphs, and simplicial complexes, thereby providing a principled algebraic foundation for relational message-passing architectures (Veličković, 2022; Rossi et al., 2024; Bodnar et al., 2021b) (see Appendix D.1). In this section, we provide some of this common examples. In particular, we introduce the relation set used in our SSN experiments, combining boundary/co-boundary maps with directed up/down adjacencies for intradimensional communication.

Let $S$ be a semi-simplicial set with set of simplices $S$. We define the following face map–induced binary relations:

$$R_{n\downarrow,i,j} := R_{d_j^n}^\top \circ R_{d_i^n} = \{(\sigma, \tau) \mid d_i^n(\sigma) = d_j^n(\tau)\} \in \mathcal{R}_d, \tag{4}$$

which relates two $n$-simplices $\sigma$ and $\tau$ if their $i$-th and $j$-th facets coincide, respectively (see Figure 6). A natural extension, often employed in message-passing architectures, is the following ternary relation:

$$\tilde{R}_{n\downarrow,i,j} := R_{n\downarrow,i,j} \bowtie R_{d_i^n} = \{(\sigma, \tau, \kappa) \mid d_i^n(\sigma) = d_j^n(\tau) = \kappa\} \in \mathcal{R}_d, \tag{5}$$

which explicitly includes the shared lower-dimensional face $\kappa$ between simplices $\sigma$ and $\tau$. Similarly,

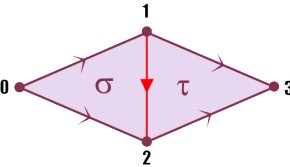

Figure 6: Two 2-simplices (triangles) $\sigma$ and $\tau$ are related such that $d_0^2(\sigma) = d_2^2(\tau) = e$, where $e$ denotes the shared edge highlighted in red. Therefore, $(\sigma, \tau) \in R_{2\downarrow,0,2}$.

we define:

$$R_{n\uparrow,i,j} := R_{d_j^{n+1}} \circ R_{d_i^{n+1}}^\top = \{(\sigma, \tau) \mid \exists \kappa \in S, \ d_i^n(\kappa) = \sigma \text{ and } d_j^n(\kappa) = \tau\} \in \mathcal{R}_d \tag{6}$$

which relates $\sigma$ and $\tau$ if they are, respectively, the $i$-th and $j$-th facets of the same $(n+1)$-simplex $\kappa$ (see Figure 7). Its corresponding ternary extension is defined as:

$$\tilde{R}_{n\uparrow,i,j} := R_{n\uparrow,i,j} \bowtie R_{d_i^{n+1}}^\top = \{(\sigma, \tau, \kappa) \mid \exists \kappa \in S, \ d_i^n(\kappa) = \sigma \text{ and } d_j^n(\kappa) = \tau\} \in \mathcal{R}_d, \tag{7}$$

which explicitly includes the common higher-dimensional simplex $\kappa$. From this point forward, we will use the same notation for both binary and ternary relations when the context is clear, as all constructions are valid in both cases. In Section D.2, unless otherwise specified, relations will be assumed to be in their ternary form. Notably, when $i \neq j$, these relations are asymmetric, thereby

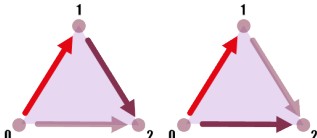

Figure 7: Two 1-simplices (edges) $\sigma$ (red) and $\tau$ (brown) are related as facets of a common $(n+1)$-simplex $\kappa$ (light purple triangle). In (left), $(\sigma, \tau) \in R_{1\uparrow,2,0}$, whereas in (right), $(\sigma, \tau) \in R_{1\uparrow,2,1}$.

inducing directed graphs over the set of simplices $S$. Well-known examples of directional face-map relations include those arising in graph and simplicial structures. For example, at the level of 0-simplices (nodes), the standard directed up adjacency relations can be derived as follows:

$$R_{\text{in}} := R_{0\uparrow,0,1} = \{ (u,v) \mid \exists e \in E,\ d_0^1(e) = u \text{ and } d_1^1(e) = v \}, \in \mathcal{R}_d \tag{8}$$

indicating that $u$ is the target of $v$ (i.e., $v$ is the source of $e$ and $u$ its sink). Moreover,

$$R_{\text{out}} := R_{0\uparrow,1,0} = \{ (u,v) \mid \exists e \in E, d_1^1(e) = u \text{ and } d_0^1(e) = v \} \in \mathcal{R}_d, \tag{9}$$

meaning $u$ is the source of $v$ (i.e., $v$ is the target of $e$, and $u$ the source). These two relations are converses of each other: $(R_{\text{in}})^\top = R_{\text{out}}$, and thus:

$$R_{\text{sym}} = R_{\text{in}} \cup R_{\text{out}} \in \mathcal{R}_d, \tag{10}$$

induces a symmetric directed graph $\mathcal{G}_{\text{sym}} = (V, E_{\text{sym}})$ in one-to-one correspondence with the undirected graph $\mathcal{G}_{\text{sym}} = (V, E)$.

**Directed Paths and Higher Order Directionality.** A directed path of length $k$ in a digraph $\mathcal{G} = (V, E)$ is a sequence of vertices $(v_1, v_2, \ldots, v_k)$ such that $(v_i, v_{i+1}) \in E$ for every $1 \leq i < k$. Similarly, given a set of simplices $\mathcal{S}$ of a semi-simplicial set equipped with a face-map induced relation $R \in \mathcal{R}_d$, an $R$-path of lenght $k$ in the semi-simplicial set is a sequence of simplices $(\sigma_1, \sigma_2, \ldots, \sigma_k)$ such that $(\sigma_i, \sigma_{i+1}) \in R$ for every $1 \leq i < k$. Notably, $(\sigma_1, \sigma_k) \in R^{\circ k}$, where $R^{\circ k}$ denotes the $k$-fold composition of $R$ with itself. Composing different instances of $R_{n\downarrow,i,j}$ yields different higher-order simplicial paths, connecting directed simplices in distinct direction-preserving manner (Riihimäki, 2023; Lecha et al., 2025) (see Figure 8).

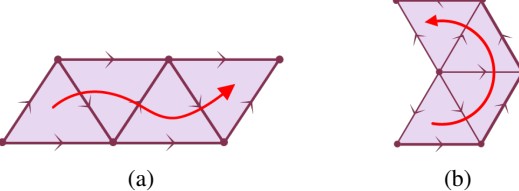

(a)  (b)

Figure 8: (a) An $R_{2\downarrow,0,2}$ directed path (red arrow) in a consistently directed two-dimensional structure with a unique source and sink, and (b) An $R_{2\downarrow,1,2}$ path (red arrow) depicting a circular flow around a source vertex.

**Boundary & Converse Boundary.** Let $\mathcal{K} = (V, \Sigma)$ be a simplicial complex with maximal dimension $N$, and let $\mathcal{K}_{\text{dir}}$ denote the associated directed simplicial complex induced by a fixed total order on $V$. For each dimension $n$ with $1 \leq n \leq N$, the $n$th *boundary relation* is defined as:

$$B_n := \bigcup_{i=0}^{n} R_{d_i^n}, \tag{11}$$

which relates each $n$-simplex to all its $(n-1)$-dimensional faces. Conversely, for $N - 1 \geq n \geq 0$, the $n$-th *converse boundary relation* is defined as

$$C_n := \bigcup_{i=0}^{n+1} (R_{d_i^{n+1}})^\top, \tag{12}$$

relating each $n$-simplex to all $(n+1)$-dimensional simplices containing it as a facet. Notably, for any $0 \leq m < n \leq N$ the composition of maps $B_{n,m} := B_{m+1} \circ B_{m+2} \circ \cdots \circ B_n$, relating an $n$-simplex with its $m$-dimensional faces. Similarly, for the converse boundary relations, define $C_{m,n} := C_{n-1} \circ \cdots C_{m+1} \circ C_m$, relating $n$-simplex with all $m$-simplices containing it.

**Aggregated Lower and Upper Adjacencies.** For each dimension $N \geq n > 0$, we define the collection of all possible $n$-*lower adjacency* relations as

$$\mathcal{R}_{\downarrow n} = \{R_{\downarrow n, i, j} \mid (i, j) \in [n]^2\}. \tag{13}$$

Accordingly, we define

$$R_{\downarrow n} := \bigcup \mathcal{R}_{\downarrow n}, \tag{14}$$

which relates a simplex $\sigma$ to all simplices $\tau$ sharing a common facet with $\sigma$. Then, for dimensions $N - 1 \geq n \geq 0$, we define the collection of all possible $n$-th *upper adjacency* relations by

$$\mathcal{R}_{\uparrow n} = \{R_{\uparrow n, i, j} \mid (i, j) \in [n]^2 \text{ and } i \neq j\}.. \tag{15}$$

Finally, we define the aggregated upper adjacency relation as

$$R_{\uparrow n} := \bigcup \mathcal{R}_{\uparrow n}, \tag{16}$$

which relates a simplex $\sigma$ to all simplices $\tau$ that share a common higher-dimensional simplex containing both as facets. Notably, $B_n, C_n, R_{\downarrow n}, R_{\uparrow n}$, coincide with the four types of adjacent simplices defined in (Bodnar et al., 2021b) under which MPSNN operate. Under this setting, we denote $\mathcal{U}_n = \{B_n, C_n, R_{\downarrow n}, R_{\uparrow n}\}$ and $\mathcal{D}_n = \{B_n, C_n\} \cup \mathcal{R}_{\downarrow n} \cup \mathcal{R}_{\uparrow n}$ with the understanding that only those relations which are defined are included. Accordingly,

$$\mathcal{U} = \bigcup_{n=0}^{N} \mathcal{U}_n \qquad \text{and} \qquad \mathcal{D} = \bigcup_{n=0}^{N} \mathcal{D}_n, \tag{17}$$

are defined. Throughout all experiments, SSNs use the relation set $\mathcal{D}$ equation 17 on 2-dimensional semi-simplicial sets, combining boundary/co-boundary maps with all directed up/down adjacencies for intradimensional communication.

## C.7  INVARIANTS, COLORINGS AND THE WEISFEILER–LEMAN TEST

**Invariants.** An *invariant* for a given class of objects is a function f assigning to each object a value that remains unchanged under isomorphisms. Formally, if $X \cong Y$, then $f(X) = f(Y)$. Objects with identical values under f share a common structural property. A classical example is set cardinality, a *complete invariant*: sets with the same cardinality are isomorphic. In contrast, vertex count is an *incomplete invariant* for graphs, as non-isomorphic graphs can share vertex counts, e.g., a complete and an edgeless graph.

**From Local (vertex) to Global (graph/complex) Invariants.** Given a digraph $\mathcal{G} = (V, E)$ or directed simplicial complex $\mathcal{K} = (V, \Sigma)$, potentially with dynamic binary functions, a *vertex invariant*

is a map $c : V \to \mathcal{C}$ or $c : V \times \{0, 1\}^T \to \mathcal{C}$ invariant under isomorphisms, meaning $c(v) = c(\psi(v))$ for all vertices $v$ under isomorphism $\psi$. Vertex invariants extend naturally to graph or complex invariants through a permutation-invariant aggregation, i.e, functions that are invariant under set isomorphism. Precisely, given a vertex invariant $c$, an induced *binary graph invariant* can be defined as: $c(\mathcal{G}_B) = \phi(\{\{c(v) \mid v \in V\}\})$, where $\phi$ is a permutation-invariant function. Consequently, if $\mathcal{G}_B \cong \mathcal{G}'_B$, then $c(\mathcal{G}_B) = c(\mathcal{G}'_B)$. Real-valued vector vertex invariants arranged as ordered sequences, $c(\mathcal{G}_B) = (c(v_0), \dots, c(v_N))$ according to a fixed vertex ordering $V = \{v_1, \dots, v_N\}$ constitute graph features, which are generally not invariant, as they explicitly depend on the chosen vertex ordering.

**Graph coloring.** Let $\mathcal{G} = (V, E)$ be a graph. A graph coloring is a vertex invariant $c : V \to \mathbb{N}$ which assigns to each vertex a positive integer (its color). For any $n \in \mathbb{N}$, the preimage $c^{-1}(n) = V_n$ partitions the vertex set into subsets of vertices sharing the same color property. The pair $\mathcal{G}_c = (\mathcal{G}, c)$ is called the *colored graph* and is an attributed graph.

**Color Refinement.** A color assigment $c'$ is a *refiment* of $c$ if for every pair $u, v \in V$, $c'(u) = c'(v)$ implies $c(u) = c(v)$. In other words, vertices that share the same color under $c'$ must also share the same color under $c$. Thus, $c$ partitions the vertex set into a potentially richer variety of color classes. Specifically, they are equivalent $c' \cong c$, if for every pair $u, v \in V$, $c'(u) = c'(v)$ if and only if $c(u) = c(v)$.

**The Weisfeiler–Leman Test.** Given a graph $\mathcal{G}$, define an initial coloring $c^0$, e.g., setting $c^0(v) = 0$ for all $v \in V$. This yields the colored graph $\mathcal{G}_0 = (\mathcal{G}, c^0)$; clearly $\mathcal{G} \cong \mathcal{G}'$ if and only if $\mathcal{G}_0 \cong \mathcal{G}'_0$. For a vertex $u \in \mathcal{G}_0$, define the invariant $c_E^l : V \times \mathbb{N} \to \mathcal{M}(\mathbb{N})$ (here, $\mathcal{M}(\mathbb{N})$ denotes the set of finite multisets of $\mathbb{N}$) by $c_E^l(u) = \{\{c^l(v) \mid (u, v) \in E\}\}$, i.e., assigning the multiset of colors of $u$'s neighbours the relation $E$. Then, we obtain a refined coloring $c^{l+1}$ by setting $c^{l+1}(u) = \text{hash}(c^l(u), c_E^l(u)) \in \mathbb{N}$, where $\text{hash} : \mathbb{N} \times \mathcal{M}(\mathbb{N}) \to \mathbb{N}$ is an injective function. The refinement process continues until the coloring stabilizes; that is, until $c^{l+1}(u) = c^l(u)$ for all $u \in V$. Once stability is reached, denote the stable coloring by $c$ and the resulting colored graph by $\mathcal{G}_c = (\mathcal{G}, c)$. Assume that the image of $c$ consists of $k$ distinct colors (which, without loss of generality, can be labeled $\{1, \dots, k\}$). A colored graph invariant $h$ is then obtained by counting the vertices of each color: $h(\mathcal{G}_c) = (\mid c^{-1}(1) \mid, \dots, \mid c^{-1}(k) \mid)$. Thus, for a pair of graphs $\mathcal{G}$ and $\mathcal{G}'$ we have $h(\mathcal{G}_c) \neq h(\mathcal{G}'_c)$, then $\mathcal{G} \ncong \mathcal{G}'$. This algorithmic procedure, yielding incomplete graph invariants discriminating some non-isomorphic pairs, is the Weisfeiler–Leman test (Weisfeiler & Leman, 1968).

# D  THEORETICAL PROPERTIES OF SSNS

A *Message Passing Semi-Simplicial Neural Network* (MPSSN) is a specific instantiation of a semi-simplicial neural network (SSN), as defined in Equation 1, in which each relation-specific transformation $\omega_R$ follows the directional message-passing paradigm introduced in the MPNN-D module (Rossi et al., 2024). In this setting, messages are inherently asymmetric and propagated along the relation $R$: that is, if $(\sigma, \tau) \in R$, then $\omega_R$ updates the representation of $\tau$ using information from $\sigma$, but not the other way around. For the remainder of this section, we adopt MPSSNs as our reference architecture for theoretical analysis.

## D.1  GENERALITY

**Proposition 1.** *Semi-simplicial neural networks (SSNs) subsume directed message-passing GNNs (Rossi et al., 2024) on directed graphs, message-passing GNNs (Gilmer et al., 2017) on undirected graphs, message-passing simplicial neural networks (Bodnar et al., 2021b) on simplicial complexes and Directed Simplicial Neural Networks (Lecha et al., 2025) on directed simplicial complexes.*

*Proof.* Let $\omega_R$ denote the message propagation operator associated with a face-map–induced relation $R$, as defined in Equation 1. We instantiate each $\omega_R$ following the directional message-passing scheme of the MPNN-D framework (Rossi et al., 2024), which propagates information along $R$: specifically, if $(\sigma, \tau) \in R$, then $\omega_R$ updates the embedding of $\tau$ using the features of $\sigma$. First, consider an attributed directed graph $\mathcal{G}_F$. If the SSN is instantiated with the relation set $\mathcal{R} = \{R_{\text{in}}, R_{\text{out}}\}$ and the directional

propagation rules of MPNN-D, the resulting model corresponds exactly to a Dir-GNN (Rossi et al., 2024). Next, if $\mathcal{G}_F$ is undirected—viewed as a symmetric directed graph—then choosing $\mathcal{R} = \{R_{\mathrm{sym}}\}$ yields an $R_{\mathrm{sym}}$-MPSSN that recovers the standard message-passing GNN (Veličković, 2022; Gilmer et al., 2017) operating over $\mathcal{G}_F$. Let $\tilde{\mathcal{K}}_F$ be an attributed simplicial complex, and let $\tilde{\mathcal{K}}_{\mathrm{dir},F}$ be its associated directed simplicial complex induced by a fixed total order on the vertices (see Appendix C.5). Then, an MPSSN operating on $\tilde{\mathcal{K}}_{\mathrm{dir},F}$ with the undirected relation set $\mathcal{U}$ (as defined in Equation 17) corresponds exactly to a message-passing simplicial neural network (MPSNN) (Bodnar et al., 2021b) acting on $\mathcal{K}_F$. Finally, let $\tilde{\mathcal{K}}_F$ be an attributed directed simplicial complex, an MPSSN operating on the set of relations $\mathcal{D}$ as in (as defined in Equation 17) corresponds exactly to a message-passing Directed Simplicial Neural Network (Dir-SNN) (Lecha et al., 2025). Therefore, by appropriately selecting the relation set $\mathcal{R} \subseteq \mathcal{R}_d$ and instantiating the message propagation operators $\{\omega_R\}$ accordingly, the SSN framework unifies and generalizes MPNNs, Dir-GNNs, MPSNNs and Dir-SNNs. $\qquad\square$

### D.2 WEISFEILER–LEMAN EXPRESSIVENESS

**Relational Weisfeiler–Leman Test (R-WL).** Let $S$ be a set equipped with a collection of binary relations $\mathcal{R}$. We extend the standard Weisfeiler–Leman (WL) test to handle multiple relation types, following the formulation of Barcelo et al. (2022). Initialize a coloring $c^0$, e.g., $c^0(\sigma) = 0$ for all $\sigma \in S$. For each $R \in \mathcal{R}$, iteration $l \geq 0$, and $\sigma \in S$, define

$$c_R^l(\sigma) = \{\!\{c^l(\tau) \mid (\sigma,\tau) \in R\}\!\},$$

the multiset of colors of elements related to $\sigma$ under $R$. The refinement step is

$$c^{l+1}(\sigma) = \mathrm{hash}\Big(c^l(\sigma),\, \big(c_R^l(\sigma)\big)_{R\in\mathcal{R}}\Big), \tag{18}$$

where $\mathrm{hash}\colon \mathbb{N} \times \mathcal{M}(\mathbb{N})^{|\mathcal{R}|} \to \mathbb{N}$ is injective. This reduces to the classical 1-WL test (Weisfeiler & Leman, 1968) when $|\mathcal{R}| = 1$.

**Refinement by Unions.** For a finite family $\mathcal{A}$ of relations on $S$, its finite union-closure is

$$\langle \mathcal{A} \rangle \;=\; \Big\{ \bigcup_{R\in F} R \;\Big|\; F \subseteq \mathcal{A},\ F \text{ finite}\Big\}.$$

That is, $\langle \mathcal{A} \rangle$ consists of all finite unions of relations from $\mathcal{A}$. Given families $\mathcal{A}, \mathcal{B}$ of relations, we say that $\mathcal{A}$ *refines* $\mathcal{B}$ *by finite unions* if $\mathcal{B} \subseteq \langle \mathcal{A} \rangle$.

**Lemma 1.** *Let $S$ be a set, and let $\mathcal{A}$ and $\mathcal{B}$ be finite collections of relations on $S$. Suppose $\mathcal{A}$ is a refinement by finite unions of $\mathcal{B}$. Then, $\mathcal{A}$-WL is at least as expressive as $\mathcal{B}$-WL.*

*Proof.* Let $a$ and $b$ denote the colorings of $\mathcal{A}$-WL and $\mathcal{B}$-WL, respectively. We show by induction that $a$ refines $b$ at each iteration $l$. Initially, we set $a^0(\sigma) = b^0(\sigma) = 0$ for all $\sigma \in S$, trivially satisfying the base case. Assume the claim holds at iteration $l$. If $a^{l+1}(\sigma) = a^{l+1}(\tau)$, injectivity of hash implies

$$a_R^l(\sigma) = a_R^l(\tau) \quad \text{for all } R \in \mathcal{A}.$$

By Lemma 2 in (Bevilacqua et al., 2022), equal multisets under a finer coloring remain equal under any coarser coloring; using the inductive hypothesis (that $a^l$ refines $b^l$), we get

$$b_R^l(\sigma) = b_R^l(\tau) \quad \text{for all } R \in \mathcal{A}.$$

Now fix $R' \in \mathcal{B}$. Since $\mathcal{A}$ refines $\mathcal{B}$ by unions, there exists $\mathcal{A}_{R'} \subseteq \mathcal{A}$ such that $R' = \bigcup_{R\in\mathcal{A}_{R'}} R$. By the WL definition, the neighbor multiset for a union relation is the multiset union with multiplicities added (bag sum) of the components:

$$b_{R'}^l(\sigma) = \bigoplus_{R\in\mathcal{A}_{R'}} b_R^l(\sigma).$$

Therefore,

$$b_{R'}^l(\sigma) = \bigoplus_{R\in\mathcal{A}_{R'}} b_R^l(\tau) = b_{R'}^l(\tau).$$

Hence the arguments to hash at iteration $l$ coincide and $b^{l+1}(\sigma) = b^{l+1}(\tau)$. This completes the induction, so $\mathcal{A}$-WL is at least as expressive as $\mathcal{B}$-WL. $\qquad\square$

**Semi-Simplicial Weisfeiler–Leman Test (SSWL).** Let $\mathcal{S}$ be an attributed semi-simplicial set with set of simplices $S$, and let $\mathcal{R} \subseteq \mathcal{R}_d$ be a collection of face-map–induced relations. The *Semi-Simplicial Weisfeiler–Leman Test (SSWL)* is the $\mathcal{R}$-WL test equation 18 applied to $S$.

**Lemma 2.** *Let $S$ be the set of attributed simplices of a semi-simplicial set $\mathcal{S}$ endowed with a collection of face-map–induced relations $\mathcal{R} \subseteq \mathcal{R}_d$. Consider an $\mathcal{R}$-MPSSN defined by equation 1. Suppose the message module $\omega_R$, the aggregator $\bigotimes$, and the update function $\phi$ are all injective. Then, an $\mathcal{R}$-MPSSN is as expressive as the $\mathcal{R}$-SSWL test.*

*Proof.* Let c and x denote the colorings produced by the $\mathcal{R}$-SSWL and the $\mathcal{R}$-MPSSN, respectively. We show by induction that c refines x. At $l = 0$, without loss of generality, we initialize $c^0(\sigma) = x^0(\sigma) = 0$ for all $\sigma \in S$, establishing the base case trivially. Assume the induction hypothesis holds at iteration $l$. Consider two $n$-simplices $\sigma, \tau \in S$ such that $c^{l+1}(\sigma) = c^{l+1}(\tau)$. By the definition of $\mathcal{R}$-SSWL, this implies identical multisets of colors aggregated from neighboring simplices at iteration $l$. By the induction hypothesis, these multisets coincide under $x^l$, ensuring equality in the arguments of the $\mathcal{R}$-SSN, thereby yielding $x^{l+1}(\sigma) = x^{l+1}(\tau)$. By induction, c refines x. Conversely, if $x^{l+1}(\sigma) = x^{l+1}(\tau)$, the injectivity of the composition of injective operators $\omega_R$, $\bigotimes$ and $\phi$, implies identical multisets at iteration $l$. By the induction hypothesis, the hash function receives identical inputs, resulting in $c^{l+1}(\sigma) = c^{l+1}(\tau)$. Thus, $c \cong x$, proving that $\mathcal{R}$-MPSSN and $\mathcal{R}$-SSWL are equally expressive. □

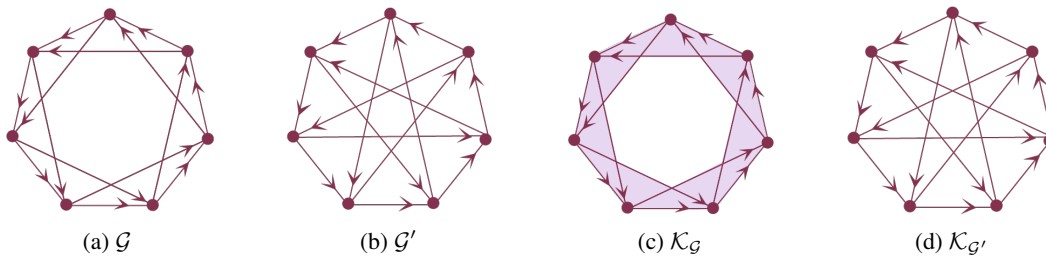

(a) $\mathcal{G}$          (b) $\mathcal{G}'$          (c) $\mathcal{K}_{\mathcal{G}}$          (d) $\mathcal{K}_{\mathcal{G}'}$

Figure 9: A pair of non-isomorphic directed graphs, shown in (a) and (b), along with their corresponding directed flag complexes in (c) and (d), respectively. While these digraphs can be distinguished by SSNs operating on $\mathcal{K}_{\mathcal{G}}$ and $\mathcal{K}_{\mathcal{G}'}$, they cannot be distinguished by Dir-GNNs (Rossi et al., 2024).

**SSN vs. Dir-GNN.** We now prove Theorem 1.

Recall from App. C.2 that a relation $R$ is *$n$-uniform* if each element of $S$ is related to exactly $n$ others.

**Theorem 1.** *There exist SSNs that are strictly more expressive than directed graph neural networks (Dir-GNNs) (Rossi et al., 2024) at distinguishing non-isomorphic directed graphs.*

*Proof.* Let $\mathcal{G}$ be a directed graph and $\mathcal{K}_{\mathcal{G}}$ its corresponding directed flag complex, with maximal dimension 2. Consider the MPSSN defined on $\mathcal{K}_{\mathcal{G}}$ with face-map–induced relations $\mathcal{D}_0 = \{R_{\text{in}}, R_{\text{out}}\}$ and $\mathcal{D}$ as in 17. Then, the proof follows from Theorem 1. of (Lecha et al., 2025). To build intuition, Fig. 12 depicts two non-isomorphic directed graphs in which both $R_{\text{in}}$ and $R_{\text{out}}$ are 2-uniform relations—that is, each node has exactly two incoming and two outgoing neighbors. Assuming constant activation features across all vertices, these graphs cannot be distinguished by Dir-GNNs. In contrast, SSNs separate them by exploiting structural differences in their associated directed flag complexes. □

**MPSSNs vs. MPSNNs.** We prove that there exist instances of MPSSNs that are strictly more powerful than MPSNNs (Bodnar et al., 2021b) at distinguishing directed simplicial complexes.

**Lemma 3.** *Let $\mathcal{K}$ be a directed simplicial complex with set of simplices $\Sigma$, and let $\mathcal{U}$ and $\mathcal{D}$ be collections of face-map–induced relations defined as in Equation 17. Then, $\mathcal{D}$-SSWL is at least as expressive as $\mathcal{U}$-SSWL.*

*Proof.* Since $\mathcal{D}$ is a refinement by unions of $\mathcal{U}$, the result immediately follows from Lemma 1. □

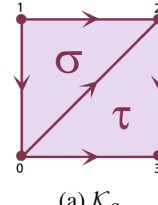 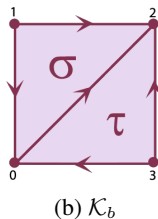 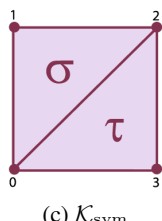

(a) $\mathcal{K}_a$        (b) $\mathcal{K}_b$        (c) $\mathcal{K}_{\mathrm{sym}}$

Figure 10: Two directed simplicial complexes: (a) $\mathcal{K}_a$, (b) $\mathcal{K}_b$, and (c) their shared symmetrized simplicial complex $\mathcal{K}_{\mathrm{sym}}$. Despite $\mathcal{K}_a \not\cong \mathcal{K}_b$, both are mapped to the same undirected complex under symmetrization.

**Lemma 4.** *There exists a pair of directed simplicial complexes that are distinguishable by $\mathcal{D}$-SSWL but indistinguishable by $\mathcal{U}$-SSWL.*

*Proof.* Let $\mathcal{K}_a$, $\mathcal{K}_b$, and $\mathcal{K}_{\mathrm{sym}}$ be as in Figure 10, with constant activation values assigned to all simplices. By construction, the $\mathcal{U}$-SSWL updates on $\mathcal{K}_a$ and $\mathcal{K}_b$ are equivalent to running SWL (Bodnar et al., 2021b) on the shared symmetrized complex $\mathcal{K}_{\mathrm{sym}}$, and thus produce identical outputs. In contrast, $\mathcal{D}$-SSWL distinguishes $\mathcal{K}_a$ and $\mathcal{K}_b$ due to their differing directional structure. $\square$

**Corollary 1.** *$\mathcal{D}$-SSWL is strictly more expressive than $\mathcal{U}$-SSWL.*

*Proof.* Direct consequence of Lemmas 3 and 4. $\square$

**Lemma 5.** *There exist $\mathcal{D}$-SSNs that are strictly more expressive than $\mathcal{U}$-SSNs in distinguishing non-isomorphic directed simplicial complexes (Bodnar et al., 2021b).*

*Proof.* Follows directly from Corollary 1 and Lemma 2. $\square$

**Theorem 2.** *There exist SSNs that are strictly more expressive in distinguishing non-isomorphic directed simplicial complexes than MPSNNs (Bodnar et al., 2021b) under symmetrization.*

*Proof.* Follows directly from Lemma 5 and Proposition 2, which ensures the symmetrization map $\pi$ to be forgetful. $\square$

### D.3 PERMUTATION EQUIVARIANCE AND INVARIANCE

Let $S_F^{\mathcal{R}}$ be an attributed set equipped with relations $\mathcal{R}$. Every permutation $\rho \in \mathrm{Aut}(S)$ induces an isomorphic relabelled structure $(S_F^{\mathcal{R}})_\rho = (\rho \cdot S, \rho \cdot R, \rho \cdot F)$ where $\rho \cdot R = \{ (\rho(\sigma), \rho(\tau)) : (\sigma, \tau) \in R \}$ for all $R \in \mathcal{R}$ and $\rho \cdot F = F \circ \rho^{-1}$. We say that a map $\phi : S_F^{\mathcal{R}} \to (S_F^{\mathcal{R}})'$ is:

- *Permutation equivariant* if, for every $\rho \in \mathrm{Aut}(S)$, it holds that $\phi \circ \rho = \rho \circ \phi$.
- *Permutation invariant* if, for every $\rho \in \mathrm{Aut}(S)$, it satisfies $\phi \circ \rho = \phi$.

Consider an attributed semi-simplicial set $\mathcal{S}$ with attributed simplices $S_F$ and face-map-induced relations $\mathcal{R}$. Representing $S_F$ as a matrix requires assigning an arbitrary global indexing (ordering) to simplices $S = \{\sigma_1, \ldots, \sigma_N\}$ (see Section 2). Neural network models processing these data must therefore exhibit invariance or equivariance under relabeled (permutations of indices) isomorphic structures. Specifically, Semi-Simplicial Neural Networks (SSNs) operate on an attributed set of simplices with face-map-induced relations $S_F^{\mathcal{R}}$, necessitating consistent behavior under reindexing of the entire structure, including both the relations and feature assignments. We prove that SSN layers indeed satisfy this property:

**Theorem 3.** *Let $\mathcal{S}_F^{\mathcal{R}}$ be an attributed semi-simplicial set with face-map-induced relations $\mathcal{R}$. Consider an SSN layer defined as in Equation equation 1. If for each relation $R \in \mathcal{R}$, the mapping $\omega_R$ and the aggregator $\bigotimes$ are permutation equivariant, then the SSN layer is permutation equivariant with respect to simplex reindexing. That is, for all $\rho \in \mathrm{Aut}(S)$, we have:*

$$\mathrm{SSN} \circ \rho = \rho \circ \mathrm{SSN}\,.$$

*Proof.* Let $X^l$ denote the matrix of features at layer $l$, and define the aggregated features $X^m = \bigotimes_{R \in \mathcal{R}} \omega_R(X^l)$. By assumption, each $\omega_R$ is permutation equivariant and $\bigotimes$ is permutation equivariant (or invariant, depending on definition), so for any $\rho \in \text{Aut}(S)$, $\bigotimes_{R \in \mathcal{R}} \omega_R(\rho(X^l)) = \rho(X^m)$. Let $\phi$ be the pointwise update applied by the SSN layer: $[x_{\sigma_i}^{l+1}] = \phi(x_{\sigma_i}^l, x_{\sigma_i}^m)$. Since $\phi$ acts independently on each simplex, for any $\rho$,

$$\phi(\rho(x_{\sigma_i}^l), \rho(x_{\sigma_i}^m)) = \phi(x_{\sigma_{\rho(i)}}^l, x_{\sigma_{\rho(i)}}^m) = x_{\sigma_{\rho(i)}}^{l+1} = \rho(x_{\sigma_i}^{l+1}).$$

Hence $\phi(\rho(X^l), \rho(X^m)) = \rho(\phi(X^l, X^m))$. Combining the equivariance of aggregation and the update yields $\text{SSN} \circ \rho = \rho \circ \text{SSN}$. This completes the proof. $\square$

# E   IMPLEMENTATION DETAILS

## E.1   SEMI-SIMPLICIAL NEURAL NETWORKS (SSNs)

We implement Semi-Simplicial Neural Networks (SSNs) leveraging PyTorch Geometric's `HeteroConv` wrapper (Paszke et al., 2019) to efficiently compute convolutions over heterogeneous graph structures. Consider a semi-simplicial set $\mathcal{S}$ equipped with an attributed set of simplices $S_F^{\mathcal{R}}$ and face-map-induced relations $\mathcal{R}$. Nodes correspond to simplices in $S$, where each node type is defined as $\text{NodeType}(\sigma) = \dim(\sigma)$. Formally, $\mathcal{R}_t^s \subseteq \mathcal{R}$ represents the subset of relations linking simplices of dimension $s$ to those of dimension $t$, with each relation $R_t^s \in \mathcal{R}_t^s$ being a binary relation $R_t^s \subset S_s \times S_t$. The EdgeTypes are thus structured as tuples $(s, R, t)$, encapsulating the relational interactions between simplices of varying dimensions. The update rule for SSNs, targeting simplices of dimension $t$, is defined by modifying the `HeteroConv` layer as follows:

$$X_t^{l+1} = \phi\left(X_t^l, \bigoplus_{s=0}^{\dim(\mathcal{S})} \bigotimes_{R \in \mathcal{R}_t^s} \omega_R(X_s^l)\right). \tag{19}$$

This formulation involves a two-step aggregation procedure: initially, messages from each source dimension $s$ are independently aggregated via relation-specific mappings $\omega_R$. Subsequently, these dimension-specific contributions are merged to update the features of simplices at dimension $t$. Importantly, the `HeteroConv` wrapper enables various convolutional operations, such as SAGE, GAT, or GCN, for implementing the relational mappings $\omega_R$.

## E.2   ROUTING SEMI-SIMPLICIAL NEURAL NETWORKS (R-SSNs)

Let $\mathcal{S}_F^{\mathcal{R}}$ denote an attributed semi-simplicial set equipped with a collection of face-map-induced relations $\mathcal{R} \subset \mathcal{R}_d$. Consider a partition $\mathcal{P}_{\mathcal{R}} = \{\mathcal{R}_1, \ldots, \mathcal{R}_n\}$ of $\mathcal{R}$ into $n$ distinct relation classes. The $l$-th layer of a Routing Semi-Simplicial Neural Network (R-SSN) updates the features $X^l$ as follows:

$$X^{l+1} = \phi\left(X^l, \bigoplus_{\hat{\mathcal{R}} \in \mathcal{P}_{\mathcal{R}}} \bigotimes_{R \in \hat{\mathcal{R}}} G_R(X^l) \cdot \omega_R(X^l)\right), \tag{20}$$

where $G_R(X^l) \in [0, 1]$ is a gating function that outputs normalized weighting scores for each expert. This gating mechanism employs a top-$k$ selection regime, dynamically identifying the $k$ most relevant experts for each subset of relations $\hat{\mathcal{R}} \in \mathcal{P}_{\mathcal{R}}$ during message aggregation. The operator $\bigoplus$ represents the final aggregation of expert representations. Following the details outlined in Appendix E.1, we partition $\mathcal{R}$ into relations $\{R_t^s\}$, where each binary relation $R_t^s \subset S_s \times S_t$. For computational efficiency and practical implementation, we adapt the `HeteroConv` framework to express the R-SSN update at layer $l$ for the $n$-dimensional feature representation:

$$X_t^{l+1} = \phi\left(X_t^l, \bigoplus_{s=0}^{\dim(\mathcal{S})} \bigotimes_{R \in \mathcal{R}_t^s} G(x_s^l) \cdot \omega_R(X_s^l)\right). \tag{21}$$

Here, the representation $x_s^l = \mathrm{P}(X_s^l)$ is obtained through a pooling operator $\mathrm{P} : \mathbb{R}^{K_n \times D^l} \to \mathbb{R}^{D^l}$, where $K_n = |S_n|$ and $D^l$ is the dimensionality of features at layer $l$. For each source dimension $s$, we denote $M = |\mathcal{R}_t^s|$ as the number of available relation experts. The soft gating function $G : \mathbb{R}^{D^l} \to [0,1]^M$ computes normalized scores for expert selection via a top-$k$ mechanism, defined explicitly as:

$$G(x_t^l) = \mathrm{Softmax}\big(\mathrm{TopK}(\mathrm{gates}(x_t^l), k)\big), \tag{22}$$

where the gating scores are calculated as:

$$\mathrm{gates}(x) = x \cdot W_g + \epsilon \cdot \mathrm{Softplus}(x \cdot W_n), \tag{23}$$

with noise $\epsilon \sim \mathcal{N}(0,1)$ and learnable parameters $W_g, W_n \in \mathbb{R}^{D^l \times M}$ that modulate clean and noisy gating scores, respectively. It is well-established that expert selection mechanisms can lead to imbalance issues, with certain experts disproportionately favored during training (Shazeer et al., 2017; Bengio et al., 2015). To address this, we adopt a soft constraint from (Shazeer et al., 2017), incorporating a regularization term in the training loss to encourage equitable distribution of samples among experts. Specifically, for each training sample $x$, we compute the probability $P(x,i)$ that the gating function $G(x)_i$ remains active upon independently re-sampling noise for the $i$-th expert, holding other noises constant. This corresponds to the probability of the $i$-th gating score $\mathrm{gates}(x)_i$, where $\mathrm{gates}(x) = x \cdot W_g + \epsilon \cdot \mathrm{Softplus}(x \cdot W_n)$, to be larger than the $k$-th greatest gating score, excluding itself, i.e.:

$$P(x,i) = Pr(\mathrm{gates}(x)_i > \mathrm{kth\_excluding}(\mathrm{gates}(x), k, i)) \tag{24}$$

where $\epsilon \sim \mathcal{N}(0,1)$ and $\mathrm{kth\_excluding}$ computes the $k$-th largest element of $\mathrm{gates}(x)$ excluding the $i$-th element. Following (Shazeer et al., 2017), we can simplify this to:

$$P(x,i) = \Phi\left(\frac{(x \cdot W_g)_i - \mathrm{kth\_excluding}(\mathrm{gates}(x), k, i)}{\mathrm{Softplus}(x \cdot W_n)_i}\right) \tag{25}$$

where $\Phi$ is the cumulative density function of the standard normal distribution. We now define the load vector $\mathrm{Load}(X)$, i.e., an estimator of the number of samples assigned to each expert given a batch $X$, whose components are

$$\mathrm{Load}(X)_i = \sum_{x \in X} P(x,i). \tag{26}$$

Finally, the additional loss term $\mathcal{L}_{\mathrm{load}}$ is:

$$\mathcal{L}_{\mathrm{load}} = \lambda_{\mathrm{load}} CV \left(\mathrm{Load}(X)\right)^2 \tag{27}$$

where $CV(\cdot)$ computes the coefficient of variation and $\lambda_{\mathrm{load}}$ is a hyperparameter that weights the contribution of this term in the total loss. Minimizing this term corresponds to minimizing the variation of the number of samples assigned to each expert, i.e., balancing the load across experts, and has been shown beneficial in practice (Shazeer et al., 2017).

### E.3 COMPUTATIONAL RESOURCES

Experiments were conducted on a single NVIDIA A100, NVIDIA L40 GPU, NVIDIA A40 or NVIDIA V100 GPU. The total training time for all experiments was approximately two weeks. Hyperparameter tuning was managed using Weights & Biases.

## F  COMPUTATIONAL COMPLEXITY

We analyze the effect of refining relations in relational message passing architectures, as we are interested in moving from undirected to directed (i.e., direction-aware) relations. In these models, a separate message is computed per relation instance of a given relation type. Formally, let $S$ be a finite attributed set (e.g., the set of attributed simplices of a semi-simplicial set). The asymptotic complexity is governed by three quantities that may grow with the input: $N$, the number of elements in $S$; $D$, the hidden feature dimension; and $E$, the total number of relation instances (edges), each corresponding to a message. In contrast, $P$, the number of relation types, is treated as an input-independent constant determined by the model design.

*Example (Dir-GNN).* Consider a directed, attributed graph $G = (V, \mathcal{E})$. Here, the attributed set is $S = V$, the set of nodes. The relation set is $\mathcal{B} = R_{\text{in}}, R_{\text{out}}$, corresponding to incoming and outgoing edges. Thus, $P = |\mathcal{B}| = 2$. While $N = |V|$ and $E = |\mathcal{E}|$ may grow with the input size, $P$ remains fixed by construction.

**Proposition 3.** *Let $\mathcal{S}$ be an attributed set of $N$ elements with matrix form $X^l \in \mathbb{R}^{N \times D^l}$, and let $\mathcal{B}$ be a collection of $P$ relations on $\mathcal{S}$, such that for each relation $R \in \mathcal{B}$ is a set of $E_R$ elements and $E = \sum_R E_R$. Let $H$ be an SSN layer of $X^{l+1} = H(X^l, \mathcal{B})$ where $X^{l+1} \in N \times D^{l+1}$ with $\omega_R$ an MPNN-D module (Rossi et al., 2024). Then, its forward pass time complexity is*

$$T_{\mathcal{B}} = \mathcal{O}(ND^2 + ED).$$

*Proof.* Each relation $R \in \mathcal{B}$ involves two operations: (i) A dense projection of the $N$ node features, which requires $X^l W_R \in \mathbb{R}^{N \times D^{(l+1)}}$ is $\mathcal{O}(ND^l D^{l+1})$. (ii) A message-passing step over the edges $E_R$ of the relation, each edge transmitting a $D^{l+1}$-dimensional message, with cost $\mathcal{O}(E_R D^{l+1})$. Summing over all $P$ relations yields: $T_{\mathcal{B}} = \sum_{R \in \mathcal{B}} [\mathcal{O}(ND^l D^{l+1}) + \mathcal{O}(E_R D^{l+1})] = \mathcal{O}(PND^l D^{l+1} + ED^{l+1})$. Assuming $D^l \approx D^{l+1} = D$ we obtain:

$$T_{\mathcal{B}} = \mathcal{O}(PND^2 + ED).$$

Moreover, if treating $P$ as a small constant, one can argue:

$$T_{\mathcal{B}} = \mathcal{O}(ND^2 + ED).$$

$\square$

**Corollary 2.** *Let $\mathcal{S}$ be a set, and let $\mathcal{A}$ and $\mathcal{B}$ be two collections of relations on $\mathcal{S}$, such that $\mathcal{A}$ is a finite refinement by unions of $\mathcal{B}$. Then, $T_{\mathcal{A}} = T_{\mathcal{B}}$.*

*Proof.* By Proposition 3, $T_{\mathcal{B}} = \mathcal{O}(ND^2 + ED)$. Since $\mathcal{A}$ refines $\mathcal{B}$ by splitting each coarse relation into at most a constant number $Q$ of finer ones, the same summation over relations yields $T_{\mathcal{A}} = \mathcal{O}(QND^2 + ED)$. Then, $T_{\mathcal{A}} = \mathcal{O}(ND^2 + ED) = T_{\mathcal{B}}$. Thus the refinement does not change the asymptotic forward-pass cost. $\square$

**Example (Dir-GNN):** The forward-pass cost of one SSN layer applied to this setup is, by our derived bound:

$$T = O(PND^2 + ED) = O(2ND^2 + ED).$$

As is standard in Big-O notation, constants can be absorbed, yielding:

$$O(ND^2 + ED),$$

as stated as a corollary in the proof of our theorem. We highlight that this bound exactly matches the complexity reported for Dir-GNN (Sec. 3, p. 6, (Rossi et al., 2024)), showing consistency between our general analysis and this specific case.

## G  TOPOLOGICAL DEEP REPRESENTATION LEARNING FOR BRAIN DYNAMICS

### G.1  DATA

We build upon a simulation that was run on a Blue Brain Project (Markram, 2006), a biologically validated digital reconstruction of a microcircuit in the somatosensory cortex of a two-week-old rat

(the NMC-model) (Markram et al., 2015) used in subsequent neurotopological studies (Reimann et al., 2017; Conceição et al., 2022; Reimann et al., 2022). The model involves two fundamental components: structural connectivity of the circuit and neuronal binary dynamics. First, *structural connectivity* is modeled by a directed graph $\mathcal{G} = (V, E)$, with neurons represented by nodes $V$ and directed edges $(u, v) \in E$ denoting synaptic connections from presynaptic neuron $u$ to postsynaptic neuron $v$. Second, a collection of *binary dynamics* $\mathcal{B} = \{B : V \to \{0, 1\}^T\}$ is defined, where each dynamic $B$ encodes neuronal activity over time, capturing neuronal firing (1) and quiescent (0) states across $T$ discrete time bins, under 8 stimulus-driven input patterns. The stimuli were delivered by activating thalamocortical afferent fibers—axons that carry sensory signals from the thalamus to the cortex, modeling how the brain receives external input. These synaptic input fibers were organized into bundles; specifically, the 2170 input fibers were partitioned into 100 spatially adjacent bundles using $k$-means clustering, reflecting the biological organization in which thalamic afferents target specific cortical zones. Each stimulus activated 10 randomly selected bundles (approximately 10% of the afferents), ensuring that the same groups of fibers were targeted for each stimulus pattern. The activation followed an adapting, stochastic spiking process, which introduces variability and biological realism while preventing the memorization of fixed patterns. Stimuli were presented as a continuous stream: every 200 ms, a decaying and adapting stochastic spiking process activated the corresponding fiber bundles during a predominant 10 ms interval. The 200 ms inter-stimulus interval was chosen based on the observation that the population response to each stimulus decayed to baseline within 100 ms. Each stimulus pattern was repeated approximately $562 \pm 4$ (mean $\pm$ std) times, yielding a total of 4495 stimulus presentations. Each stimuli simulation time is segmented into a fixed number of $T$ time bins, for each bin the set of neurons that became active (i.e., exhibited a binary state of 1) is recorded, thereby defining a set of 4495 binary dynamics $\mathcal{B} = \{V \to \{0, 1\}^T\}$. In particular, following Conceição et al. (2022), we focus on the time subinterval $\Delta t = [10\,\mathrm{ms}, 60\,\mathrm{ms}]$, where spiking activity is mostly concentrated. This interval is subdivided into two 25-ms segments, yielding a set of 4495 binary dynamics $\mathcal{B} = \{V \to \{0, 1\}^2\}$. Here, $B_t(v) = 1$ indicates that a neuron $v$ is active during the $t$-th 25-ms segment (with $t \in \{0, 1\}$) in the experiment.

## G.2 Dynamical Activity Complexes

In Section 4, we introduced a lifting procedure that maps dynamic binary digraphs to dynamic binary directed simplicial complexes. This transformation enables the representation of directed higher-order neural co-activation motifs in a structured format that is both expressive and compatible with graph-based and TDL models. To establish the soundness of this lifting, we verify two fundamental properties. First, isomorphism preservation: if two dynamic binary digraphs are isomorphic, their lifted representations must also be isomorphic. This ensures that equivalent neural dynamics yield identical higher-order structures. Second, consistency with the neurotopological pipeline: a Dynamical Activity Complex (DAC) must encode the time series of *functional complexes*—that is, the directed flag complexes derived from the subgraphs induced by active neurons at each time step. In this section, we formally prove both properties. Given a dynamic binary graph $\mathcal{G}_B$, or more generally a dynamic binary directed simplicial complex $\mathcal{S}_B$, let $V^{1,t}$ denote the set of active vertices at time $t$, and $\Sigma^{1,t}$ the corresponding set of active simplices.

First, the following proposition guarantees that isomorphic digraph dynamics yield identical higher-order co-activation structures.

**Proposition 4.** *Let $\mathcal{G}_B \cong \mathcal{G}'_B$ then $\mathcal{K}_{\mathcal{G},\tilde{B}} \cong \mathcal{K}_{\mathcal{G}',\tilde{B}}$.*

*Proof.* Recall that two binary graphs are isomorphic if there exists a bijection between their vertex sets that simultaneously preserves edge relations and node attributes. The directed flag complex lifting is known to be invariant under digraph isomorphisms. Therefore, it suffices to show that our assignment of binary activation sequences to simplices remains invariant under isomorphism. Concretely, for each simplex $\sigma$, we assign the activation pattern

$$\tilde{B}(\sigma) = \left[\min_{v \in \sigma} B_1(v), \ldots, \min_{v \in \sigma} B_T(v)\right] \in \mathbb{B}^T. \tag{28}$$

where $B_t(v) \in \{0, 1\}$ denotes the activation of vertex $v$ at time $t$. Since the minimum function $\min$ is permutation-invariant, any reindexing of vertices induced by a graph isomorphism preserves these activation sequences. Hence, the entire dynamic binary lifting commutes with graph isomorphisms, concluding the proof. More generally, any permutation-invariant aggregation function would suffice. $\qquad\square$

Second, we show that the lifted structure encodes the full time series of *functional complexes* as subcomplexes.

**Proposition 5.** *Let $\mathcal{G}_B$ be dynamic binary graph and $\mathcal{K}_{\mathcal{G},\tilde{B}}$ its associated DAC. If $\mathcal{G}^{1,t} = \mathcal{G}[V^{1,t}]$ is the functional digraph with functional complex $\mathcal{K}_{\mathcal{G}^{1,t}}$, then:*

$$\mathcal{K}_{\mathcal{G}^{1,t}} = \mathcal{K}_{\mathcal{G},\tilde{B}}[\Sigma^{1,t}]$$

*Proof.* A simplex belongs to $\Sigma^{1,t}$ exactly when all vertices are active at time $t$. Therefore $\Sigma^{1,t}$ is closed under taking faces implying that $\mathcal{K}_{\mathcal{G},\tilde{B}}[\Sigma^{1,t}]$ forms a subcomplex of $\mathcal{K}_{\mathcal{G},\tilde{B}}$. In particular, simplices of $\mathcal{K}_{\mathcal{G},\tilde{B}}[\Sigma^{1,t}]$ correspond precisely to the cliques of the functional digraph $\mathcal{G}^{1,t}$, thereby $\mathcal{K}_{\mathcal{G}^{1,t}} = \mathcal{K}_{\mathcal{G},\tilde{B}}[\Sigma^{1,t}]$. $\qquad\square$

### G.3 TOPOLOGICAL INVARIANTS FOR DYNAMICAL ACTIVITY COMPLEXES

Let $\mathcal{G}_B$ be a dynamic binary digraph and $\mathcal{K}_{\mathcal{G},\tilde{B}}$ its corresponding Dynamical Activity Complex (DAC). We describe the dynamic binary topological invariants frequently employed in neurotopological studies of (Reimann et al., 2017; Conceição et al., 2022; Reimann et al., 2022), leveraging their structure and associated face-map-induced relations. These invariants play a central role in characterizing the evolving topological structure of brain activity. Given a face-map-induced relation $R$ on $\mathcal{G}_B$ or $\mathcal{K}_B$, we define its $k$-hop composition restricted to the active simplices at time $t$ as

$$R^{1,t,k} = \{(\sigma,\tau) \in R^{\circ k} \mid \tau \in \Sigma^{1,t}\},$$

where $\Sigma^{1,t}$ denotes the set of simplices active at time $t$. These restricted relational structures provide the foundation for extracting topological descriptors of time-evolving brain activity from DACs.

**Size.** Let $R_{\text{sym}}$ be defined as in Equation 10. Define the *$k$-hop functional size* of vertex $u$ as:

$$\text{size}(u,k) = \left[|R_{\text{sym}}^{1,t,k}(u)|\right]_{t=0}^{T},$$

counting active $k$-hop neighbors per node at each time step, i.e., $k$-hop synaptically connected active neuron. This invariant, despite its simplicity, effectively distinguishes stimuli and outperforms other invariants in practice (Conceição et al., 2022).

**Euler Characteristic.** For a dynamic binary directed simplicial complex $\mathcal{K}_B$ the *functional Euler Characteristic* is defined as the alternating sum:

$$\text{ec}(\mathcal{K}_B) = \sum_{n=0}^{N}(-1)^n \left[|\Sigma_n^{1,t}|\right]_{t=0}^{T},$$

where $N$ is the maximal dimension of $\mathcal{K}$. For a dynamic binary digraph $\mathcal{G}_B$, the functional Euler characteristic of its associated DAC $\text{ec}(\mathcal{K}_{\mathcal{G}_B})$ coincides with the time series of the Euler characteristic of the functional flag complexes $[\chi(\mathcal{K}_{\mathcal{G}^{1,t}})]_{t=1}^{T}$, computing the alternating sum of active simplices across dimensions at each time step (see Proposition 6). The variation in the amplitude of the Euler characteristic time series of functional complexes effectively characterises stimuli (Reimann et al., 2017). Moreover, it has been reported as a top-performing feature for stimulus classification (Conceição et al., 2022).

**Proposition 6.** *Let $\mathcal{G}_B$ be dynamic binary graph and $\mathcal{K}_{\mathcal{G},\tilde{B}}$ its associated DAC. Then,*

$$\text{ec}(\mathcal{K}_{\mathcal{G},\tilde{B}}) = [\chi(\mathcal{K}_{\mathcal{G}^{1,t}})]_{t=0}^{T}.$$

*Proof.* By definition $\text{ec}(\mathcal{K}_{\mathcal{G},\tilde{B}}) = [\chi(\mathcal{K}_{\mathcal{G},\tilde{B}}[\Sigma^{1,t}])]_{t=0}^{T}$. From Proposition 5 it follows that $\text{ec}(\mathcal{K}_{\mathcal{G},\tilde{B}}) = [\chi(\mathcal{K}_{\mathcal{G}^{1,t}})]_{t=0}^{T}$. $\qquad\square$

**Transitive Degree.** Focusing on 2-dimensional directed flag complexes (consistent with our experimental setup in Section 5), let $C_{0,2}$ be defined as in Section C.5. The *transitive degree* of vertex $v$ is the number of directed 3-cliques (equivalently, 2-simplices) containing $v$:

$$\text{td}(v) = \left[|C_{0,2}^{1,t}(v)|\right]_{t=0}^{T},$$

the active directed 2-simplices (transitive synaptic triads) containing vertex $v$ at each time $t$.

**Graph Level Directionality.** Let $R_{\text{in}}$ and $R_{\text{out}}$ be as defined in Equations 8 and 9. he *active $k$-hop in-degree* and *out-degree* of a vertex $u$ are defined as:

$$\text{indeg}(u, k) = \left[ |R_{\text{in}}^{1,t,k}(u)| \right]_{t=0}^{T} \text{ and } \text{outdeg}(u, k) = \left[ |R_{\text{out}}^{1,t,k}(u)| \right]_{t=0}^{T},$$

respectively counting the number of active $k$-hop incoming and outgoing neighbours. The resulting *$k$-hop functional signed degree* is:

$$\text{dir}(u, k) = \text{indeg}(u, k) - \text{outdeg}(u, k),$$

quantifying local asymmetry in neuronal activity (Govc et al., 2021).

**High-Order Directionality.** Let $R_{\downarrow n,i,j}$ be defined as in Equation 4. For an $n$-simplex ($n > 0$), define the $(i,j)$-th *$k$-hop functional signed degree* as:

$$\deg_{n,i,j}(\sigma, k) = \left[ |R_{\downarrow n,i,j}^{1,t,k}(\sigma)| \right]_{t=0}^{T},$$

which relates $n$-simplices to their lower $(i,j)$ $k$-hop adjacent active neighbors. For $i \neq j$, define:

$$\text{hodir}_{n,i,j}(\sigma, k) = \deg_{n,i,j}(\sigma) - \deg_{n,i,j}(\sigma).$$

This invariant was shown to be critical for characterizing structural directed flag complexes in brain connectivity (Riihimäki, 2023).

**Reciprocity.** Let $R_{rc} = R^{\text{in}} \cap R^{\text{out}}$. The *$k$-hop functional reciprocal degree* of vertex $u$ is defined as:

$$\text{rc}(u, k) = \left[ |R_{\text{rc}}^{1,t,k}(u)| \right]_{t=0}^{T},$$

counting active vertices simultaneously serving as in- and out-neighbors of $u$. This invariant effectively differentiates stimulus classes (Conceição et al., 2022).

Following the procedure described in App. D.2, for any simplex- or vertex-level invariant $\text{t} \in \{\text{size}, \text{dir}, \text{hodir}, \text{rc}, \text{td}\}$, we define the corresponding global graph or complex-level invariant as

$$\text{t}(\mathcal{K}_{\mathcal{G}, \tilde{B}}) = \phi(\{\{\text{t}(v)\}\}_{v \in V}),$$

where $\phi$ is a permutation-invariant aggregation function.

## G.4 TOPOLOGICAL NEURAL NETWORKS AND DYNAMICAL ACTIVITY COMPLEX INVARIANTS

From now on, let denote $\mathcal{T} = \{\text{size}, \text{ec}, \text{td}, \text{dir}, \text{hodir}, \text{rc}\}$ denote the collection invariants.

**Lemma 6.** *Let $\mathcal{K}_B$ be a DAC with a labeled set of simplices $S$ and corresponding binary feature matrix $X \in \mathbb{B}^{|S| \times T}$, where each row encodes the activation pattern of a simplex over $T$ discrete time steps. For every invariant $\text{T} \in \mathcal{T}$ there exists a subcollection $\mathcal{R}_{\text{T}} \subset \mathcal{R}_d$ of face-map–induced relations and corresponding operators $\{\lambda_R\}_{R \in \mathcal{R}_T}$ such that*

$$\text{T}(\mathcal{K}_B) = \phi( \sum_{R \in \mathcal{R}_T} \lambda_R(A_R X)),$$

*where $\phi$ is a permutation-invariant function.*

*Proof.* For a fixed relation $R$, consider the matrix product $(A_R X)$. The $(i,t)$-th entry is

$$(A_R X)_t^i = (A_R)^i \cdot (X)_t = \sum_{j=0}^{|S|} \mathbb{1}_{\sigma_j \in R(\sigma_i)} \cdot \mathbb{1}_{(X)_t^j = 1}.$$

This sum counts the number active simplices $\sigma_j$ at time $t$ that belong to $R(\sigma_i)$. In other words, $(A_R X)_t^i = |\{\sigma_j \in R(\sigma_i) : X_t^j = 1\}| = |R^{1,t}(\sigma_i)|$, where we define $R^{1,t}(\sigma_i)$ as the set of simplices in $R(\sigma_i)$ with an active feature at time step $t$. Hence, the $i$-th row is $(A_R X)^i = (|R^{1,0}(\sigma_i)|, \ldots, |R^{1,T}(\sigma_i)|)_{t=0}^{T}$, counting active neighbours of $\sigma_i$ under the relation $R$ across the $T$ time steps. For each relation $R$, we choose $\lambda_R$ to further process the count matrix $A_R X$. We set $\lambda_R = Id$ to be the identity for relations $R \in R_{in}, R_{sym}, R_{rc}, C_{0,2}, R_{\downarrow,i,j}\}$, the negation $\lambda_{R_{out}} = -Id$ for the relation

$R_{out}$ and $\lambda_{\mathrm{id}_s} := (-1)^s$ sum (a row-sum with sign adjustment) for $\mathrm{id}_s$, denoting the identity relation restricted to elements of dimension $s$. These choices ensure that the processed outputs capture the necessary counts (with any required sign modifications) for computing the invariant. Let $\mathcal{R}_T \subseteq \mathcal{R}$ be the subcollection of relations relevant for the invariant T. Define the aggregated representation as $Z = \sum_{R \in \mathcal{R}_T} \lambda_R(A_R X)$. Since the order of simplices in $S$ is arbitrary, we apply a permutation invariant function $\phi$ (such as the row-sum, row-mean, or row-max) to $Z$ yielding an invariant quantity: $\mathrm{T}(\mathcal{K}_B) = \phi(Z)$. This completes the proof. $\square$

**Corollary 3.** *Let $\mathcal{K}_B$ be a DAC. There exists a Semi-Simplicial Neural Network (SSN) that computes each invariant in $\mathcal{T}$.*

*Proof.* For each relation $R \in \mathcal{R}$, define $\omega_R(X) := A_R X W$, with $W = I_{T \times T}$ (the $T \times T$ identity matrix). Next, let an operator $\bigotimes$ act on the output of $\omega_R(X)$ corresponding to the operator $\lambda_R$. Then, define an aggregation operator $\bigoplus$ as the summation over the subcollection $\mathcal{R}_T \subseteq \mathcal{R}$ necessary for the computation of a given invariant: $\bigoplus_{R \in \mathcal{R}_T} \bigotimes_R \omega_R(X) = \sum_{R \in \mathcal{R}_T} \lambda_R(A_R X)$. Finally, applying a permutation invariant function $\phi$ to this aggregated output yields the invariant:

$$\mathrm{T}(\mathcal{K}_B) = \phi\Big( \bigoplus_{R \in \mathcal{R}_T} \bigotimes_R \omega_R(X)\Big).$$

Thus, an SSN structured in this way can compute each invariant in $\mathcal{T}$. $\square$

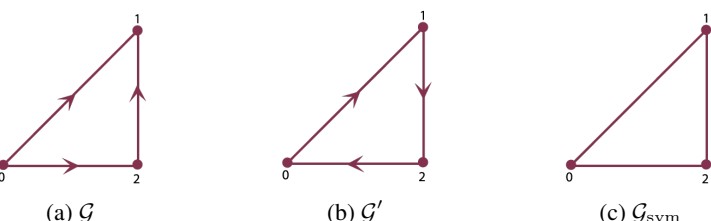

(a) $\mathcal{G}$        (b) $\mathcal{G}'$        (c) $\mathcal{G}_{\mathrm{sym}}$

Figure 11: Two directed graphs: (a) $\mathcal{G}$, (b) $\mathcal{G}'$, and (c) their shared symmetrized graph $\mathcal{G}_{\mathrm{sym}}$. Despite $\mathcal{G} \not\cong \mathcal{G}'$, both are mapped to the same undirected complex under symmetrization.

Our strategy is straightforward: construct non-isomorphic objects that (i) are distinguishable by invariants but (ii) remain indistinguishable to the corresponding architecture due to bounded expressivity—or, when required, become isomorphic under symmetrization $\pi$ (App. C.3). The following lemmas establish this through explicit examples. Throughout, symmetrization $\pi$ implicitly extends to constantly attributed simplices, where permutations of vertices collapse into a single canonical simplex with identical constant features.

**Lemma 7.** *MPNNs and MPSNNs cannot compute invariants* dir, hodir *and* rc.

*Proof. Directionality.* Let $V = \{0, 1, 2\}$ and define two binary digraphs with constant activation $B(i) = 1$ for all $i \in V$. Graph $\mathcal{G}$ (Fig. 11(a)) with edges $E = (0, 1), (0, 2), (1, 2)$ forms a transitive 3-clique with $\mathrm{dir}(\mathcal{G}_B) = \{\{-2, 0, 2\}\}$, corresponding respectively to source (vertex 0), an intermediate vertex (vertex 1), and a sink (vertex 2). Graph $\mathcal{G}'$ (Fig. 11(b)) with the edge set $E' = \{(0, 1), (1, 2), (2, 0)\}$ forms a 3-cycle. Every vertex has equal in- and out-degree, giving $\mathrm{dir}(\mathcal{G}'_B) = \{\{0, 0, 0\}\}$. Thus dir separates $\mathcal{G}$ and $\mathcal{G}'$, but their symmetrizations coincide: $\pi(\mathcal{G}) \cong \pi(\mathcal{G}')$ (Fig. 11(c)).

*Higher-Order Directionality.* On $V = \{0, 1, 2, 3\}$ consider $\mathcal{K}_\mathcal{G}$ and $\mathcal{K}_{\mathcal{G}'}$ from Fig. 5(b), both with constant activations. Their higher order directional signatures differ, e.g.

$$\mathrm{hodir}_{2,0,2}(\mathcal{K}_{\mathcal{G}_B}) = \{\{1^{\times 2}, -1^{\times 2}\}\} \neq \{\{0^{\times 4}\}\} = \mathrm{hodir}_{2,0,2}(\mathcal{K}_{\mathcal{G}'_B}),$$

as $\mathcal{K}_\mathcal{G}$ contains two directionally consistent 2-simplicial paths, while $\mathcal{K}_{\mathcal{G}'}$ yields disconnected 2-simplices; yet their symmetrizations coincide, $\pi(\mathcal{K}_{\mathcal{G}_B}) \cong \pi(\mathcal{K}_{\mathcal{G}'_B})$.

*Reciprocity.* Let $\mathcal{G}_B$ with edges $E = \{(0, 1), (0, 2), (1, 2), (2, 1)\}$ and $\mathcal{G}'_B$ with $E' = \{(0, 1), (0, 2), (1, 2)\}$. Only $\mathcal{G}_B$ contains reciprocal edges, hence $\mathrm{rc}(\mathcal{G}_B) \neq \mathrm{rc}(\mathcal{G}'_B)$. Still, $\pi(\mathcal{G}_B) \cong \pi(\mathcal{G}'_B)$, and the same holds for their directed flag complexes.

In all cases, symmetrization erases the distinctions. Since MPNNs and MPSNNs operate only on the symmetrized $\pi(\cdot)$ objects, they cannot compute these invariants. □

Recall from App. C.2 that a relation $R$ is *n-uniform* if each element of $S$ is related to exactly $n$ others.

**Lemma 8.** *MPNNs and Dir-GNNs cannot compute invariants* ec*,* td*, and* hodir*.*

*Proof.* Consider the binary directed complexes $\mathcal{K}_{\mathcal{G}_B}$ and $\mathcal{K}_{\mathcal{G}'_B}$ in Fig. 9(a)–(b), both with constant activation. Each has 7 vertices and 14 edges, but $\mathcal{K}_{\mathcal{G}_B}$ contains exactly 7 transitive 3-cliques, whereas $\mathcal{K}_{\mathcal{G}'_B}$ includes none. Hence their invariants differ:

$$\mathrm{tc}(\mathcal{K}_{\mathcal{G}_B}) = \{\{3^{\times 7}\}\} \neq \{\{0^{\times 7}\}\} = \mathrm{tc}(\mathcal{K}_{\mathcal{G}'_B}),$$

$$\mathrm{ec}(\mathcal{K}_{\mathcal{G}_B}) = 7 - 14 + 7 = 0 \neq -7 = 7 - 14 = \mathrm{ec}(\mathcal{K}_{\mathcal{G}'_B}),$$

$$\mathrm{hodir}_{2,0,2}(\mathcal{K}_{\mathcal{G}_B}) = \{\{1^{\times 7}\}\} \neq \{\{\emptyset\}\} = \mathrm{hodir}_{2,0,2}(\mathcal{K}_{\mathcal{G}_B}).$$

Dir-GNNs propagate only through $R_{\mathrm{in}}$ and $R_{\mathrm{out}}$, each a 2-uniform relation (every node connected to two neighbors). MPNNs, by contrast, operate via $R_{\mathrm{sym}}$, a 4-uniform relation obtained by merging them. With constant activations, such uniformity makes neighborhood multisets indistinguishable, preventing these models from recovering the invariants above. □

**Theorem 4.** *Let $\mathcal{G}_B$ be a dynamic binary digraph with corresponding DAC $\mathcal{K}_{\mathcal{G}, \tilde{B}}$. For every invariant $\mathrm{T} \in \mathcal{T}$, there exists a set of face-map-induced relations $\mathcal{R}_{\mathrm{T}} \subset \mathcal{R}_d$ and a Semi-Simplicial Neural Network SSN as in Equation equation 1 such that:*

$$\mathrm{SSN}(X, \mathcal{R}_{\mathrm{T}}) = \mathrm{T}(\mathcal{K}_{\mathcal{G}, \tilde{B}}).$$

*Moreover, the class of invariants recoverable by SSNs strictly exceeds that of message-passing neural networks (Gilmer et al., 2017), directed GNNs (Rossi et al., 2024), and message-passing simplicial networks (Bodnar et al., 2021b).*

*Proof.* Follows immediately from Corollary 3 and Lemmas 7 and 8. □

## H   ADDITIONAL NUMERICAL RESULTS

**SSN Relations.** Throughout, we operate with the collection of relations $\mathcal{D}$ equation 17 on semi-simplicial sets of dimension 2, comprising standard boundary/co-boundary maps together with all directed up/down adjacencies enabling intradimensional communication.

### H.1   DYNAMICAL BRAIN ACTIVITY CLASSIFICATION

We provide comprehensive details on the dataset, hyperparameter configurations, TopoFeat+SVM baseline and runtimes for the main experiments in Sections 5.1 and 5.2. Additionally, we report results for an attention-based variant of SSN and we present numerical evidence demonstrating the robustness of our model under an alternative, non-invariant readout setting—applicable exclusively to tasks without induced structural variability, such as brain dynamics representation on fixed neuronal samples.

#### H.1.1   EXPERIMENTAL DETAILS

**Fixed Volumetric Samples.** Directed simplices have been shown to be overexpressed motifs in brain networks at all scales (Sizemore et al., 2018; Tadić et al., 2019; Sizemore et al., 2019; Andjelković et al., 2020). Table 4 presents the structural statistics of the simplicial complexes derived from three representative volumetric samples: $(4, 125\mu m)$, $(4, 325\mu m)$, and $(8, 175\mu m)$ used in our experimental evaluation. These statistics highlight the intrinsic structural complexity and high-dimensional

| Cell Type | $(4, 125\mu\text{m})$ | $(4, 325\mu\text{m})$ | $(8, 175\mu\text{m})$ |
|---|---|---|---|
| Nodes | 600 | 600 | 600 |
| Edges | 19,209 | 7,450 | 10,852 |
| Triangles | 62,481 | 5,322 | 14,141 |
| Tetrahedra | 26,450 | 349 | 2,240 |
| Pentachorons | 1,757 | 4 | 61 |
| Hexaterons | 30 | 0 | 0 |

Table 4: Simplex count per dimension for each volumetric sample used in the experiments.

organization present within localized regions of the neocortical microcircuit, underscoring the relevance of higher-order topological representations in modeling neural computation.

**Hyperparameter Configuration.** For all tested models (SSNs and competitors), hyperparameters were optimized as follows: number of layers $\in \{2, 4, 6\}$; hidden dimension $\in \{16, 32, 64\}$ for non-topological models (GNN, Dir-GNN), and $\in \{16, 32\}$ for topological models (MPSNN, SSN). Additional settings were fixed across all models: dropout rate of $0.3$; inner aggregation set to $\text{sum}$; outer aggregation set to $\text{mean}$; batch size of $16$; Adam (Kingma & Ba, 2014) optimizer with learning rate of $0.001$; early stopping with a patience of $25$ validation steps; and validation performed at every training step. The best-performing configurations across five splits are reported in Table 5, along with their corresponding parameter counts. To ensure further fair comparisons, we also scaled the non-topological baselines (GNN-256, Dir-GNN-256 with hidden dimension 256) and the undirected topological baseline (MPSNN-64 with hidden dimension 64).

| Model | Hid Dim | # Layers | # Params | Par. Ratio (%) |
|---|---|---|---|---|
| DS | 64 | 2 | 1,680 | 2% |
| DS-256 | 256 | 2 | 70,672 | 68% |
| GNN | 64 | 2 | 5,392 | 5% |
| GNN-256 | 256 | 2 | 70,672 | 68% |
| Dir-GNN | 64 | 2 | 9,744 | 9% |
| Dir-GNN-256 | 256 | 2 | 137,232 | 133% |
| MPSNN | 32 | 4 | 23,888 | 23% |
| MPSNN-64 | 64 | 4 | 90,768 | 88% |
| **R-SSN (Ours)** | 32 | 6 | 18,084 | 18% |
| **SSN (Ours)** | 32 | 6 | 103,184 | 100% |

Table 5: Model architecture details. Parameter counts (# Params) and relative percentages (Par. Ratio %) are reported compared to SSNs.

**Runtime.** Table 6 reports the average runtime per epoch for both validation and training. On larger complexes, the routing mechanism in R-SSN proves advantageous: for the $(4, 125, \mu m)$ case ($\approx 60k$ simplices), R-SSN requires only about 70% of the time of a standard SSN, while for $(8, 175, \mu m)$ ($\approx 27k$ simplices), the time is reduced to 88%. For the smaller $(4, 325, \mu m)$ complex ($\approx 13k$ simplices), by contrast, the validation speedup is marginal, and during training SSN is slightly faster than R-SSN. This runtime benefit stems from the fact that the number of active experts (and, consequently, active relations) is decided in advance and can be significantly less than the number of relations present in SSN—even at training time. As a result, overall computation is reduced compared to the full model. Moreover, in backpropagation only the gradients of the weights tied to active experts need to be updated, i.e., those that actually contribute to the output of the network. At the same time, R-SSNs introduce a small extra cost due to the gating mechanism parameters. The net advantage arises only when the savings from discarding relation experts outweigh the overhead of gating. In practice, this effect depends on complex size: for small complexes, eliminating relations may not fully offset the added $\approx 5.8k$ parameters of the gating mechanism. For larger datasets, however, the reduction in computation is substantial, showing that R-SSN scales more efficiently and can provide meaningful training speed improvements as the number of simplices increases.

| Model | Valid. Avg. Runtime (s/epoch) | | | Train Avg. Runtime (s/epoch) | | |
|---|---|---|---|---|---|---|
| | $(4, 125\mu m)$ | $(4, 325\mu m)$ | $(8, 175\mu m)$ | $(4, 125\mu m)$ | $(4, 325\mu m)$ | $(8, 175\mu m)$ |
| DS | $23.37 \pm 0.09$ | $8.42 \pm 0.04$ | $9.60 \pm 0.17$ | $94.18 \pm 0.46$ | $39.56 \pm 1.50$ | $33.96 \pm 0.32$ |
| DS-256 | $23.68 \pm 0.07$ | $8.46 \pm 0.03$ | $9.75 \pm 0.19$ | $94.85 \pm 0.42$ | $39.40 \pm 0.33$ | $33.92 \pm 0.32$ |
| GNN | $21.35 \pm 0.05$ | $7.87 \pm 0.15$ | $9.00 \pm 0.26$ | $85.73 \pm 0.17$ | $32.08 \pm 0.56$ | $35.70 \pm 0.39$ |
| GNN-256 | $22.04 \pm 0.41$ | $7.66 \pm 0.19$ | $8.19 \pm 0.22$ | $104.94 \pm 0.77$ | $33.16 \pm 0.15$ | $42.46 \pm 2.01$ |
| Dir-GNN | $21.49 \pm 0.01$ | $7.83 \pm 0.13$ | $8.90 \pm 0.13$ | $86.18 \pm 0.10$ | $32.17 \pm 0.53$ | $36.41 \pm 0.32$ |
| Dir-GNN-256 | $22.50 \pm 0.49$ | $7.79 \pm 0.12$ | $8.39 \pm 0.20$ | $103.6 \pm 2.3$ | $33.75 \pm 0.07$ | $40.23 \pm 0.84$ |
| MPSNN | $37.18 \pm 0.25$ | $9.98 \pm 0.05$ | $12.58 \pm 0.06$ | $190.5 \pm 0.8$ | $44.71 \pm 0.13$ | $61.08 \pm 0.15$ |
| MPSNN-64 | $52.36 \pm 0.17$ | $11.12 \pm 0.16$ | $15.60 \pm 0.19$ | $298.8 \pm 0.48$ | $53.35 \pm 0.48$ | $81.60 \pm 0.19$ |
| **SSN (Ours)** | $52.18 \pm 0.15$ | $9.98 \pm 0.22$ | $16.58 \pm 0.22$ | $300.3 \pm 0.8$ | $52.68 \pm 0.51$ | $91.22 \pm 0.39$ |
| **R-SSN (Ours)** | $37.79 \pm 0.14$ | $9.30 \pm 0.19$ | $14.15 \pm 0.15$ | $211.7 \pm 0.3$ | $54.18 \pm 0.63$ | $80.93 \pm 0.51$ |

Table 6: Avg. runtime per validation epoch (left) and training epoch (right) for different model configurations.

### H.1.2 ADDITIONAL RESULTS.

**Topological features baseline.** To assess the impact of end-to-end feature extraction, we compare with a linear SVM on a feature vector of precomputed topological invariants as defined in Appendix G.3. For each DAC, we compute the following invariants: Euler characteristic ($ec$), Graph-level directionality ($dir$), Neighborhood size ($size$) and higher-order directionality ($hodir$), computed on the (0,1) relation for edges and on the (0,1), (1,2), (0,2) relations for triangles. Node-level invariants ($size$, $dir$, $hodir$) are summed to obtain complex-level values. The $dir$, $size$ and $hodir$ invariants can be computed for different neighborhood orders $K$. All features are computed across two time bins and concatenated, resulting in a feature vector of size $2(6K + 1)$ per sample, where 2 are the time bins, $ec$ has size 1, $size$ and $hodir$ have size $K$ and $hodir$ has size $4K$. The parameter count for the SVM classifier is based on the adopted one-vs-all classification strategy, which trains one SVM for each class. Each linear SVM has $2(6K + 1) + 1$ parameters (i.e., the input feature size plus one), which leads to a total of $16(6K + 1) + 8$ parameters. In Table 3 we report the number for $K = 3$ which leads to the best performance. The results in Table 7 show that increasing $K$ leads to better classification performance, corroborating the importance of considering higher-order relations in the complex.

| $K$ | $(4, 125\mu m)$ | $(4, 325\mu m)$ | $(8, 175\mu m)$ | M = 1 | M = 3 | M = 5 |
|---|---|---|---|---|---|---|
| 1 | $35.17 \pm 0.41$ | $29.81 \pm 1.05$ | $37.37 \pm 2.11$ | $27.94 \pm 0.94$ | $26.97 \pm 0.97$ | $27.60 \pm 0.38$ |
| 2 | $39.60 \pm 0.82$ | $35.15 \pm 1.68$ | $43.22 \pm 1.28$ | $27.63 \pm 0.89$ | $27.23 \pm 0.97$ | $28.44 \pm 0.27$ |
| 3 | $42.14 \pm 1.19$ | $35.91 \pm 2.36$ | $45.32 \pm 1.68$ | $27.76 \pm 0.66$ | $27.87 \pm 0.89$ | $28.86 \pm 0.42$ |

Table 7: Accuracy for the TopoFeat+SVM baselines for varying $K$ (%, higher is better ↑). The top **1st**, **2nd**, and **3rd** results are highlighted.

**Attention-based SSN.** We further evaluate SSNs and baselines by using a GAT message-passing scheme (Veličković et al., 2018) as $\omega_R$ in equation 1 instead of GraphSAGE (used in Table 3). Table 8 shows that SSN largely outperforms all baselines also in this configuration on the $(4, 325\mu m)$ and $(M = 3)$ datasets, corroborating its improved capability to leverage higher-order directed connectivity information under different message-passing schemes.

| Model | $(4, 325\mu m)$ | M = 3 |
|---|---|---|
| GAT | $22.46 \pm 1.48$ | $24.42 \pm 0.62$ |
| Dir-GAT | $42.03 \pm 0.31$ | $28.59 \pm 0.62$ |
| MPSNN-GAT | $32.97 \pm 8.59$ | $29.55 \pm 0.81$ |
| **SSN-GAT (Ours)** | $77.45 \pm 3.58$ | $51.27 \pm 1.85$ |
| **Gain** | ↑35.42% | ↑21.72% |

Table 8: Accuracy for GAT message-passing scheme (%, higher is better ↑). The top **1st**, **2nd**, and **3rd** results are highlighted. **Gain** reports the absolute accuracy improvement (↑%) of our model relative to the best performing baseline.

### H.1.3 NON-INVARIANT READOUTS.

In this work, we develop a model capable of robustly processing arbitrary localized structural regions within neural microcircuits, represented as Dynamical Activity Complexes (DACs). Unlike prior methods that rely on pooled neuronal samples, our approach addresses the inherent structural variability present in localized microcircuit data. As an initial evaluation, we assess our model on a task involving fixed topology but varying brain dynamics. To prevent artificially inflated accuracy due to consistent neuron indexing across samples, we employ permutation-invariant readouts. Remarkably, our model demonstrates robustness to shuffled spike trains as a natural consequence of its design, highlighting its ability to learn solely from topological activation patterns. To further challenge and validate its generalization capabilities, we additionally evaluate the model under non-invariant readout settings, where a consistent ordering of the fixed feature space eases the task of recovering stimulus identity through localized consistent activation patterns.

| Model | $(4, 125\mu\mathrm{m})$ | $(4, 325\mu\mathrm{m})$ | $(8, 175\mu\mathrm{m})$ |
|---|---|---|---|
| DS | **86.65 ± 1.06** | **92.26 ± 1.07** | **88.72 ± 0.82** |
| GNN | 77.67 ± 1.96 | 87.88 ± 0.45 | 84.98 ± 1.43 |
| Dir-GNN | 85.07 ± 0.91 | 90.94 ± 0.81 | 87.42 ± 1.43 |
| MPSNN | **86.95 ± 0.52** | **92.09 ± 1.29** | **88.72 ± 1.20** |
| SSN (Ours) | **87.63 ± 0.43** | **92.02 ± 0.82** | **88.98 ± 1.10** |
| Gain | ↑**0.68%** | ↓**0.24%** | ↑**0.26%** |

Table 9: Binary dynamics classification results (%, higher is better ↑) across volumetric samples. The top **1**ˢᵗ, **2**ⁿᵈ, and **3**ʳᵈ results are highlighted. **Gain** reports the absolute accuracy improvement (↑%) or drop (↓%) of our model relative to the best performing baseline.

**Hyperparameter Configuration.** For all evaluated models (SSNs and baselines), hyperparameters were optimized over the following grid: number of layers $\in \{2, 4, 6\}$, hidden dimension $\in \{16, 32, 64\}$, and dropout rate $\in \{0.3, 0.8\}$. The following settings were fixed across all models: inner aggregation set to sum, outer aggregation to mean, batch size of 16, and the Adam optimizer (Kingma & Ba, 2014) with a learning rate of 0.001. Early stopping was applied with a patience of 25 validation steps, and validation was conducted after every training iteration. The best-performing configurations averaged over five data splits are reported in Table 9.

**Results.** In this setting, all baselines perform significantly better than in our main experiments (see Table 3), reflecting the advantage of fixed structure settings for stimulus identification. Nevertheless, SSN achieves the highest accuracy in two of the three configurations and performs comparably in the remaining one, underscoring its ability to extract meaningful topological features even without structural variability. Notably, SSN's performance correlates with the topological complexity of each volumetric sample (see Table 4). Moreover, methods that rely on a fixed neuron ordering are intrinsically limited to those specific samples, making the models in Table 9 strong baselines only in these constrained scenarios.

### H.1.4 INCREASED TEMPORAL RESOLUTION AND VARIATION IN VOLUMETRIC SAMPLING

We provide additional numerical results for Section 5.2, evaluating two alternative scenarios: (i) increased temporal resolution ($T = 4$ time bins), and (ii) variation in volumetric sampling.

**Experimental Settings.** We extend our evaluations in the more challenging data-scarce setting ($N = 1$), exploring two additional regimes: component $(4, 325\mu\mathrm{m})$ with increased temporal resolution ($T = 4$ time bins), and component $(4, 125\mu\mathrm{m})$ to assess volumetric consistency. This comprehensive experimental framework is designed to rigorously evaluate the robustness of our model under both temporal and volumetric sampling variability.

**Results.** Table 10 shows that SSNs consistently achieve the highest classification accuracy across all evaluated settings, substantially outperforming baseline models in both the increased sub-neighborhood volume scenario and the finer temporal resolution setting ($T = 4$). Notably, despite

| Model | $125\mu$m | $T = 4$ |
|---|---|---|
| DS | $22.06 \pm 0.26$ | $23.13 \pm 0.39$ |
| DS-256 | $21.62 \pm 0.61$ | $23.65 \pm 0.36$ |
| GNN | $24.45 \pm 0.65$ | $22.87 \pm 0.49$ |
| GNN-256 | $24.03 \pm 0.60$ | $22.36 \pm 0.18$ |
| Dir-GNN | $23.46 \pm 0.39$ | $23.18 \pm 0.23$ |
| Dir-GNN-256 | $25.03 \pm 1.35$ | $22.67 \pm 0.35$ |
| MPSNN | $25.88 \pm 0.66$ | $24.79 \pm 1.21$ |
| MPSNN-64 | $26.44 \pm 1.26$ | $24.43 \pm 0.34$ |
| **SSN (Ours)** | $46.30 \pm 2.11$ | $39.14 \pm 8.09$ |
| **Gain** | ↑**19.86%** | ↑**14.35%** |

Table 10: Binary Dynamical Complex classification results (%, higher is better ↑). The top **1**[st], **2**[nd], and **3**[rd] results are highlighted. Absolute accuracy Gain over the second-best model are also reported.

these variations, SSNs exhibit comparable performance to previously tested configurations, indicating that the model's accuracy is largely invariant to changes in temporal resolution and sampling volume.

## H.2 Edge Regression Traffic Task

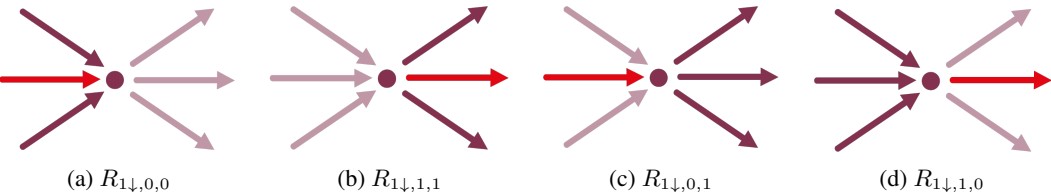

(a) $R_{1\downarrow,0,0}$    (b) $R_{1\downarrow,1,1}$    (c) $R_{1\downarrow,0,1}$    (d) $R_{1\downarrow,1,0}$

Figure 12: Examples of the four lower directed edge adjacencies used for SSN in the experiments on edge flow prediction. The first one (a) connects each edge to those sharing the same destination node; the second one (b) connects each edge to those sharing the same source node; the third one (c) connects each edge to those leaving from its destination node; the fourth one (d) connects each edge to those entering in its source node.

| Model | Anaheim | Barcelona | Chicago | Winnipeg |
|---|---|---|---|---|
| MLP | 0.096 | 0.150 | 0.111 | 0.165 |
| LineGraph | 0.087 | 0.147 | 0.112 | 0.160 |
| HodgeGNN | 0.270 | 0.170 | 0.109 | 0.174 |
| Hodge+Inv | 0.096 | 0.146 | 0.108 | 0.162 |
| Hodge+Dir | 0.081 | 0.146 | 0.110 | 0.159 |
| EIGN | 0.069 | 0.130 | 0.061 | 0.090 |
| **SSN (Ours)** | 0.077 | 0.130 | 0.045 | 0.116 |
| **Gain** | ↑0.008 | 0.000 | ↓0.016 | ↑0.026 |

Table 11: Traffic dynamics regression results (RMSE, lower is better ↓). The top **1**[st], **2**[nd], and **3**[rd] results are highlighted. Absolute Gain over the best-performing baseline model is also reported (negative ↓ is better).

We evaluate our model on a traffic assignment problems using 4 traffic datasets from (for Research Core Team, Accessed 06.08.2024) (Anaheim, Barcelona, Chicago and Winnipeg). These datasets contain the street networks from the corresponding cities and the relative oriented traffic flows from the Traffic Assignment Problem (Patriksson, 2015). Some streets can be traversed in both ways, others only in one way, providing the problem setting with an inherent notion of directionality. Furthermore, traffic flows are edge data, making the problem topological. We frame the task as a traffic simulation

problem following Fuchsgruber et al. (2025): given the traffic flows on a portion of the streets and the network topology, the goal is to predict the flows on the remaining streets.

**Experimental Settings.** We follow the same data-preprocessing as (Fuchsgruber et al., 2025). In detail, we define the following edge features (if available): capacity, length, free flow time (i.e., travel time with no congestion), B factor and power, (calibration parameters for the Traffic Assignment Problem), if there is a toll on the link, and the link type, e.g., highway. Differently from (Fuchsgruber et al., 2025), if a street is traversed in both directions, we do not transform it into a unique undirected edge, but we maintain the two directed edges in opposite directions, keeping their original directed flows. We whiten features and normalize the target flows between 0 and 1. To compare our results with those in (Fuchsgruber et al., 2025), where undirected edges have a unique flow, we consider the difference between the flow predictions of the two directed edges in opposite directions at evaluation time (i.e., to compute the RMSE). For training, instead, we design a loss to account for both the intrinsic directionality of the problem and the existence of directed edges in opposite directions. Specifically, our training loss is composed of two terms:

$$\mathcal{L}_{\text{train}} = \alpha \mathcal{L}_{\text{dir}} + (1-\alpha)\mathcal{L}_{\text{diff}} \tag{29}$$

where $\mathcal{L}_{\text{dir}}$ is a regression loss (e.g., MSE) on the directed flows, $\mathcal{L}_{\text{diff}}$ is the loss between the predicted and ground truth flow differences for undirected edges and $\alpha \in [0,1]$ is a hyperparameter. We test our SSN using different combinations of the 4 directed edge adjacencies illustrated in Figure 12, and we report results for the best one consisting of adjacencies (b) and (c) in Figure 12 plus the undirected edge adjacency consisting of the union of (b) and (c). We compare with six baselines: MLP; LineGraph, a spectral GNN applied at node-level; HodgeGNN (Roddenberry & Segarra, 2019), based on the edge Laplacian; Hodge+Inv, a variant of HodgeGNN modeling orientation-invariant features as orientation equivariant; Hodge+Dir, a variant of HodgeGNN that treats all edges as directed; EIGN (Fuchsgruber et al., 2025), a GNN for edge signals that explicitly differentiates between orientation and direction invariance and equivariance for edge features. All the results for the baselines are taken from (Fuchsgruber et al., 2025) as well as the experimental setting.

**Hyperparameter Configuration.** We selected the hyperparameters through a grid search among the following values: convolution strategy $\in \{\text{GCN}, \text{SAGE}\}$, batch normalization $\in \{\text{True}, \text{False}\}$, hidden size $\in \{32, 64, 128\}$, $\alpha \in \{0, 0.5, 1\}$, number of layers $\in \{1, 3, 5, 9\}$. We fixed the following other parameters for all models: inner aggregation set to mean, single batch, dropout rate of 0.1, learning rate of 0.01, 1500 epochs with early stopping with a patience of 80 validation steps and validation performed every training step. We report the best parameter configurations for each dataset in Table 12.

**Results.** Table 11 shows the MSE on the edge flow prediction for SSN and baselines. SSN outperforms all baselines on Chicago and matches the best-performing model on Barcelona, demonstrating the capability of its directed relations scheme to model edge flow invariance and equivariance to directionality. On Anaheim and Winnipeg, SSN is outperformed by EIGN, but is close or better than the second baseline, showing an improvement w.r.t. undirected approaches which, again, validates the importance of considering directed adjacencies on this task.

| Dataset | Hid dim | # Layers | Conv. | BN | $\alpha$ |
|---|---|---|---|---|---|
| Anaheim | 32 | 9 | GCN | False | 1 |
| Barcelona | 64 | 5 | GCN | True | 1 |
| Chicago | 64 | 5 | SAGE | False | 0 |
| Winnipeg | 64 | 9 | GCN | False | 0.5 |

Table 12: Model architecture details for edge-level datasets.

## H.3 NODE CLASSIFICATION

We further assess SSN's performance on a node classification task involving both homophilic and heterophilic graphs. In homophilic graphs, nodes with the same label tend to be connected, while in heterophilic graphs, connected nodes typically belong to different classes—making these settings particularly challenging for GNNs. Rossi et al. (2024) showed that modeling graphs as directed

| Dataset | Type | Path Length | # $R$-Paths | $R_{\downarrow1,0,0}(\%)$ | $R_{\downarrow1,0,1}(\%)$ | $R_{\downarrow1,1,0}(\%)$ | $R_{\downarrow1,1,1}(\%)$ |
|---|---|---|---|---|---|---|---|
| Cora ML | Homophilic | 1 | 355,808 | 57.50 | 12.05 | 12.05 | 18.41 |
| | | 2 | 31,672,710 | 93.99 | 0.70 | 0.70 | 4.60 |
| Citeseer | Homophilic | 1 | 76,344 | 77.22 | 6.46 | 6.46 | 9.87 |
| | | 2 | 2,665,146 | 98.06 | 0.23 | 0.23 | 1.48 |
| Roman-Empire | Heterophilic | 1 | 222,096 | 14.90 | 30.23 | 30.23 | 24.65 |
| | | 2 | 1,384,820 | 7.93 | 27.29 | 27.29 | 37.50 |

Table 13: Higher-order edge directionality statistics for the Cora-ML, Citeseer (Bojchevski & Günnemann, 2018) and Roman-Empire (Platonov et al., 2023) datasets. The analysis reveals a pronounced chain-like structure in the Roman-Empire graph, characterized by a higher proportion of fully directed edge paths. In contrast, Cora-ML and Citeseer exhibit predominantly homophilic patterns, with directionality concentrated in edge pairs sharing a common source or target, and a markedly lower prevalence of fully directed paths.

| Model | Roman-Empire |
|---|---|
| GraphSAGE | $91.06 \pm 0.27$ |
| Dir-GNN Rossi et al. (2024) | $91.23 \pm 0.32$ |
| Polynormer | $92.55 \pm 0.37$ |
| MPSNN | $88.76 \pm 0.63$ |
| SSN (Ours) | $93.52 \pm 0.28$ |
| Gain | $\uparrow 0.96\%$ |

| Model | Cora-ML | Citeseer |
|---|---|---|
| GNN | $87.06 \pm 1.47$ | $94.34 \pm 0.56$ |
| Dir-GNN | $86.60 \pm 1.43$ | $94.09 \pm 0.48$ |
| MPSNN | $86.58 \pm 1.41$ | $94.14 \pm 0.83$ |
| SSN (Ours) | $86.64 \pm 1.30$ | $94.52 \pm 0.53$ |
| Gain | $\downarrow 0.42\%$ | $\uparrow 0.18\%$ |

Table 14: Node classification results (accuracy in %, higher is better ↑) on the Roman-Empire dataset (Left column) (Platonov et al., 2023) and on Cora-ML and Citeseer (Right column) datasets (Bojchevski & Günnemann, 2018). The top two performing methods are highlighted as follows: 1st and 2nd. **Gain** reports the accuracy improvement ($\uparrow\%$) or drop ($\downarrow\%$) of our model relative to the best performing baseline. In the Roman-Empire table, the 2nd entry corresponds to the previous state-of-the-art method.

in heterophilic contexts can induce meaningful relationships that effectively increase the graph's homophily, highlighting the potential benefits of leveraging directional information. In this work, we focus on the Roman-Empire dataset, whose distinctive chain-like topology makes it especially well-suited for higher-order directed modeling. We evaluate our method on this heterophilic benchmark, alongside two widely used homophilic citation graphs: Cora-ML and Citeseer (Bojchevski & Günnemann, 2018). Our empirical findings align with the conclusions of Rossi et al. (2024): directionality—whether higher-order or not—offers minimal gains in traditional strongly homophilic benchmarks but yields substantial improvements in highly directed and heterophilic Roman-Empire dataset (see Table 13). In particular, we observe that higher-order directionality plays a key role in boosting performance in such settings, setting a new state of the art on the Roman-Empire dataset (cf. Table 14).

**Roman Empire (A Chain-Like Graph).** The Roman-Empire graph of Platonov et al. (2023) is constructed from the Roman Empire Wikipedia article, where each node represents a word token and directed edges encode either immediate word succession or syntactic dependency. This process yields a highly directed, heterophilic graph (heterophily score ≈ 0.05), enriched with shortcut edges that capture long-range grammatical dependencies and characterized by a chain-dominated topology—featuring the smallest average per-node degree (2.91) and largest diameter (6824) among commonly used benchmark datasets. Table 13 quantifies the higher-order edge directionality: at path length 1, fully directed edge pairs ($R_{\downarrow1,0,1}$) account for 30.23% of one-hop interactions—more than double the proportion observed in Cora-ML (12.05%) and Citeseer (6.46%), where edges typically share a common source or target. This structure makes Roman-Empire an ideal stress test for our Semi-Simplicial Neural Network (SSN). While traditional undirected topological models inherently neglect directionality, and recent approaches like Dir-GNN (Rossi et al., 2024) leverage only first-order directional cues, SSN explicitly captures higher-order directed simplicial motifs, effectively modeling long-range syntactic chains and dependencies.

**Baselines.** For the Roman-Empire dataset, we benchmark against several strong baselines: Dir-GNN (Rossi et al., 2024), which previously established the efficacy of direction-aware models; Graph-SAGE (Hamilton et al., 2017), with extensive hyperparameter tuning by Luo et al. (2024)—yielding the most competitive configurations for undirected graph-based methods; and Polynormer (Deng et al., 2024), a Graph Transformer (GT) that currently holds state-of-the-art performance. The tuning methodology in (Luo et al., 2024) aligns with that of Polynormer (Deng et al., 2024). Previous results are taken directly from their respective works. Additionally, we evaluate MPSNN (Bodnar et al., 2021b), enabling a comprehensive comparison across models that incorporate, or omit, higher-order and directional interactions. For the homophilic datasets Cora-ML and Citeseer, we benchmark SSN against standard baselines: GNN (Hamilton et al., 2017), Dir-GNN (Rossi et al., 2024), and MPSNN (Bodnar et al., 2021b), as detailed in Table 14.

**Experimental Setup.** For the Roman-Empire dataset, we adopt the data splits from (Platonov et al., 2023). For Cora-ML and Citeseer, we use 10 random splits with a 50/25/25 train-validation-test ratio, reporting mean accuracy and standard deviation across splits (see Table 14). In our SSN model, we restrict simplicial dimension to 1, lifting node-pair embeddings into common directed simplices (syntactic or citation-based connections) via the boundary converse operator $C_0$. We propagate embeddings across four directional edge relationships ($R_{\downarrow 1,0,0}$, $R_{\downarrow 1,0,1}$, $R_{\downarrow 1,1,0}$, and $R_{\downarrow 1,1,1}$), updating node embeddings using boundary operator $B_1$.

**Hyperparameters.** Consistent with (Rossi et al., 2024) and (Luo et al., 2024), we employ concatenation-based Jumping Knowledge. Differing from (Luo et al., 2024), who utilized hidden dimensions of 256 (GraphSAGE) and 512 (GAT and GCN), we uniformly set a smaller hidden dimension of 128 and utilize SAGE-like aggregations ($\omega_R$) across all five relations. Hyperparameter search covers the number of layers $\{5, 7, 9\}$, dropout rates $\{0.3, 0.5, 0.7\}$, inner aggregation as max, outer aggregation as sum, and the Adam optimizer (learning rate = 0.01).

**Results.** Table 14 demonstrates that our SSN sets a new state of the art on the Roman-Empire dataset, surpassing the Polynormer Graph Transformer baseline. SSN notably improves accuracy by $2.46\%$ over classical graph methods on best tuning known performed in (Luo et al., 2024) and outperforms Dir-GNN by $2.29\%$. This highlights SSN's superior ability to leverage higher-order directionality in effectively capturing complex relational structures in heterophilic, chain-like graphs. Additionally, SSN consistently outperforms MPSNN, reinforcing the critical importance of explicitly modeling directed higher-order structures. For the homophilic datasets, SSN achieves competitive performance—surpassing all baselines by $0.18\%$ on Citeseer and remaining within $0.42\%$ of the best-performing method on Cora-ML. These results align with the findings of (Rossi et al., 2024), confirming that incorporating directionality yields minimal benefit in strongly homophilic settings. An additional possible explanation for the limited gains from higher-order directionality is provided by the edge directionality statistics reported in Table 13, which reveal that message passing in both Cora-ML and Citeseer predominantly occurs along edges with shared sources or targets—a structural hallmark of citation networks. This is accompanied by a marked collapse in the proportion of fully directed paths. As a result, the diversity of directional relationships becomes largely redundant, diminishing the marginal utility of higher-order directed modeling in these scenarios.

