# OpenReview forum: "Directed Semi-Simplicial Learning with Applications to Brain Activity Decoding"
_ICLR.cc/2026/Conference — ICLR 2026 Poster_

### Official Review · Reviewer_7KeL · 2025-10-31

**Soundness:** 3
**Presentation:** 3
**Contribution:** 3
**Rating:** 6
**Confidence:** 1

**Summary:**

The paper introduces Semi-Simplicial Neural Networks (SSNs), which learn on directed, higher-order structures (semi-simplicial sets) rather than just pairwise edges. A scalable variant (Routing-SSN) learns which relations to use. The authors prove SSNs are more expressive than standard GNNs and existing topological models and can recover key topological descriptors of brain activity. On simulated neocortical stimulus-decoding tasks, SSNs achieve state-of-the-art results and sizable gains over baselines.

**Strengths:**

- The paper has a well-motivated focus on directionality and multi-way interactions, important for brain data.
- General framework that subsumes GNNs, directed GNNs, and simplicial networks.
- Theoretical results on WL-expressivity and recovery of neuro-topological invariants.
- Strong empirical gains on brain-dynamics classification.

Overall, I find the paper to be strong. It introduces a novel framework that is well-grounded in theory, effectively subsuming several pioneering baselines. The experimental evaluation also appears comprehensive, covering a wide range of datasets, tasks, and key baselines, which collectively serve as an implicit ablation study for the proposed architecture.

That said, I am not deeply familiar with the field of brain activity modeling, and from my view it seems that the baselines considered are primarily taken from [4] (e.g., GNNs, DeepSets, MPSNN, etc.), potentially missing other relevant "high order message passing approaches" approaches such as [1, 2, 3]. Given my limited expertise in this domain of brain activity—particularly regarding datasets and evaluation protocols—I would prefer to calibrate my final assessment after reviewing feedback from other reviewers who may have more specialized knowledge in this area.

**References:**

[1] Cin++: Enhancing topological message passing. Giusti et.al. 2023

[2] Cycle invariant positional encoding for graph representation learning. Yan et al. 2024

[3] Attending to topological spaces: The cellular transformer. Ballester et. al 2024

[4] Position: Graph learning will lose relevance due to poor benchmarks Bechler-Speicher et al., 2025

**Weaknesses:**

- There is no comparison with other topological or higher-order message-passing baselines (e.g., [1, 2, 3]), if they are applicable on these tasks. If these methods are indeed not relevant or comparable in this context, it would be helpful for the authors to clarify why that is the case.
- Minor weakness: the paper lacks an ablation study showing which types of relations contribute most to performance or how sensitive results are to the choice of relations.

**References:**

[1] Cin++: Enhancing topological message passing. Giusti et.al. 2023

[2] Cycle invariant positional encoding for graph representation learning. Yan et al. 2024

[3] Attending to topological spaces: The cellular transformer. Ballester et. al 2024

**Questions:**

1. How should practitioners pick the top-k in Routing-SSN beyond grid search?
2. Can routing scores or other signals identify which directed motifs drove a prediction?

---

> ### Author Response · Authors · 2025-11-18
>
> We thank the reviewer for the thoughtful assessment.  We appreciate the recognition of the paper’s motivation and the acknowledgement of both the theoretical soundness of our approach and the strong empirical results on brain-dynamics decoding. We value the reviewer’s care in evaluating the work despite noting limited familiarity with brain-activity modeling, since the core contributions of the paper are general: the theoretical results and methodology apply broadly, and the neuroscience task serves as an impactful but controlled setting where higher-order directionality is both well defined and empirically necessary, while also contributing a new real-world dataset at a time when meaningful TDL benchmarks remain scarce. We appreciate the constructive questions, which help further clarify the scope and contributions of the work. We address each point in detail below.
>
>
> # [Q1]
> We thank the reviewer for raising this question, as the role of k in Routing-SSN differs from that of a standard task-specific hyperparameter. Routing-SSN is not meant to introduce an additional dimension of tuning; rather, k acts purely as a computational budget, limiting how many relation-experts may be active in each layer. This allows one to work with a large relational vocabulary while keeping both the forward and backward pass proportional only to the k selected experts.
>
> In practice, the practitioner specifies the full set of meaningful directed relations for the domain—for example, the six directed face/upper/lower adjacencies in dimension 2—chooses a computational budget k, and the routing mechanism automatically learns which relations are most informative for the task. Because k controls computation rather than expressivity, it does not require grid search. It is simply a user-chosen budget: in all our experiments we fix k = 2.
>
> In short, k should be viewed as a budget parameter rather than a domain-specific hyperparameter. The practitioner selects a feasible computational cost, and Routing-SSN determines which relations matter.
>
> # [Q2]
> We thank the reviewer for the question. Routing-SSN does provide signals that can help identify which relations and thus which classes of directed interactions the model relies on. Since the routing mechanism activates only the top-k relation-experts at each layer, one can directly inspect which directed adjacencies are consistently selected across batches or data splits. This produces a coarse but informative interpretability signal about the directional patterns the network finds useful.
>
> At the same time, interpreting these selections at the level of task-specific directed motifs is nontrivial. Relation-experts act in a feature-dependent manner, and their influence propagates across layers, so the selected relations may vary with the input, the layer, and the training split. A principled mapping from routing scores to explicit motif-level explanations would therefore require dedicated methodological development, which lies beyond the scope of the present work.
>
> We nevertheless performed a qualitative analysis to illustrate the type of insights routing scores can offer. On the (4, 325) dataset, setting k = 2, we examined the most frequently selected triangle relations at layer 2 and found stable patterns within each split (90 batches per split):
>
> Split 1: (0,0), (2,0) — 90/90
>
> Split 2: (0,2), (2,1) — 68/90
>
> Split 3: (2,0), (2,1) — 86/90
>
> Split 4: (0,0), (2,2) — 87/90
>
> Split 5: (2,0), (2,1) — 82/90
>
> These selections point to coherent directed structures: for instance, (0,2) and (2,0) relations propagate along directed triangle chains, whereas (2,1) captures flows looping around a central vertex. Such intra-split consistency suggests that Routing-SSN is leveraging structured directional information. That said, developing a systematic motif and relational level interpretation remains an interesting direction for future work rather than a claim we make here.

---

> ### Author Response · Authors · 2025-11-18
>
> # [W1]
>
> We thank the reviewer for raising this point. Our evaluation is deliberately aligned with the mathematical and real-world neuroscientific setting in which our theoretical guarantees hold. DACs are binary, dynamic, directed flag complexes encoding the full temporal evolution of functional complexes and their co-activation events—exactly the constructions used (and proved successful) in prior neuroscience work to compute neurotopological invariants for brain-activity decoding. Our theoretical results (Props. 4–5, Thm. 4) are derived for this data model, and the goal of our experiments is to validate these guarantees when the appropriate structural assumptions hold. This justifies the SotA results we show, and also represents a real-world but controlled setting in which we can evaluate whether SSNs, when applied to directed simplicial data, indeed improve over existing topological architectures.
>
> Crucially, this setting dictates the appropriate baselines. Removing directionality from a DAC collapses the dynamic directed flag complex to its undirected specialization; restricting to pairwise interactions yields a directed graph; removing both yields an undirected graph. This naturally produces a hierarchy of relational assumptions, and our baselines are structured to mirror the theoretical expressivity ladder established in the paper:
>
> no relational information (DS) → undirected pairwise (GNN) → directed pairwise (Dir-GNN) → undirected higher-order (MPSNN) → directed higher-order (SSN).
>
> This progression isolates where directed and higher-order structure becomes beneficial, both empirically and in relation to our theoretical results.
>
> Under this data regime, the methods in [1–3] aren’t applicable in a meaningful way and for meaningful reasons. In particular, discarding DAC directionality yields an undirected simplicial complex, which is also already a regular cell complex—exactly the domain for which Cin++ and the Cellular Transformer are designed. In other words, the domain to which [1][3] would be applied is the same as MPSN, i.e., the same undirected simplicial complex (being a cell complex too). In this situation, for example, the message-passing operators of [1] reduce to the same boundary, coboundary, and adjacency relations used in MPSNNs, and would therefore behave identically or nearly so to our undirected higher-order baseline. CycleNet [2] proposes a structural positional-encoding module for undirected graphs. It is not designed for complexes. It can be reasonably viewed as a method tailored to (a) undirected graphs and (b) positional encodings, rather than as an architecture whose computations are constrained by higher-order structure. By contrast, our work focuses on structural architectures that encode inductive biases directly in the message-passing operators, not on graph positional encodings. Moreover, the data regime we study is fundamentally different: we work with directed simplicial data. Overall, although it is true that [2][3] aren’t 1-1 instances of MPSNs, they would add no value to the hierarchy of inductive biases described above, as including them would not probe the directed higher-order phenomena that are central to our contributions (and that allow SSNs to reach SotA results).
>
> In short, our choice of baselines is dictated by the theoretical hierarchy developed in the paper. The proposed alternatives either collapse to existing baselines under our data assumptions or lie outside the question our work is designed to investigate. We are happy to discuss this more and provide further clarifications if needed.

---

> ### Author Response · Authors · 2025-11-18
>
> # [W2]
>
> We thank the reviewer for the suggestion. In our setting, the choice of relations is indeed central—both theoretically and empirically—and the paper already includes a controlled ablation at the level that is most meaningful for our theory (as discussed in [W1]). Specifically, we compare (i) the full family of direction-aware adjacencies present in DACs with (ii) their undirected specializations obtained by making directional structure opaque. This directly tests whether exploiting directed higher-order relations provides a measurable benefit. The empirical results mirror the theoretical predictions: SSNs with direction-aware adjacencies consistently outperform their symmetrized counterparts, especially on genuinely directed datasets. This shows that finer-grained direction-aware relations contribute substantially to performance.
>
> If the reviewer’s concern pertains to a finer-grained ablation over individual directed relations—such as evaluating all subsets of the six triangle adjacencies in dimension 2—this is combinatorially and computationally demanding. Instead, Routing-SSN provides a scalable proxy: by activating only the top-k relations per layer, it learns which relations the model finds informative while staying within a manageable computational budget.
>
> In summary, the paper evaluates relation choice at the theoretically relevant level—directed versus undirected adjacencies—and Routing-SSN might offer an implicit, computationally tractable mechanism for probing more granular contributions. This said, we agree with the reviewer that the SotA results achieved by SSNs motivate further work on their interpretability, which could benefit neuroscientific arguments as well. However, we believe this is material for a whole other paper.

---

> > ### Comment · Reviewer_7KeL · 2025-11-27
> > **Response to authors**
> >
> > I thank the authors for their detailed response, they have addressed all of my comments and questions. I would like to maintain my positive score.

---

> > > ### Author Response · Authors · 2025-11-28
> > >
> > > Thank you for the time and care you put into reviewing our paper. We are glad that the revisions addressed all your comments and questions, and we appreciate your positive recommendation.

---

### Official Review · Reviewer_jgH5 · 2025-11-01

**Soundness:** 3
**Presentation:** 2
**Contribution:** 3
**Rating:** 6
**Confidence:** 3

**Summary:**

The paper proposes Semi-Simplicial Neural Networks (SSNs), a TDL architecture that operates on semi-simplicial sets rather than classical simplicial complexes, enabling joint modeling of directionality and higher-order structure. It further introduces Dynamical Activity Complexes (DACs) to represent time-varying neuronal co-activation and provides formal results showing SSNs can recover a broad class of topological invariants. Experiments span brain-dynamics classification, traffic edge-regression, and node classification.

**Strengths:**

- The paper gives clean algebraic definitions, distinguishes orientation vs directionality, and proves that SSNs strictly contain the expressive power of GNNs/Dir-GNNs/MPSNNs while being able to recover key invariants.

- Using semi-simplicial sets allows the model to treat different vertex orderings as distinct simplices, preserving directionality information that is lost in traditional TDL architectures.

- The experiments span three application domains: brain dynamics, traffic flow regression, and node classification. The results are consistent with the theoretical claims, showing that the advantage of SSNs increases with the strength of directionality in the data.

**Weaknesses:**

1. Although the paper presents DACs as a novel construct, similar formulations of dynamic higher-order connectivity already exist in prior work such as [1]. The paper should clarify what is genuinely new.

2. The semi-simplicial formalism is conceptually close to combinatorial complexes [2] and several Combinatorial-Complex Neural Networks (CCNNs) already exist [3], [4]. The paper lacks a conceptual and empirical comparison, leaving ambiguity as to whether SSNs represent a subset, superset, or merely a re-parameterization of these frameworks.

3. The latest higher-order model compared is MPSNN (2021). More recent approaches such as combinatorial-complex networks, cell-complex GNNs, or hypergraph GNNs are not included.

4. In the brain-related experiments, 2-simplices (triangles) are included, but in edge regression and node classification, only 1-simplices are used. The absence of experiments with higher-order (>2) simplices weakens the empirical evidence supporting the claimed expressivity advantages.

[1] Higher-order connectomics of human brain function reveals local topological signatures of task decoding, individual identification, and behavior (Nature Communications, 2024)

[2] Combinatorial Complexes: Bridging the Gap Between Cell Complexes and Hypergraphs

[3]Topological Deep Learning: Going Beyond Graph Data

[4] TopoTune: A Framework for Generalized Combinatorial Complex Neural Networks

**Questions:**

Can the proposed SSNs be generalized to combinatorial complexes with cells of dimension greater than two?

Can DACs be extended from binary activation to real-valued activity?

---

> ### Author Response · Authors · 2025-11-18
>
> We thank the reviewer for the thoughtful and constructive evaluation. We appreciate the recognition of the paper’s formal clarity, its treatment of directionality, and the alignment between theory and experiments. The points raised are helpful clarifications, and articulating them more explicitly in the paper will further strengthen its presentation and positioning.
>
> # [Q1 + W2]
>
> We thank the reviewer for prompting a clearer discussion of the relationship between Combinatorial Complexes (CCs) [1] and Semi-Simplicial Sets (SSSs) . As noted in l.1064, we already cite this line of work and briefly emphasize the key distinction: CCs encode ranked, unordered set-type relations, whereas SSSs provide strictly richer topological structure by encoding ordered simplices equipped with coherent face maps. The order-awareness in SSSs enables directionality in higher-order interactions, which CCs fundamentally cannot represent. Thus, neither CCs nor SSSs subsume each other, as they encode fundamentally different combinatorial structures. Given the popularity of CCs in recent TDL research, we agree that making this distinction explicit strengthens the paper.  We have therefore extended the explanation in l.1064 to clarify this distinction more clearly (see C. Related Works, Topological Deep Learning paragraph, highlighted in blue). We provide a clarification below.
>
> **(Combinatorial complexes: Formal definition from [1]).**
>
> A CC is a triple $(S,X,rk)$ where $S$ is a set, $X \subseteq P(S) \{∅}$ is a family of non-empty subsets, and $rk: X→Z≥0$ satisfies:
> 1. for all $s \in S, {s} \in X$, and
>
> 2. if $x \subseteq y$, then $rk(x) \leq rk(y)$.
>
> From this definition, CC cells are sets and therefore unordered. As a consequence, CCs cannot distinguish different vertex orderings of the same collection of vertices and cannot encode directional or order-sensitive higher-order interactions. For this reason, structures such as directed graphs, directed simplicial complexes, ordered motifs, or tournaplexes cannot be represented as CCs.
>
> CCs relax the structural requirements of classical topological domains, such as simplicial or cellular complexes, while retaining a basic set-theoretic viewpoint in which cells are subsets of vertices endowed with a rank or hierarchy. This reduction of axioms explains why CCs can subsume several classical objects: by discarding additional algebraic constraints, CCs keep only the minimal structure needed to recover the augmented poset used for unsigned message propagation in traditional topological architectures (e.g., boundary/co-boundary and upper/lower adjacencies in MPSNNs [2] and CCNNs [3]).
>
> However, this expressiveness stems precisely from weakening classical structural constraints. In CCs there is no notion of orientation, no signed incidence structure, and no boundary or Hodge operators, in the algebraic-topological sense, and most importantly for our setting, there is no mechanism for encoding higher-order directionality.
>
> **Semi-Simplicial Sets**  provide exactly the missing structure while remaining formal algebraic objects that retain many classical algebraic-topological tools. They allow multiple ordered simplices on the same vertex set together with coherent face maps that preserve this order and satisfy the simplicial identities. This structure enables principled modeling of directionality and order-aware higher-order interactions. The presence of ordered faces and their algebraic coherence is essential both for the theoretical guarantees we establish and for the applications considered in our work.
>
> **Expressive relation (neither superset nor subset).** The structural differences described above imply that CCs and SSSs capture different domains of objects and therefore neither framework subsumes the other.
>
> **References**
>
> [1] Hajij et al., Topological Deep Learning: Going Beyond Graph Data.
>
> [2] Bodnar et al., Weisfeiler and Lehman Go Topological: Message Passing Simplicial Networks.
>
> [3] Bodnar et al., Weisfeiler and Lehman Go Cellular: CW Networks.

---

> ### Author Response · Authors · 2025-11-18
>
> # [Q2]
>
> We thank the reviewer for this valuable question. Below, we clarify (i) why DACs are introduced in binary form for our brain-activity decoding setting, and (ii) how the construction naturally generalizes to real-valued neuronal signals.
>
> ### Why DACs use binary activations in the brain-activity decoding task
>
> **Theoretical and experimental compatibility with neurotopological pipelines.**  DACs are defined as binary, time-evolving directed simplicial complexes that explicitly encode neuronal co-activation events, ensuring methodological compatibility with prior neurotopology works. In this setting, DACs (i) preserve isomorphisms and (ii) encode the full temporal sequence of functional complexes used in these studies (Propositions 4–5, Appendix H.2). These properties are crucial for our theoretical analysis: they enable the formal guarantees showing that SSNs can provably recover neurotopological invariants associated with stimulus identity (Theorem 4), whereas previous models that treat directionality or higher-order interactions in isolation cannot. The binary formulation, therefore, provides a robust real-world setting in which our theoretical contributions can be validated unambiguously.
>
> **Fair comparison with undirected simplicial-complex baselines.** Because a simplex is active iff all its constituent neurons are active, the symmetrization used in undirected baselines induces a well-defined and compatible symmetrization of the associated binary signals. Structure and features are therefore projected consistently. This cleanly isolates the structural effect of directionality: activation depends only on the set of constituent neurons, so all directional influence enters exclusively through the higher-order directional information encoded by the semi-simplicial structure, rather than through feature-engineering choices. This separation makes both the empirical results and the theoretical guarantees easier to interpret, while still leaving room for richer signal formulations when applications require them.
>
> ### Extension to real-valued activity.
>
> At the same time, our methodological formalism naturally accommodates more expressive liftings in which simplex features depend on ordered structure—for example, direction-dependent or order-aware aggregations—allowing richer signal representations when the task at hand requires them.
>
> In particular, the DAC construction generalizes directly to continuous neuronal signals. Let $G = (V, E)$ be a directed structural graph, and let $B : V \to \mathbb{R}^T$ assign a real-valued time series $B_t(v)$ to each vertex. Let $K_G$ be the directed flag complex of $G$, and let $\Phi : \mathbb{R}^{n+1} \to \mathbb{R}$ be an aggregation operator (e.g., mean). For each simplex $\sigma = (v_0,\ldots,v_n)$, define $\tilde{B}(\sigma)_t = \Phi(B_t(v_0), \ldots, B_t(v_n))$, and let $\tilde{B} = (\tilde{B}_1,\dots,\tilde{B}_T)$.
>
> The resulting real-valued DAC, $K_{G_\tilde{B}} = (K_G, \tilde{B})$, preserves the semi-simplicial structure while incorporating continuous neuronal intensities. Different choices of $\Phi$ enable permutation-invariant, order-aware, or causally informed simplex-level signals, depending on the dataset and neuroscientific hypothesis.
>
> More generally, one may incorporate temporal structure by using $\Phi : (\mathbb{R}^{n+1})^{W} \to \mathbb{R},$ $W \le T$, allowing $\Phi$ to process the joint activity of the vertices of $\sigma$ across a sliding window of $W$ consecutive time steps. This enables DACs to encode temporally smoothed statistics, short-term dynamics, or higher-order cross-neuron correlations, opening new avenues for modeling dynamic neural codes.

---

> ### Author Response · Authors · 2025-11-18
>
> # [W1]
>
> We thank the reviewer for the observation. The method in [1] and our DAC construction differ fundamentally in both mathematical structure and purpose, and we will explicitly clarify this distinction in the revised manuscript (see C. Related Works, Neurotopology paragraph, highlighted in blue).
>
> The framework in [1] derives weighted, undirected simplicial complexes by algebraically combining low-order fMRI signals (e.g., element-wise products) to generate higher-order co-fluctuation time series. These higher-order interactions are reconstructed from low-order data and embedded into a simplicial complex solely as weights.
>
> By contrast, DACs introduce a new object: a directed simplicial complex obtained from the directed structural connectome via the directed flag construction, equipped with binary, non-reconstructed time-varying activations that encode explicit co-activation events. DACs preserve digraph isomorphisms, retain full temporal functional structure, and—crucially—are designed to be processed by Semi-Simplicial Neural Networks, enabling the theoretical guarantees we provide (Props. 4–5, Thm. 4).
>
> Thus, while both frameworks are dynamic and higher-order, [1] produces weighted undirected complexes from signal, whereas DACs define directed higher-order relational structure coupled with genuine activity and tailored to topological deep learning and SSNs. The mathematical objects, their information content, and their role within learning architectures are therefore distinct.

---

> ### Author Response · Authors · 2025-11-18
>
> # [W3]
>
> DACs preserve the full temporal evolution of functional complexes and encode explicit co-activation events, matching the constructions used to compute neurotopological invariants in prior neuroscience studies. Working in this meaningful, real-world but controlled setting allows us to test our theoretical guarantees under the exact conditions in which these invariants are defined and interpreted—namely on binary, dynamic, directed flag complexes.
>
> When directionality is removed via symmetrization, the directed flag complex underlying a DAC reduces to its natural undirected specialization, which is precisely an undirected simplicial complex. This object is simultaneously a regular cell complex and a combinatorial complex (see [W2–Q1]). As a consequence, applying a cellular- or combinatorial-complex neural network to this symmetrized structure collapses to exactly the same boundary, coboundary, and adjacency operators used in MPSNNs [2]. In this regime, all these architectures behave identically to MPSNNs and therefore do not offer additional modeling capacity for our setting. For hypergraph GNNs, the situation is similar: after symmetrization and the loss of hierarchical structure, a DAC reduces to a hypergraph where nodes exchange information with hyperedges only through shared vertices. This removes the hierarchical topological structure (including upper adjacencies) and erases all information about the relational type of higher-order interactions.
>
> Looking beyond the present setting, other neural data modalities and applications may motivate different liftings or higher-order representations—including hypergraphs, cellular complexes, or combinatorial complexes—whenever their structural assumptions align with the available data. Each application, hypothesis and neural data modality carries its own neuroscientific hypotheses and would require dedicated data representation, which lies outside the scope of this work. Directed structure inferred from signal-based directed correlation measures in fMRI is one such example that could naturally extend our framework to be applied to alternative neural data modalities. Nonetheless, these possibilities highlight the broader landscape in which semi-simplicial architectures operate and point toward promising future directions at the interface of topology, learning, and neural data.

---

> ### Author Response · Authors · 2025-11-18
>
> # [W4]
>
> We thank the reviewer for raising this point. The restriction to 1-simplices in the edge-regression and node-classification tasks is deliberate and arises from the need for fair and controlled comparisons with existing directed-graph baselines.
>
> **Use of 1-simplices.** For the edge regression task, we match the experimental setting of [1], whose architecture is explicitly designed for directed graphs with edge signals and whose desiderata concern orientation equivariance/invariance for edge features. To ensure a fair comparison, we restrict SSNs to the same information domain (1-simplices) and evaluate on EIGN’s own benchmark (Appendix I.3). Higher-order simplices are not present in their datasets, and introducing them would give SSNs access to information unavailable to EIGN.  For node classification, we evaluate against higher-order baselines such as MPSNNs [2], which operate up to dimension-2 simplices. In this setting, SSNs—restricted to directed 1-simplices—achieve competitive or superior performance, especially on chain-like directed benchmarks such as DRE. ​​This highlights that directionality in lower-dimensional simplices can provide a stronger and more informative signal than higher-order undirected structure when the underlying data exhibit directed relational patterns. We also conducted exploratory experiments including 2-simplices, obtaining marginal improvements, so we opted for the more controlled and efficient 1-simplex setting.
>
> **Improved expressivity w.r.t. to undirected models and directed graph models does not require >=2 simplices.** Crucially, our expressivity results do not necessarily rely on the presence of higher-dimensional simplices. In particular, explicitly using directed 1-simplices (directed edges) already induces a strictly richer inter-cell adjacency structure than undirected edges: SSNs distinguish the ordered roles of vertices in each 1-simplex and propagate information through directed face relations, whereas undirected message-passing schemes communicate across edges that share a vertex but cannot distinguish which vertex or in which order (Fig. 12). Thus, the expressivity advantages of SSNs w.r.t. to undirected (both graph and higher-order) models do not require higher-order simplices. Similarly, explicitly including 1-simplices is also sufficient to improve expressivity over directed graph (i.e., node-only) models. However, higher-order simplices are useful to distinguish more nuanced directed complexes.
>
> [1] Fuchsgruber et al., Graph Neural Networks for Edge Signals: Orientation Equivariance and Invariance.
> [2] Bodnar et al., Weisfeiler and Lehman Go Topological: Message Passing Simplicial Networks.

---

> ### Author Response · Authors · 2025-11-28
>
> Thank you for the thoughtful and constructive evaluation of our work. Your questions offered helpful opportunities to clarify and further strengthen the presentation and positioning of the paper. We have refined the text accordingly.
>
> We believe we have addressed all your concerns in the responses, and we would be grateful for any further feedback you might have. Your perspective would help us ensure that everything is resolved clearly and accurately.
>
> Thank you again for the time and care you invested in reviewing the paper.

---

### Official Review · Reviewer_DBfX · 2025-11-01

**Soundness:** 3
**Presentation:** 3
**Contribution:** 3
**Rating:** 8
**Confidence:** 4

**Summary:**

This paper introduces Semi-Simplicial Neural Networks (SSNs), a new class of topological deep learning models for directed higher-order structures. SSNs operate on semi-simplicial sets using face-map–induced relations to propagate messages between simplices, extending both GNNs and simplicial neural networks. A scalable variant, Routing-SSN, dynamically selects relevant relations.
The authors prove that SSNs are strictly more expressive than directed GNNs and simplicial networks under the WL hierarchy and can recover a richer family of topological invariants relevant to brain activity. Experiments on 13 datasets show large gains over prior models (up to 50% over GNNs and 27% over other TDL baselines).

**Strengths:**

- The paper is well written and easy to follow.

- The paper provides strong theoretical results on WL-expressivity, permutation equivariance, and invariant recovery, establishing clear theoretical advantages over prior models.

- The integration with brain dynamics modeling via Dynamical Activity Complexes (DACs) connects deep learning with neurotopology in a rigorous, data-driven way, replacing handcrafted invariants.

- SSN is able to achieve large accuracy gains (up to 50% over GNNs) on challenging stimulus classification tasks, validating the theory with well-controlled baselines and consistent robustness.

- The Routing-SSN mechanism provides a practical path to scaling directed topological models while retaining expressivity and efficiency.

**Weaknesses:**

- Recent literature has shown that transformers can be considered message passing neural networks [1]. However, results using multi-head attention over node features are missing. Since SSNs generalize message passing to relation-aware updates, such attention-based comparisons would clarify whether the proposed relational framework yields benefits beyond architectural scaling, or the transformer model is able to learn relations.

- The ablation on relation classes (e.g., face-map–induced vs. interdimensional vs. intradimensional) is not reported. Understanding which classes contribute most to performance would help assess the inductive bias of the relational algebra and the necessity of routing.

- Several of the paper’s most important theoretical results, including the corollaries, are deferred to long appendices. Condensing or summarizing them in the main text would improve accessibility and emphasize the technical depth.

- On standard node classification benchmarks (e.g., cora), SSNs do not outperform existing baselines, and these graphs are relatively small. This weakens the general claim that SSNs are broadly superior rather than domain-specific to brain networks. Results on datasets like OGB would emphasize the scalability and generalizability of the model.

[1] Joshi, Chaitanya K. "Transformers are graph neural networks." arXiv preprint arXiv:2506.22084 (2025).

**Questions:**

- Which face-map–induced relation classes (e.g., interdimensional, intradimensional, directed adjacency) contribute most to performance?

- The framework mainly uses binary activations for brain data. How would SSNs handle continuous-valued features or multimodal signals?

- How will the routing behave under noise or shuffled connectivity?

---

> ### Author Response · Authors · 2025-11-18
>
> We thank the reviewer for the thoughtful, constructive and positive assessment. We appreciate the recognition of the paper’s theoretical contributions and the novel integration with brain-dynamics modeling, connecting topological deep learning with neurotopology in a rigorous way. We are also grateful for the positive feedback regarding the clarity of presentation and the empirical gains achieved by our models. The reviewer raises several insightful questions that help sharpen both the conceptual framing and the empirical evaluation, and we address them in detail below.
>
> # [Q1 + W2]
>
> We thank the reviewer for the question. In our paper, we work within a principled experimental setting that allows us to test both the general expressivity results for SSNs (Thm. 1–2) and the invariant-recovery guarantees specialized to DACs (Thm. 4). DACs are dynamic, directed flag complexes encoding the full temporal evolution of functional complexes (Prop. 4–5), matching the constructions used in prior neurotopology work to compute topological invariants for brain-activity decoding.
>
> Within this setting, the relevant face-map–induced relations are both interdimensional (lower/upper adjacencies across simplex dimensions) and intradimensional (adjacencies between simplices of the same dimension). If directionality is removed, a DAC collapses to its undirected simplicial specialization; if higher-order structure is removed, it collapses further to a directed or undirected graph. This yields a natural ladder of relational assumptions:
>
> DeepSets -> no intra/interdimensional relations
>
> GNN -> intradimensional, undirected (1-simplices only)
>
> Dir-GNN -> intradimensional, directed
>
> MPSNN -> intradimensional + interdimensional, undirected
>
> SSN -> intradimensional + interdimensional, directed
>
> This progression constitutes a theoretically meaningful ablation: each baseline removes or symmetrizes a specific family of face-map–induced relations. This allows us to pinpoint exactly where directional and higher-order information begins to matter.
>
> Empirically, the pattern is clear across datasets: SSNs, which jointly exploit directed intradimensional relations and directed interdimensional (lower/upper) adjacencies, consistently outperform all symmetrized or lower-order variants. These gains mirror the theoretical predictions, demonstrating that direction-aware relational structure across dimensions is crucial in genuinely directed higher-order domains.
>
> A finer-grained ablation over individual directed adjacencies (e.g., enumerating all subsets of the six triangle relations in dimension 2) is combinatorially infeasible. In this regard, Routing-SSN already serves as a scalable proxy: by selecting the top-k relations at each layer, it highlights which relation families the model consistently uses. Building a full interpretability framework, however, lies beyond the scope of the present work.

---

> ### Author Response · Authors · 2025-11-18
>
> # [Q2]
>
> We thank the reviewer for this valuable question. Below, we clarify (i) why DACs are introduced in binary form for our brain-activity decoding setting, and (ii) how the construction naturally generalizes to real-valued neuronal signals.
>
> ### Why DACs use binary activations in the brain-activity decoding task
>
> **Theoretical and experimental compatibility with neurotopological pipelines.**  DACs are defined as binary, time-evolving directed simplicial complexes that explicitly encode neuronal co-activation events, ensuring methodological compatibility with prior neurotopology works. In this setting, DACs (i) preserve isomorphisms and (ii) encode the full temporal sequence of functional complexes used in these studies (Propositions 4–5, Appendix H.2). These properties are crucial for our theoretical analysis: they enable the formal guarantees showing that SSNs can provably recover neurotopological invariants associated with stimulus identity (Theorem 4), whereas previous models that treat directionality or higher-order interactions in isolation cannot. The binary formulation, therefore, provides a robust real-world setting in which our theoretical contributions can be validated unambiguously.
>
> **Fair comparison with undirected simplicial-complex baselines.** Because a simplex is active iff all its constituent neurons are active, the symmetrization used in undirected baselines induces a well-defined and compatible symmetrization of the associated binary signals. Structure and features are therefore projected consistently. This cleanly isolates the structural effect of directionality: activation depends only on the set of constituent neurons, so all directional influence enters exclusively through the higher-order directional information encoded by the semi-simplicial structure, rather than through feature-engineering choices. This separation makes both the empirical results and the theoretical guarantees easier to interpret, while still leaving room for richer signal formulations when applications require them.
>
> ### Extension to real-valued activity.
>
> However, our methodological formalism naturally accommodates more expressive liftings in which simplex features depend on ordered structure—for example, direction-dependent or order-aware aggregations—allowing richer signal representations when the task at hand requires them.
>
> In particular, the DAC construction generalizes directly to continuous neuronal signals. Let $G = (V, E)$ be a directed structural graph, and let $B : V \to \mathbb{R}^T$ assign a real-valued time series $B_t(v)$ to each vertex. Let $K_G$ be the directed flag complex of $G$, and let $\Phi : \mathbb{R}^{n+1} \to \mathbb{R}$ be an aggregation operator (e.g., mean). For each simplex $\sigma = (v_0,\ldots,v_n)$, define $\tilde{B}(\sigma)_t = \Phi(B_t(v_0), \ldots, B_t(v_n))$, and let $\tilde{B} = (\tilde{B}_1,\dots,\tilde{B}_T)$.
>
> The resulting real-valued DAC, $K_{G_\tilde{B}} = (K_G, \tilde{B})$, preserves the semi-simplicial structure while incorporating continuous neuronal intensities. Different choices of $\Phi$ enable permutation-invariant, order-aware, or causally informed simplex-level signals, depending on the dataset and neuroscientific hypothesis.
>
> More generally, one may incorporate temporal structure by using $\Phi : (\mathbb{R}^{n+1})^{W} \to \mathbb{R},$ $W \le T$, allowing $\Phi$ to process the joint activity of the vertices of $\sigma$ across a sliding window of $W$ consecutive time steps. This enables DACs to encode temporally smoothed statistics, short-term dynamics, or higher-order cross-neuron correlations, opening new avenues for modeling dynamic neural codes.

---

> ### Author Response · Authors · 2025-11-18
>
> # [Q3]
>
> We thank the reviewer for this interesting question. Routing-SSN might inherit some of the general behavior of sparse MoE mechanisms: top-k gating is input-dependent, and perturbations can change which experts are selected. Although a clear and thorough understanding of the robustness properties of MoE under noise is an open problem, several works provide relevant insights [1–3]. Because sparse routing is inherently discontinuous, small input perturbations may switch the active experts, thus leading to poor robustness, especially if the experts implement very different functions. At the same time, prior work also shows that MoE architectures can be more robust than dense models when expert specialization provides redundancy or when individual experts are themselves robust [1]. Router-level regularization and adversarial training have been proposed as effective stabilizing strategies [2,3].
>
> If by shuffled connectivity the reviewer means permutations of simplex indices, Routing-SSN is unaffected: the architecture is fully permutation-equivariant (Thm. 3), relation types are preserved under relabeling, and routing scores remain unaltered. If instead the reviewer refers to perturbations of the underlying topology—adding or removing simplices—then routing stability depends on the stability of the underlying relational message-passing operator (e.g., MPNN, GAT, SAGE) on which the expert updates are built.
>
> In summary, studying the stability of routing under topological perturbations is an important open problem. Addressing it would require extending existing MoE robustness frameworks to graph/topological architectures, which lies beyond the present scope. We view this as a valuable direction for future research and appreciate the reviewer highlighting it.
>
> [1] Puigcerver, Joan, et al. On the adversarial robustness of a mixture of experts.
>
> [2] Pavlitska, Svetlana, et al. Robust Experts: the Effect of Adversarial Training on CNNs with Sparse Mixture-of-Experts Layers.
>
> [3] Kada, Masahiro, et al. Robustifying Routers Against Input Perturbations for Sparse Mixture-of-Experts Vision Transformers.

---

> ### Author Response · Authors · 2025-11-18
>
> # [W1]
>
> We thank the reviewer for highlighting the connection between Transformers and message passing, and for the principled suggestion. As noted in [1], multi-head attention in Transformers (up to positional/structural encodings) is formally equivalent to message passing in Graph Attention Networks (GATs) with query/key/value attention: Transformers can be viewed as specific GATs operating on a complete graph, and conversely, GATs are Transformers with attention masks induced by a given neighborhood structure.
>
> We already include attention-based variants of all relevant baselines in Appendix I.1.2. Since our framework is relation-general, any message-passing update (including SSNs) can be instantiated with attention by replacing the aggregation operators with any attention-based operator. These attention-based models were included for completeness rather than as part of the main theoretical narrative, but they directly address the reviewer’s concern.
>
> The results are:
>
> (4, 325): GAT: 22.46 ± 1.48 | DIR-GAT: 42.03 ± 0.31 | MPSNN-GAT: 32.97 ± 8.59 | SSN-GAT: 77.45 ± 3.58
>
> (M = 3): GAT: 24.42 ± 0.62 | DIR-GAT: 28.59 ± 0.62 | MPSNN-GAT: 29.55 ± 0.81 | SSN-GAT: 51.27 ± 1.85
>
> These findings show that attention alone does not remove the need for richer relational structure: even under attention-based updates, SSNs remain substantially stronger than attention-equipped GNNs or MPSNNs. This supports the central claim that the gains arise from the relational algebra—directed, interdimensional, and intradimensional adjacencies—rather than from architectural scaling through attention.
>
> If the reviewer is specifically suggesting a Transformer operating on the complete graph as an additional baseline, we here present the results on the brain activity decoding tasks obtained with a standard stack of Transformer Decoder layers with 4 attention heads in each layer.
>
> TASK 1: (4,325) 19.2881 +- 0.8493 | (4,125) 12.1468 +- 0.4810 | (8,175) 12.0133 +- 0.9071.
>
> TASK 2: M=1 25.8065 +- 0.7747 | M=3 24.5235 +- 0.7727 | M=5 25.5706+-0.4353.
>
> As the reviewer can notice and as expected, results remain poor, given the lack of access to directed and (inductively biased) higher-order information. This could be (only) partly compensated by extensively exploring and testing positional/structural encodings, but we believe it falls way beyond the scope of this paper. Ideally, studying novel classes of positional/structural encodings able to capture higher-order directionality as SSNs do could be itself an (interesting) paper to work on. If the reviewer is instead referring to some specific Graph Transformer architecture and/or class of structural encodings, we are happy to continue this conversation further.

---

> ### Author Response · Authors · 2025-11-18
>
> # [W3]
>
> We thank the reviewer for the suggestion. The main text states all core theoretical results (Theorems 1–4), while the complete proofs are placed in the appendices to preserve readability and avoid overloading the exposition with technical detail. If the reviewer has specific suggestions on which results should be highlighted more, we are more than happy to incorporate the required modifications in the revised manuscript.
>
> # [W4]
>
> We thank the reviewer for raising this point. The behavior of SSNs on standard node-classification benchmarks reflects a broader issue in graph ML: GNNs help only when their structural assumptions match the data [2], and many commonly used benchmarks suffer from fundamental limitations [4]. In the directed setting, this problem is particularly acute. Datasets such as Chameleon and Squirrel were originally released as undirected graphs [1], raising concerns about their suitability for evaluating direction-aware models. Prior work ([2], [5])—and our own results—indicate that Cora-ML and Citeseer contain almost no meaningful directional information: 94–98% of higher-order edge directions collapse into a single trivial pattern. In such cases, direction-aware models, including SSNs, offer no systematic benefit, and this is consistent with the underlying data rather than with model limitations.
>
> By contrast, when meaningful directional and higher-order structure is present, SSNs excel. Roman-Empire exhibits strong, chain-like asymmetries, and SSNs achieve state-of-the-art performance there (App. I.3, Tables 13–14). Similarly, on the direction-sensitive traffic task (App. I.2, Table 11), SSNs match or surpass [6] despite not being designed for the benchmark and under a fair comparison restricted to node- and edge-level signals. The broader challenge is that TDL currently lacks benchmarks where topological structure genuinely matters [20].
>
> This motivates one of the core contributions of our work: a principled, meaningful real-world benchmark where directionality and higher-order interactions are not assumed but validated by domain experts as biologically meaningful. DACs encode dynamic directed flag complexes derived from neural activity, and in this setting, SSNs are the only architecture capable of jointly capturing the required directed and higher-order structure. As Table 3 shows, neither directionality alone (Dir-GNN), nor higher-order structure alone (MPSNN), nor their symmetrized variants can solve the task.
>
> Regarding OGB, we note that many OGB datasets are pairwise and undirected, and therefore do not probe the directed higher-order inductive biases that SSNs are designed to leverage. Among the directed OGB datasets, arXiv is strongly homophilic, and direction-aware models such as DIR-GNN already underperform there—indicating that directional structure plays little role in its topology. The two remaining directed datasets, *papers100M* and *mag*, contain tens of millions to billions of edges, which makes full SSN training computationally prohibitive in the standard evaluation pipeline. Although this is surely a matter of (extreme) scalability, these datasets are usually employed to test GNNs that are natively designed (multi-GPU, sampling-based, with optimized kernels,…) to scale at the billion-edge scale. Unfortunately, testing on these benchmarks is unfeasible given our computational resources, even with the Routing SSN.
>
> [1] Rozemberczki, B., et. al. Multi-scale attributed node embedding.
>
> [2] Bechler-Speicher, M., et al. Graph Neural Networks Use Graphs When They Shouldn’t.
>
> [3] Zhang, X. et al., MagNet: A Neural Network for Directed Graphs.
>
> [4] Bechler-Speicher, M., et al, Position: Graph Learning Will Lose Relevance Due To Poor Benchmarks.
>
> [5] Rossi, E., et. al., Edge directionality improves learning on heterophilic graphs.
>
> [6] Fuchsgruber, D., et al., Graph Neural Networks for Edge Signals: Orientation Equivariance and Invariance.

---

> ### Author Response · Authors · 2025-11-28
>
> Thank you for the constructive feedback and for engaging so thoughtfully with our paper. We especially appreciated your suggestion regarding a multi-head Transformer baseline on the nodes; the results align with our finding that attention alone cannot recover the higher-order directed inductive biases that SSNs are designed to capture in complex tasks.
>
> We believe we have addressed all of your concerns, and we would be grateful for any further feedback you may have, as your perspective would help ensure that everything is fully resolved.
>
> Thank you again for your positive evaluation and for the time and care you invested in reviewing the paper.

---

### Author Response · Authors · 2025-12-01
**General Author Response to the Area Chair**

**TL;DR.** Across all reviews, the paper is consistently assessed as theoretically strong, novel, clearly written, and empirically robust. None of the raised points were objections regarding correctness, novelty, or validity; they were primarily requests for clarification. All concerns have been addressed in full, and we believe no unresolved issues remain. We thank the AC for the time and care invested in evaluating our work, especially under these unusual circumstances.

---

Dear Area Chair,

Thank you for the time and care you have invested in reviewing our paper, particularly under the unusual circumstances of this review cycle. Since the paper has now been reassigned, we provide below a concise summary of the reviewers’ assessments and the questions raised, along with how we addressed them, in the hope that it helps streamline your evaluation.

### Overall assessment and strengths

Reviewers evaluated the paper consistently positively. The work was recognized as theoretically strong, novel in its use of semi-simplicial sets for directed higher-order learning, clearly written, and empirically robust.

**Theoretical and methodological assessment.**

Reviewers *jgH5* and *7KeL* emphasized the methodological novelty of using semi-simplicial sets to model directed higher-order structure—preserving directionality that traditional TDL architectures necessarily lose. *All reviewers* underscored the depth of the theoretical results, including WL-expressivity, permutation-equivariance, and invariant-recovery guarantees. Reviewer *DBfX* also highlighted the novel, principled, and rigorous integration of topological deep learning with neuroscience. In addition, *DBfX* noted the clarity and accessibility of the presentation, while reviewer *jgH5* praised the clean theoretical exposition—reflecting our efforts to maintain full mathematical precision while ensuring accessibility for practitioners.

**Empirical assessment.**

Reviewers *7KeL* and *jgH5* remarked on the breadth and consistency of the empirical evaluation across datasets, tasks, and baselines. Reviewer *jgH5* noted that the empirical advantage of SSNs increases with the strength of directionality in the data, in line with the theoretical predictions. *All reviewers* recognized the substantial performance gains achieved on the challenging brain-dynamics classification tasks, further validating the theoretical insights.


### Questions, comments and concerns

**Comments.**

Importantly, none of the reviewers raised concerns about correctness, novelty, or empirical validity. Their questions were primarily clarifications, such as:

(i) the scope of DACs and their extension to real-valued signals (*DBfX*; *jgH5*);

(ii) the distinction between semi-simplicial sets and combinatorial complexes (*jgH5*);

(iii) clarification of the relational assumptions underlying our baseline hierarchy, and the applicability of other higher-order models to directed simplicial data (*DBfX*; *7KeL*; *jgH5*);

(iv) routing-SSN behavior and interpretability signals (*DBfX*; *7KeL*);

(v) and whether a multi-head Transformer baseline on nodes could match the inductive biases of SSNs (*DBfX*).

None of these required changes to the core content or conclusions of the paper.

**How we addressed them.**

We provided detailed, point-by-point clarifications, including:

(i) a precise explanation of how DACs differ from existing constructions, why they are the correct data model for our theoretical guarantees, and how they extend naturally to continuous neuronal signals (*jgH5* [Q2]);

(ii) an explicit distinction between combinatorial complexes and semi-simplicial sets, explaining why neither subsumes the other and why SSSs are the principled domain for directed higher-order structure (*jgH5* [Q1 + W2]; *DBfX* [Q2]);

(iii) a justification of the baseline hierarchy, noting that no existing architecture handles directed multi-way relations such as directed simplicial data, and that several recent higher-order models collapse—after symmetrization—into the same operators used by our baselines (*DBfX* [Q1 + W2]; *jgH5* [W3]; *7KeL* [W1,W2]);

(iv) clarifications regarding routing-SSNs and the interpretability of routing signals, while noting that a full interpretability theory is beyond the present scope (*DBfX* [Q3]; *7KeL* [Q1,Q2]);

(v) and results for the multi-head Transformer baseline, whose behavior aligns with our finding that attention alone cannot recover the directed higher-order inductive biases captured by SSNs (*DBfX* [W1]).

**Current status.** Given the above, we believe all reviewer concerns have been fully addressed in our responses and that every point raised has been resolved clearly and accurately. Because the discussion period was frozen shortly after we posted our clarifications, several reviewers likely did not have the opportunity to update or acknowledge the resolutions.

Thank you again for the time and consideration devoted to our submission.

The Authors

---

### Meta-Review · Area_Chair_qSJE · 2026-01-07

**Summary:**

The article presents a type graph neural network for directed higher order structures.

Reviews are generally positive and find the work provides strong theory, good framework, and convincing experiments.

Therefore I am recommending accept.

**Reviewer Concerns:**

Reviewer DBfX:
The ablation on relation classes (e.g., face-map–induced vs. interdimensional vs. intradimensional) is not reported.
-> Response in rebuttal + ``Building a full interpretability framework, however, lies beyond the scope of the present work.''

On standard node classification benchmarks (e.g., cora), SSNs do not outperform existing baselines, and these graphs are relatively small.
-> Response in rebuttal + ``Unfortunately, testing on these benchmarks is unfeasible given our computational resources, even with the Routing SSN.''

Reviewer jgH5:
similar formulations of dynamic higher-order connectivity already exist in prior work
-> Response in rebuttal
conceptually close to combinatorial complexes
-> Response in rebuttal

**Reviewer Scores:**

For each review, specify how you think the reviewer would have changed their score if they had been able to participate fully in the discussion.

Reviewer DBfX: 8 -> 8
Reviewer jgH5: 6 -> 6
Reviewer 7KeL: 6 -> 6

---

### Decision · Program_Chairs · 2026-01-26

Accept (Poster)